# Verbalized Graph Representation Learning: A Fully Interpretable Graph Model Based on Large Language Models Throughout the Entire Process

## Abstract

Representation learning on text-attributed graphs (TAGs) has attracted significant interest due to its wide-ranging real-world applications, particularly through Graph Neural Networks (GNNs). Traditional GNN methods focus on encoding the structural information of graphs, often using shallow text embeddings for node or edge attributes. This limits the model to understand the rich semantic information in the data and its reasoning ability for complex downstream tasks, while also lacking interpretability. With the rise of large language models (LLMs), an increasing number of studies are combining them with GNNs for graph representation learning and downstream tasks. While these approaches effectively leverage the rich semantic information in TAGs datasets, their main drawback is that they are only partially interpretable, which limits their application in critical fields. In this paper, we propose a verbalized graph representation learning (VGRL) method which is fully interpretable. In contrast to traditional graph machine learning models, which are usually optimized within a continuous parameter space, VGRL constrains this parameter space to be text description which ensures complete interpretability throughout the entire process, making it easier for users to understand and trust the decisions of the model. We conduct several studies to empirically evaluate the effectiveness of VGRL and we believe this method can serve as a stepping stone in graph representation learning. The source code of our model is available at `https://anonymous.4open.science/r/VGRL-7E1E`

## 1 Introduction

Many real-world graphs incorporate textual data, forming what are known as Text-Attributed Graphs (TAGs) (Yang et al., 2021). In TAGs, nodes represent textual entities such as papers, while edges denote relationships between them, such as citations or co-authorships. For instance, the Cora dataset can be modeled as a TAG, where each node represents a research paper, and the node attributes include features such as the paper's title, abstract, and keywords. By integrating textual attributes with graph topology, TAGs facilitate more effective representation learning, making them valuable for tasks like document classification, clustering (Wang et al., 2023), citation analysis, and recommendation systems (Zhu et al., 2021; Zhang et al., 2023a). This combination of textual and relational data offers deeper insights, especially when both content and connections are essential to the analysis.

Although traditional Graph Neural Network (GNN) models, such as Graph Convolutional Network (GCN) (Kipf & Welling, 2016) and Graph Attention Network (GAT) (Veličković et al., 2017), have achieved significant performance improvements across multiple tasks, they generally suffer from a lack of interpretability. As these models largely rely on complex network architectures and implicit feature learning processes, understanding their internal decision mechanisms and how specific features influence task outcomes becomes challenging, thereby limiting their transparency and trustworthiness in practical applications. To address this issue, researchers have proposed several interpretable GNN models. These interpretable methods can generally be divided into input interpretability, training process interpretability, and decision-making process interpretability. For

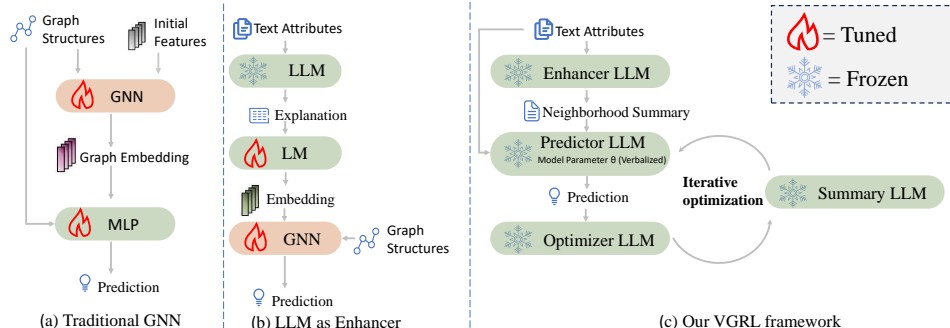

Figure 1: Comparison of Graph Representation Learning Methods **(a)** Traditional Graph Neural Networks (GNNs) rely on graph structures and initial features for embedding generation and prediction. **(b)** Incorporating a Language Model (LM) enhances GNNs, where a Large Language Model (LLM) provides explanations that refine the embedding process for improved predictions. **(c)** Our proposed Verbalized Graph Representation Learning (VGRL) framework introduces an iterative optimization process involving multiple frozen LLMs (Enhancer, Predictor, Optimizer, and Summary), emphasizing interpretability and parameter tuning through verbalized model adjustments.

example, GNNExplainer (Ying et al., 2019) is a method for input interpretability, which selects a small subgraph of the input graph together with a small subset of node features that are most influential for the prediction as an explanation, XGNN (Yuan et al., 2020) is a method for training process interpretability which reveals the basis of the model's predictions by generating interpretable graph structures, and SE-SGformer (Li et al., 2024a) is a method for decision-making process interpretability which constructs a novel explainable decision process by discovering the $K$-nearest (farthest) positive (negative) neighbors of a node for predicting edge signs. Clearly, while these methods all have a certain degree of interpretability, they can only explain one part of the entire process of model input, training, and output. Therefore, our goal is to implement comprehensive interpretability by simultaneously achieving input interpretability, training process interpretability, and decision-making process interpretability.

In recent years, with the breakthroughs of large language models (LLMs) in the field of natural language processing, researchers have gradually begun to integrate them with GNNs to enhance model performance and capabilities. For instance, LLMs can act as predictors (Tang et al., 2024), generating more accurate predictions by analyzing node features and structural information for the TAGs. Also, TAPE (He et al., 2023) prompts a powerful LLM to explain its predictions and serve explanations as supplementary text attributes for the downstream LMs and GNN models. Due to the powerful text inference capabilities of LLMs, they are capable of processing TAGs, reasoning about the node classification prediction process of TAGs, and generating explanations in text that is comprehensible to humans. Therefore, we consider the use of LLMs to achieve comprehensive interpretability. However, using LLMs to handle graph tasks and provide interpretability is not easy. Specifically, there are currently two main approaches to applying LLMs in the field of graph: one is to pre-train or fine-tune LLMs to adapt to various graph downstream tasks. But due to the vast number of parameters typically found in LLMs, the cost of fine-tuning LLMs is quite high and the training time is long. The second is to directly freeze the LLMs for inference but this method does not yield good results. For example, we directly froze the predictor LLMs for node classification prediction in subsequent experiments, and the prediction accuracy was generally not high, as shown in Table 3.

In summary, we face two major challenges to achieve comprehensive interpretability with LLMs:

**Challenge 1:** How can we ensure that a model is interpretable in terms of input, training process, and decision-making simultaneously?

**Challenge 2:** How can we optimize the performance of LLMs without fine-tuning the model parameters to reduce costs?

To address these challenges, we propose the **V**erbalized **G**raph **R**epresentation **L**earning (VGRL) method. **For Challenge 1**, VGRL utilizes a verbalized approach to create intuitive connections

between input features and predictions and VGRL generates textual explanations at each iteration stage, helping researchers and practitioners better grasp the training dynamics of the model. Also, VGRL provides natural language descriptions for the model's predictions, clearly explaining the rationale behind each decision. **For Challenge 2**, instead of relying on costly fine-tuning of the LLM parameters, VGRL leverages a prompt-based optimization strategy. This involves crafting task-specific prompts to guide the LLM in generating optimal predictions without modifying its internal parameters. By utilizing prompt engineering techniques, VGRL maintains high performance while significantly reducing computational costs associated with traditional fine-tuning methods. Additionally, this approach allows the model to remain versatile across various tasks, as it can be adapted to new datasets or problems simply by adjusting the prompts, further enhancing its efficiency and scalability.

Our contributions are as follows:

- We propose a novel verbalized graph learning framework that ensures complete interpretability throughout the entire process, from input to training and decision-making, enabling users to fully understand the operational mechanisms of the model.
- We seek to reduce the high GPU overhead associated with pre-training or fine-tuning in current graph plus LLMs paradigms by utilizing a new model optimization approach, known as Iterative Training through Prompt Optimization.
- We validate the effectiveness of this method from multiple perspectives on real-world datasets.

## 2 PRELIMINARIES

In this section, we introduce the essential concepts, notations, and problem settings considered in this research. Our primary focus is on the node classification task over text-attributed graphs, which represents a fundamental downstream task within the field of graph learning. We begin by defining text-attributed graphs.

**Text-Attributed Graphs**. Text-attributed graphs (TAGs) can be formally described as $\mathcal{G} = (\mathcal{V}, \mathcal{A}, \{\mathcal{X}_n\}_{n \in \mathcal{V}})$, where $\mathcal{V}$ represents a set of $\mathcal{N}$ nodes, $\mathcal{A} \in \mathbb{R}^{\mathcal{N} \times \mathcal{N}}$ is the adjacency matrix, and $\mathcal{X}_n \in \mathcal{D}^{\mathcal{L}_n}$ denotes a sequential text associated with each node $v_n \in \mathcal{V}$. Here, $\mathcal{D}$ is the dictionary of words or tokens, and $\mathcal{L}_n$ is the length of the sequence. In this paper, we focus on the problem of node classification in TAGs. Specifically, given a subset of labeled nodes $\mathcal{L} \subseteq \mathcal{V}$, the task is to predict the labels of the remaining unlabeled nodes $\mathcal{U} = \mathcal{V} \setminus \mathcal{L}$. And iterates over the input mini-batch $\mathcal{B}$ one-pass input.

**One-hop neighbors**. Given a node $v_i \in \mathcal{V}$, the set of one-hop neighbors, denoted as $\mathcal{N}(v)$, where $\mathcal{N}(v_i) = \{v_j \in \mathcal{V} | (v_i, v_j) \in \mathcal{E}\}$

$k$-**hop neighbors**. Given a node $v_i$, for $k \geq 2$, the $k$-hop neighbors of $v_i$ can be denoted as $\mathcal{N}^k(v_i)$, where $\mathcal{N}^k(v_i) = \{v_j \in \mathcal{V} \mid \exists v_m \in \mathcal{N}^{k-1}(v_i), (v_m, v_j) \in \mathcal{E} \wedge v_j \notin \mathcal{N}^{k-1}(v_i)\}$.

## 3 RELATED WORK

In this section, we review the existing literature related to integrating Large Language Models (LLMs) and Graph Neural Networks (GNNs). Prior work has focused on several key areas, including traditional methods for trusted GNNs, the role of LLMs in graph-based tasks, and recent advances in optimization frameworks utilizing LLMs. We explore these approaches to highlight their contributions and limitations, establishing the foundation for our proposed Verbalized Graph Representation Learning (VGRL) framework.

### 3.1 GRAPH AND LLMs

**Traditional approaches to trusted GNNs.** There are currently two main approaches: post-hoc explanation methods and self-interpretable models . The former tries to interpret the output of the model by adding a model-independent interpreter, for example (Ying et al., 2019; Vu & Thai, 2020;

Zhang et al., 2023b). However, this can lead to incomplete explanatory information in the output, or even generate explanatory information that is incorrect in the opinion of humans. The latter tries to solve this problem by constructing models that themselves have interpretable principles, for example (Dai & Wang, 2021; Zhang et al., 2022a). However, these interpretable principles are based on their inductive bias, and only experts in the relevant fields can accurately judge whether such inductive bias is reasonable or not.

**LLM in Graph.** Existing methods are mainly categorized into three types: (1) LLM as Enhancer which mainly enhances the performance of GNNs by adding LLM-generated information, for example (He et al., 2023; Chen et al., 2024; Ni et al., 2024); (2) LLM as Predictor which mainly performs a downstream task by directly inputting the graph structure into the LLM, for example (Tang et al., 2024; Qin et al., 2023); (3) LLM as Alignment which mainly enhances the performance by aligning embedding spaces of GNNs and LLMs, for example (Yang et al., 2021; Mavromatis et al., 2023). Among them, there is explanation-based LLM-as-Enhancer approach (He et al., 2023), which achieves better performance by letting LLM generate natural language explanation information of graph structures and then embedding it into GNNs for downstream tasks. However, after the embedding from natural language to graph structure is not directly visible as a black box to humans, and can only be proven effective indirectly through the performance of downstream tasks.

## 3.2 LLMs Optimization

**LLMs for planning and optimization**. Large language models (LLMs) have been successfully applied to planning tasks for embodied agents (Song et al., 2023; Xie et al., 2023; Li et al., 2022; Liang et al., 2023), enabling them to follow natural language instructions and complete complex tasks. More recently, LLMs have also been utilized to tackle optimization problems by generating new solutions from prompts that incorporate previously generated solutions and their associated loss values. While these LLM-based optimization (Xiao et al., 2024; Yang et al., 2024) methods bear some resemblance to our approach, as we also use LLMs to address optimization challenges, a key limitation of existing work is that it has not yet been explored in the graph domain. To address this gap, we propose an extension of this framework to the graph domain, introducing Verbalized Graph Representation Learning (VGRL), which applies LLMs to graph neural networks (GNNs) and opens new possibilities for solving graph-based optimization problems through natural language interactions.

**Prompt engineering and optimization**. Numerous prompting techniques (Wei et al., 2022; Zhang et al., 2022b; Zhou et al., 2022; Wang et al., 2022; Yao et al., 2024; 2023; Weston & Sukhbaatar, 2023) have been developed to enhance the reasoning capabilities of LLMs. To minimize the manual effort required in designing effective prompts, various automatic prompt optimization approaches (Zhang et al., 2022b; Zhou et al., 2022; Yang et al., 2024; Pryzant et al., 2023; Wen et al., 2024; Deng et al., 2022; Li et al., 2024b; Ma et al., 2024; Sordoni et al., 2023) have been introduced. However, traditional prompt optimization methods primarily focus on refining the text prompt without changing its underlying semantic meaning. In contrast, our VGRL framework goes beyond mere prompt adjustments by directly updating the parameters of the language-based model through the integration or modification of prior information. This not only improves optimization but also ensures that the learner model remains fully interpretable in its predictions, offering a more robust and transparent solution for graph-based learning tasks.

**LLMs for multi-agent systems**. Given their strong instruction-following capabilities, LLMs can assume various roles within multi-agent systems (Qian et al., 2023; Wu et al., 2023; Hong et al., 2023; Li et al., 2023). For instance, explore multi-agent collaboration systems designed to solve complex tasks such as software development. In the VGRL framework, this concept is extended to a two-agent system, where one LLM functions as the learner and the other as the optimizer.

Our approach sidesteps the problem of modeling black boxes by having the LLM generate human-readable information as promt of another LLM making it perform the downstream task. This can be viewed as a "guidance-feedback-redirection" process between models, which, after many iterations, returns the optimal guidance solution for a given task, which is directly human-readable.

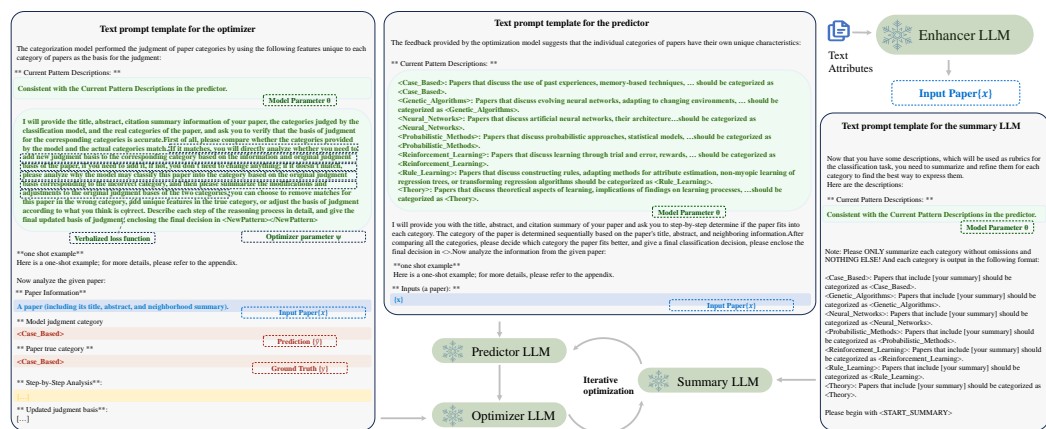

Figure 2: An overview of iterative optimization and text prompt templates for the predictor, optimizer, and summary LLM in the node classification example

# 4 PROPOSED METHOD

In this paper, we present the Verbalized Graph Representation Learning (VGRL) framework, a pioneering approach that integrates large language models (LLMs) with graph-based tasks while ensuring full interpretability throughout the process. Our methodology encompasses four innovative components, each designed to enhance both the performance and the transparency of LLMs in handling graph data.

## 4.1 INCORPORATING GRAPH STRUCTURE INTO LLM PREDICTIONS

Although Large Language Models (LLMs) can achieve competitive zero-shot performance on specific datasets without considering graph structures, their performance often lags behind Graph Neural Networks (GNNs) on benchmark datasets such as CORA, CITESEER, and OGBN-ARXIV. This gap underscores the importance of graph structural information in tasks like node classification, indicating the need to explore how incorporating graph structures into prompts could enhance LLM performance.

Given that LLMs (e.g., ChatGPT) are not natively designed to process adjacency matrices or perform graph-based computations, it is impractical to directly integrate graph operations into LLMs. Thus, an alternative approach is to verbalize graph information as text that LLMs can process effectively. This transformation allows LLMs to interpret node relationships and structural dependencies in natural language format. In (Chen et al., 2024), various methods are evaluated to represent node connections textually, aiming to enhance LLM reasoning capabilities for graph-based tasks.

One effective method is the 'ego-graph' approach, which focuses on the local subgraph surrounding a target node. By constraining the LLM's focus to a limited number of nodes, this method reduces complexity while preserving key local graph structure. To simulate the neighborhood aggregation process typical in GNNs, the input prompt incorporates a summary of attributes from neighboring nodes. Thus, important information from the graph is conveyed to the LLM without altering its reasoning mechanisms. This process can be formalized as:

$$Z_{v_i}^1 = f_e \left( \mathcal{X}_{v_i}, \left\{ \mathcal{X}_{v_j} \mid v_j \in \mathcal{N}(v_i) \right\} \right) \tag{1}$$

where $Z_{v_i}^1$ is the enhanced representation of node $v_i$ with one-hop neighbor information, $\mathcal{X}_{v_i}$ represents the features of node $v_i$, and $\mathcal{N}(v_i)$ denotes the set of one-hop neighbors of $v_i$. The function $f_e$ encapsulates the process of verbalizing neighborhood information and processing it by the LLM.

Inspired by this ego-graph approach, we have also introduced a method for incorporating structural information into our model. By embedding the attributes and relationships of neighboring nodes into the prompt, we aim to enable the LLM to better capture the interactions between nodes. Below is an example of a neighbor summary in Table 1:

Table 1: Prompts used to generate neighbor summary.

**Prompts used to summarize the neighboring information**

I will now give you basic information about all the papers cited in a paper; this information includes: the abstracts and categories of the cited papers. The following list records some papers related to the current one.

[{ "content": "This paper firstly provides ...", "category": "Rule Learning"... }, ...]

**# Instruction**

Please summarize the information above with a short paragraph, find some common points which can reflect the category of this paper.

Note: ONLY your summary information and NOTHING ELSE!

Please start with "The papers cited in this essay".

## 4.2 VERBALIZING MODEL PARAMETERS FOR INTERPRETABILITY

Traditional machine learning models, such as neural networks, rely on numerical parameters, $\theta = \{\theta_1, \theta_2, \ldots, \theta_t\}$, which are often difficult to interpret. These parameters are typically represented as abstract numerical values, making it complex and non-intuitive to understand or explain the internal workings of the model. In contrast, the Verbalized Graph Representation Learning (VGRL) framework leverages large language models (LLMs) to express model parameters through natural language, providing full interpretability.

In VGRL, the model parameters $\theta_t$ are defined by a text prompt, which consists of human-readable natural language tokens, $\theta_t \in \Theta_{\text{language}}$, where $\Theta_{\text{language}}$ is the set of all interpretable text sequences. This approach contrasts with traditional models where parameters are abstract numbers, which are hard to interpret directly. The VGRL framework unifies both data and model parameters into a natural language-based format that is inherently understandable.

The key features of this framework include:

- **Discrete Parameters**: The natural language used to express parameters $\theta$ is inherently discrete. This is in contrast to the continuous parameter representations in traditional models, enhancing the intuitiveness of parameter interpretation.

- **Sequential Structure**: The parameters exhibit a sequential structure, as $\theta = \{\theta_1, \theta_2, \ldots, \theta_t\}$, reflecting the temporal or contextual relationships between parameters. This sequential nature aids in capturing and understanding the dynamics between parameters.

- **Human Interpretability**: Since the parameters $\theta_t$ are verbalized in natural language, they are inherently comprehensible to humans. This allows the model's reasoning process and learning mechanisms to be more transparent, facilitating interpretability and easier analysis.

An advantage of using natural language for model parameters is that it enables the integration of prior knowledge and inductive biases directly into the model. As the model updates its parameters $\theta_t$, the changes are fully interpretable, providing clear insights into what the model is learning. For example, changes in $\theta_t$ can be directly mapped to natural language descriptions, offering an intuitive understanding of the model's learning process.

Our empirical evidence demonstrates that text-based parameters often correspond to recognizable patterns in the data, further reinforcing the interpretability and transparency of the VGRL approach. This natural language parameterization not only enhances the intuitiveness of model but also improves its application, offering clearer insights into model tuning and interpretation in real-world scenarios.

## 4.3 LEVERAGING LLMs FOR NODE CLASSIFICATION

Our approach centers on utilizing LLMs as interpretable predictors by querying them in an 'open-ended' manner. Unlike existing methods that primarily rely on message passing mechanisms, our

method employs a label feature matching mechanism. We match based on the inherent characteristics of the nodes themselves and the information from their neighbors. This label feature matching mechanism places a stronger emphasis on the intrinsic attributes of node, as it aligns with the insights provided in the prompt.

The core of this method is represented by the following equation:

$$\hat{y}_{v_i} = f_p(Z_{v_i}^k, \theta_{t-1}) \tag{2}$$

Here, $\hat{y}_{v_i}$ denotes the predicted label for node $v_i$, and $Z_{v_i}^k$ represents the enhanced node representation incorporating $v_i$'s $k$-hop neighbors. $\theta_{t-1}$ refers to the LLM's parameters at the previous step, enabling the model to leverage its prior knowledge and reasoning capabilities to generate the prediction. The function $f_p$ serves as the predictor that utilizes the enhanced representation and model parameters to produce the label output. This formulation emphasizes the LLM's role as a predictor, focusing on generating interpretable outputs.

For each node $v_i \in \mathcal{V}$, a prompt is crafted that includes not only the node's features, such as the paper title and abstract, but also relevant graph structure information. Specifically, the attributes of neighboring nodes up to the $k$-hop neighborhood are embedded in the prompt, as encapsulated in $Z_{v_i}^k$. This enables the LLM to better understand the node's context and surroundings within the graph, leading to more informed and accurate predictions.

### 4.4 LLM AS AN OPTIMIZER WITH INTERPRETABLE OPTIMIZATION PROCESS

For the predictor LLM, we provide textual descriptions of node categories, which serve as model parameter $\theta$, and the model determines which category the input node $v_i$ belongs to based on the given descriptions. The quality of node category descriptions $\theta$ directly affects the performance of LLM predictions; hence, obtaining suitable node category descriptions is very important. Additionally, for better explainability, VGRL imposes a strong constraint on $\theta$, ensuring that the updated $\theta$ still belong to natural language sequences that humans can understand.

Under these conditions, it is not advisable to use classical machine learning optimization methods such as gradient descent to optimize $\theta$. Inspired by Xiao et al. (2024), the optimizer LLM can output natural language that satisfies the constraints, so we only need to ask the LLM to play the role of an optimizer, then optimized category descriptions are also in natural language understandable by humans. Therefore, we directly use another LLM to optimize $\theta$. Given a mini-batch $\mathcal{B}$, the optimization process is as follows:

$$\widetilde{\theta}_{v_i}^t = g_{opt}(Z_{v_i}^k, y_{v_i}, \hat{y}_{v_i}, \theta_{t-1}, \Psi), v_i \in \mathcal{B} \tag{3}$$

where $y_{v_i}$ is the true label of $v_i$, $\widetilde{\theta}_{v_i}^t$ represents the intermediate parameter values for node $v_i$ during the $t$-th iteration, and $\Psi$ denotes the parameter of the optimizer LLM, which is a text prompt. Specifically, we optimize the intermediate parameter value $\widetilde{\theta}_{v_i}^t$ of each node $v_i$ in $\mathcal{B}$, and then summarize the intermediate parameter values of these nodes through a summary LLM (Section 4.5) to obtain a new round of parameter $\theta_t$. The overall framework for optimizer optimization and the text prompt template are given in Figure 2. The parameter $\Psi$ of the optimizer LLM is actually a text prompt provided by humans and is not updated. The text prompt linguistically specifies the optimization loss function, guiding the optimizer LLM to optimize $\theta$. The LLM-parameterized optimizer allows users to interact with it directly, which not only helps to trace model failures but also permits the incorporation of prior knowledge to enhance optimization. In addition, we also guide the LLM to output explanations of the optimization process, demonstrating the explainability of the VGRL optimization process.

### 4.5 SUMMARY LLM

The role of the Summary LLM is to aggregate and summarize the updated intermediate parameters from the optimizer LLM, generated during the previous minibatch, to obtain updated $\theta$. Specifically, given a set of updated parameters from the last minibatch $\mathcal{B}$, the Summary LLM consolidates these updates into a new set of parameters, $\theta_t$. This process can be formalized as:

$$\theta_t = f_s \left( \{ \widetilde{\theta}_{v_i}^t \mid v_i \in \mathcal{B} \} \right) \tag{4}$$

Here, $\widetilde{\theta}_{v_i}^t$ represents the intermediate parameter values for node $v_i$ during the $t$-th iteration, and $\mathcal{B}$ denotes the set of nodes in the current minibatch. The function $f_s$ operates by combining these parameter updates to produce a cohesive set of parameters, $\theta_t$, which reflects the overall learning progress across the minibatch. This aggregation ensures that key information from each node's updated parameters is captured while maintaining coherence in the overall optimization process.

### 4.6 CHAIN-OF-THOUGHT PROMPTING

Inspired by (Wei et al., 2022), we introduce the zero-shot and one-shot Chain-of-Thought (CoT) methods in prompt. For the zero-shot method, we encourage the LLM to perform step-by-step text generation by restricting and guiding the LLM to make the generated explanatory information as structured and precise as possible, in order to achieve a better final result generation based on the self-generated information. Although zero-shot VGRL is already fully interpretable, we still want to customize the interpretation in specific domains to ensure that the interpretation information is more in line with the norms of the human mind and thus enhance the model's performance. Therefore, we introduce the one-shot method by manually constructing a sample of the CoT, so that the model can generate the interpretation information and the final output based on the sample. The motivation for the one-shot approach is that we believe that the content generated by the LLM based on a sample that conforms to the logic of the human mind will better contribute to the completion of the final task.

## 5 EXPERIMENTS

In this section, We will compare the performance of the VGRL framework with diverse backbone models for the TAG node classification task. We will answer the following questions:

- **Q1:** Can VGRL framework increase the performance of backbone models?
- **Q2:** Do each part of the VGRL framework play a positive role?

### 5.1 BASELINE AND EXPERIMENT SETTING

We use two LLM-as-predictor models as backbones (Chen et al., 2024), and add our framework on top of them for comparisons. Information on our equipment can be found at Table 2.

- **Node only:** 'node only' refers to the features considering only the node itself, excluding any neighbor information.
- **Summary**: 'Summary' indicates that we used an independent LLM to summarize the node's $k$-hop information, which can be viewed as the introduction of an enhancer LLM for encoding the graph structure. The prompt for the enhancer LLM is shown in Table 1.

During the experiments, we used one-hop neighbor information for summarization and set model temperature $\tau = 0.1$ as default. Additionally, we introduced prior knowledge in our comparison by manually constructing prior knowledge as the initial optimize $\theta$ for iterative processing. And we setting a mini-batch training process with a batch size of 8, i.e. $|\mathcal{B}| = 8$.

Table 2: Information on our equipment

| Devices | |
|---|---|
| OS | Ubuntu 22.04.4 LTS x86_64 |
| Language | Python 3.10.14 |
| Frameworks | pytorch 2.4.0 + cuda 12.4 |
| CPU | Intel Xeon Silver 4310 (48) @ 3.300GHz |
| GPU | 3 * NVIDIA L20 (48G) |
| Memory | 128508MiB |

### 5.2 MAIN RESULTS (Q1)

We conducted evaluations on the Cora TAG (McCallum et al., 2000) dataset (See AppendixB) by comparing our optimization iterative process with the baseline that excludes the VGRL framework

(Chen et al., 2024). The results are presented in Table 3. We extracted a subset of nodes from the Cora dataset as our experimental data. For further steps, we blurred the concept of epochs and treated each batch as a single step.

Table 3: Node classification accuracy for the Cora dataset

| Cora | w/ prior | | w/o prior | |
|---|---|---|---|---|
| | zero-shot | one-shot | zero-shot | one-shot |
| Node only | 0.625 | 0.400 | **0.675** | 0.100 |
| Node only + VGRL | **0.650** | **0.625** | **0.675** | **0.475** |
| Summary | 0.650 | 0.550 | 0.700 | 0.475 |
| Summary + VGRL | **0.800** | **0.700** | **0.875** | **0.700** |

Table 4: Ablation study on the Cora dataset, showing the effects of different variants base on Summary + VGRL on the accuracy performance

| Cora Summary + VGRL | w/ prior | | w/o prior | |
|---|---|---|---|---|
| | zero-shot | one-shot | zero-shot | one-shot |
| original method | **0.800** | **0.700** | **0.875** | **0.700** |
| w/o optimizer LLM | 0.650 | 0.550 | 0.700 | 0.475 |
| w/o summary LLM | 0.650 | 0.625 | 0.725 | 0.625 |

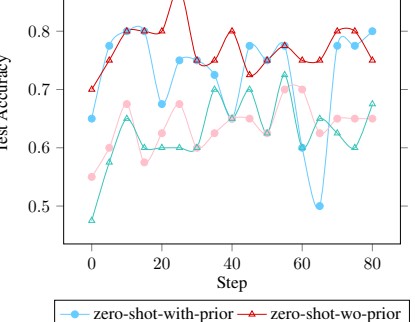

Figure 3: Summary+VGRL Acc-Step

Our comparison reveals that our framework, through the iterative process, achieves better performance, demonstrating the effectiveness of the VGRL framework in representation learning. VGRL gradually refines the label features through repeated iterations, as shown in Figure 3, which illustrates the change in test accuracy during the mini-batch iterations. Additionally, we used the open-source Llama3.1 8B model for all experiments, which not only significantly reduced costs but also proved the optimization capability of the framework itself.

## 5.3 ABLATION EXPERIMENTS (Q2)

We conducted ablation experiments on the Summary + VGRL architecture to assess the importance and relevance of each module. The results of the ablation experiments are shown in Table 4.

- **w/o optimizer LLM:** This variant removes the optimizer LLM, i.e., there is no iterative optimization process, which is equivalent to using the predictor LLM to make the final decision.
- **w/o summary LLM:** This variant removes the summary LLM, i.e., after each optimization update, instead of summarizing the information through the summary LLM, the results of a batch update are directly used in the next iteration.

## 5.4 CASE STUDY

To explore the impact of the VGRL framework on the TAG node classification task, we conducted an analysis of a particular training sample from the Cora dataset, as shown in Figure 4. In the paper 'Evolving Sensors in Environments of Controlled Complexity' the one-hop neighboring nodes all have the label 'Genetic_Algorithms' while the actual label of the node is 'Reinforcement_Learning' This heterogeneity can significantly disrupt the node's feature information during neighborhood aggregation, resulting in biased classification results. However, VGRL is able to effectively capture unique characteristics of each category, using them as a basis for matching the node's own features. This addresses the issue of information corruption caused by the propagation mechanism in heterogeneous graphs.

Moreover, in the Cora dataset, paper categories cannot be strictly divided into binary classes. It is not uncommon for some nodes to belong to two categories simultaneously. In such cases, the label-feature matching mechanism proves to be more reasonable than the message-passing mechanism, as it focuses more on the node's own information (as can be inferred from the formulation of $\theta$). Making judgments and decisions based on one's existing knowledge ($\theta$) is the most fundamental decision-making process for humans.

'Judgment' and 'Step-by-Step Analysis' represent the model's label matching process, which is also human-readable and interpretable. Whether its the Predictor LLM's process of analyzing the

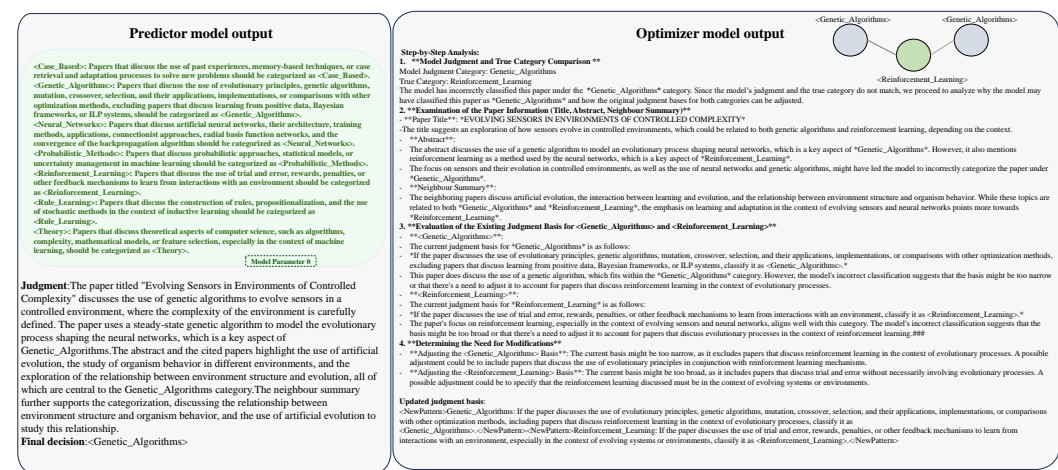

Figure 4: Case study for one-shot wo prior Summary + VGRL: (1) The left figure shows the explanation information and prediction labels output by predictor LLM; (2) The right figure shows the optimization process of optimizer LLM for the predicted content of predictor LLM in the left figure.(3) The top-right figure shows an example of the one-hop neighbors of a predicted sample.

node's own features and supplementing it with neighborhood information, or the Optimizer LLM's analysis and adjustment of the two categories involved in classification errors, both demonstrate a complete and interpretable optimization process. The model explains each update iteration in detail, presenting it in human-readable language. With the help of the Summary LLM, the Predictor LLM and Optimizer LLM communicate and feedback effectively, ultimately constructing the best decision-making basis from scratch for the node classification task on the current dataset.

For a detailed training process see Appendix C to Appendix G.

## 6 THEORETICAL ANALYSIS

In this section, our goal is to demonstrate that the category descriptions generated by LLM can provide useful information for predicting label categories. Specifically, if the obtained category descriptions can faithfully represent the information of each category, then they are useful. At the same time, the LLM is non-redundant, as it can provide information that $X$ cannot provide. Let $\theta$ be the textual category descriptions generated by LLM; $H_l$ are the embeddings of category from the LLM; $X$ are the input of graph structure embeddings, $y$ is the target and $H(\cdot|\cdot)$ is the conditional entropy. The specific proof process can be found in Appendix A.

**Theorem.** *Given the following conditions: 1) Fidelity: $\theta$ can faithfully represent the information of $H_l$ such that $H(H_l|\theta) = \epsilon$, with $\epsilon > 0$; 2)Non-redundancy: $H_l$ contains information not present in $X$, that is, $H(y|X, H_l) = H(y|X) - \epsilon'$, with $\epsilon' > \epsilon$. Then it follows that $H(y|X,\theta) < H(y|X)$.*

## 7 CONCLUSION

This paper introduces Verbalized Graph Representation Learning (VGRL), a novel approach to text-attributed graph learning that ensures full interpretability by representing learned parameters as textual descriptions instead of continuous vectors. This method enhances transparency and user understanding of the decision-making process, fostering greater trust in the model's outputs. While the current application is limited to foundational graph learning paradigms, VGRL shows promise for broader use in more complex models, offering potential advancements in explainable AI and graph-based learning systems.

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

APPENDIX

## A  THEORETICAL ANALYSIS

In this section, our goal is to demonstrate that the category descriptions generated by LLM can provide useful information for predicting label categories. We formulate our theorem as follows:

**Theorem.** *Given the following conditions:*
*1) Fidelity: $\theta$ can faithfully represent the information of $H_l$ such that*

$$H(H_l|\theta) = \epsilon, \epsilon > 0; \tag{5}$$

*2) Non-redundancy: $H_l$ contains information not present in $X$, that is*

$$H(y|X, H_l) = H(y|X) - \epsilon', \epsilon' > \epsilon; \tag{6}$$

*Then we can obtain:*

$$H(y|X, \theta) < H(y|X). \tag{7}$$

where $\theta$ be the textual category descriptions generated by $LLM$; $H_l$ are the embeddings of category from the $LLM$; $X$ are the input of graph structure embeddings, $y$ is the target and $H(\cdot|\cdot)$ is the conditional entropy.

*Proof.* We aim to demonstrate that $H(y|X, \theta) < H(y|X)$, the process is following:
Start with:

$$H(y|X, \theta) \tag{8}$$

We decompose the original expression Equation 8 into two parts based on the properties of entropy:

$$H(y|X, \theta) = H(y|X, H_l, \theta) + I(y; H_l|X, \theta) \tag{9}$$

Based on the definition of mutual information, we can obtain:

$$I(y; H_l|X, \theta) = H(H_l|X, \theta) - H(H_l|X, \theta, y) \tag{10}$$

Due to the non-negativity of conditional entropy, we have:

$$I(y; H_l|X, \theta) \leq H(H_l|X, \theta) \tag{11}$$

By substituting Equation 11 into Equation 9, we further obtain:

$$H(y|X, \theta) \leq H(y|X, H_l, \theta) + H(H_l|X, \theta) \tag{12}$$

When conditional variables decrease, the conditional entropy increases; so we have:

$$H(y|X, \theta) \leq H(y|X, H_l) + H(H_l|\theta) \tag{13}$$

Applying the two aforementioned conditions and substituting Equations 5 and 6 into Equation 12, we can obtain:

$$H(y|X, \theta) \leq H(y|X) + \epsilon - \epsilon' < H(y|X) \tag{14}$$

The conclusion is thus proven. $\square$

## B  DATASET DESCRIPTION

Cora (McCallum et al., 2000): The Cora dataset consists of Machine Learning papers. These papers are classified into one of the following seven classes: Case_Based, Genetic_Algorithms, Neural_Networks, Probabilistic_Methods, Reinforcement_Learning, Rule_Learning, Theory. The papers were selected in a way such that in the final corpus every paper cites or is cited by atleast one other paper. There are 2708 papers and 5429 links in the whole corpus.

## C ONE-SHOT CoT

The one-shot example.

---

**one-shot predictor LLM CoT**

**Paper information:**

(Paper Title: Case-Based Planning to Learn )
(Abstract: Learning can be viewed as a problem of planning a series of modifications to memory. We adopt this view of learning and propose the applicability of the case-based planning methodology to the task of planning to learn. We argue that relatively simple, fine-grained primitive inferential operators are needed to support flexible planning. We show that it is possible to obtain the benefits of case-based reasoning within a planning to learn framework.)
(Neighbour Summary: The papers cited in this essay discuss the challenges and limitations of artificial intelligence, specifically in the areas of case-based reasoning and inference control. The first paper presents a new approach to acquiring case adaptation knowledge, where adaptation problems are initially solved by reasoning from scratch and then stored as cases for future adaptation. The second paper addresses the problem of combinatorial explosion of inferences and proposes a method to generate explicit desires for knowledge to control inference and facilitate goal pursuit. Both papers highlight the importance of knowledge acquisition and inference control in artificial intelligence.
Common points that reflect the category of this paper include the focus on case-based reasoning, the need for knowledge acquisition and inference control, and the emphasis on artificial intelligence.)

**Judgment:**

The paper titled "Case-Based Planning to Learn" presents the idea that learning can be viewed as a problem of planning modifications to memory, and it applies case-based reasoning to support flexible planning in learning. It emphasizes the reuse of past cases to inform future planning, which aligns with case-based methodologies.
The core of the paper discusses case-based reasoning, which involves using past experiences (in the form of cases) to guide future learning and problem-solving. The approach focuses on planning based on previously solved cases and adapting them to new scenarios, which is a hallmark of Case-Based methodologies.
The abstract and the cited papers highlight the importance of case retrieval and adaptation processes for handling learning and reasoning, both central concepts to Case-Based reasoning.
The neighbour summary further supports the categorization, discussing the role of case adaptation knowledge and its use in addressing new problems, aligning with the Case-Based category.
Final decision:
<Case_Based>

---

**one-shot Optimizer LLM CoT**

**Paper Information:**

(Paper Title: Case-Based Planning to Learn )
(Abstract: Learning can be viewed as a problem of planning a series of modifications to memory. We adopt this view of learning and propose the applicability of the case-based planning methodology to the task of planning to learn. We argue that relatively simple, fine-grained primitive inferential operators are needed to support flexible planning. We show that it is possible to obtain the benefits of case-based reasoning within a planning to learn framework.)
(Neighbour Summary: The papers cited in this essay discuss the challenges and limitations of artificial intelligence, specifically in the areas of case-based reasoning and inference control. The first paper presents a new approach to acquiring case adaptation knowledge, where adaptation problems are initially solved by reasoning from scratch and then stored as cases for future adaptation. The second paper addresses the problem of combinatorial explosion of inferences and proposes a method to generate explicit desires for knowledge to control inference and facilitate goal pursuit. Both papers highlight the importance of knowledge acquisition and inference control in artificial intelligence.
Common points that reflect the category of this paper include the focus on case-based reasoning, the need for knowledge acquisition and inference control, and the emphasis on artificial intelligence.)

**Model judgment category:**

<Case_Based>

**Paper true category:**

<Case_Based>

**Step-by-Step Analysis:**

1. **Model Judgment and True Category Comparison**
- **Model Judgment Category**: *<Case_Based>*
- **True Category**: *<Case_Based>*
The model has correctly classified this paper under the *<Case_Based>* category. Since both the model's judgment and the true category match, we move on to analyzing whether the original judgment basis for *<Case_Based>* is accurate or needs to be updated based on this paper's information.
2. **Examination of the Paper Information (Title, Abstract, Neighbour Summary)**
- **Paper Title**: *Case-Based Planning to Learn*
- The title directly suggests the focus on *Case-Based* methods. Specifically, it highlights the use of *planning* in conjunction with *case-based reasoning* to facilitate learning.
- **Abstract**:
- The abstract provides clear evidence that the paper is centered around *case-based reasoning*, as it proposes the use of a *case-based planning methodology* to plan learning processes. It also discusses storing past solutions for future use, which is central to *case-based reasoning*.
- The mention of *modifications to memory* and *benefits of case-based reasoning* reinforces the paper's focus on adapting previous experiences to solve new problems, which is a key aspect of *<Case_Based>* reasoning.
- **Neighbour Summary**:
- The neighbouring papers also focus on *case-based reasoning*, *knowledge acquisition*, and *inference control*. These are essential to *<Case_Based>* reasoning, as they emphasize solving new problems by leveraging past experiences and stored cases.
3. **Evaluation of the Existing Judgment Basis for <Case_Based>**
The current judgment basis for *<Case_Based>* is as follows:
- *If the paper discusses the use of past experiences, memory-based techniques, or case retrieval and adaptation processes to solve new problems, classify it as <Case_Based>.*
The paper fits this description well:
- **Use of past experiences**: The paper discusses retrieving and using past cases (i.e., stored solutions) to plan learning processes.
- **Memory-based techniques**: The reference to *modifications to memory* and storing past adaptations falls under memory-based techniques.
- **Case retrieval and adaptation**: The paper directly mentions using case-based reasoning within a planning framework, which includes adapting stored cases to new learning challenges.
4. **Determining the Need for Modifications**
The current judgment basis for *<Case_Based>* is well-aligned with the content of this paper. The focus on using past experiences, memory modifications, and case adaptation processes are clearly articulated in the abstract and neighbor summary.
- **No new features need to be added** to the *<Case_Based>* category based on this paper, as it fits perfectly within the existing framework.

**Updated judgment basis:**

<NewPattern>Case_Based: If the paper discusses the use of past experiences, memory-based techniques, or case retrieval and adaptation processes to solve new problems, classify it as <Case_Based>.</NewPattern>

---

# D  TRAINING DETAIL FOR SUMMARY + VGRL

## D.1  ONE-SHOT W/O PRIOR SUMMARY + VGRL STEP 1

---

**one-shot predictor LLM CoT wo prior Step 1**

The feedback provided by the optimization model suggests that the individual categories of papers have their own unique characteristics:
Judge from the information given which category the following essay belongs to.
I will provide you with the title, abstract, and citation summary of your paper and ask you to step-by-step determine if the paper fits into each category. First, please analyze whether the paper matches category <Case_Based>. If it does, please explain how it matches; if it doesn't, please point out where it doesn't. Then please compare the categories <Genetic_Algorithms> and vice versa. After comparing all the categories, please decide which category the paper fits better, and give a final classification decision, please enclose the final decision in <>.
Now analyze the information from the given paper:
====================================================
Here is a one-shot example; for more details, please refer to the appendix.
====================================================
Paper information:
(Paper Title: Stochastic Propositionalization of Non-Determinate Background Knowledge )
(Abstract: It is a well-known fact that propositional learning algorithms require "good" features to perform well in practice. So a major step in data engineering for inductive learning is the construction of good features by domain experts. These features often represent properties of structured objects, where a property typically is the occurrence of a certain substructure having certain properties. To partly automate the process of "feature engineering", we devised an algorithm that searches for features which are defined by such substructures. The algorithm stochastically conducts a top-down search for first-order clauses, where each clause represents a binary feature. It differs from existing algorithms in that its search is not class-blind, and that it is capable of considering clauses ("context") of almost arbitrary length (size). Preliminary experiments are favorable, and support the view that this approach is promising.)
(Neighbour Summary:
The papers cited in this essay discuss various techniques and approaches in the field of Rule Learning, specifically in Inductive Logic Programming (ILP). The first paper re-appraises the development of techniques for inverting deduction and introduces Mode-Directed Inverse Entailment (MDIE) as a generalisation and enhancement of previous approaches. The second paper proposes an extension of the feature-vector representation to allow the value of a feature to be a set of strings, which can be efficiently and naturally represented in real-world learning problems. The third paper describes a system named SPOIL that uses a stochastic search method to alleviate the local optimization problem in greedy algorithms. Overall, these papers highlight the importance of efficient and effective techniques in Rule Learning and ILP.)

Judgment:

The paper titled "Stochastic Propositionalization of Non-Determinate Background Knowledge" discusses the construction of good features for inductive learning by devising an algorithm that searches for features defined by substructure. The algorithm conducts a top-down search for first-order clauses, which represents a binary feature. This approach is different from existing algorithms in that its search is not class-blind and can consider clauses of almost arbitrary length.
The abstract and the cited papers highlight the importance of feature engineering and the use of stochastic search methods to alleviate local optimization problems, both of which are central concepts to Rule Learning and ILP.
The neighbour summary further supports the categorization, discussing the development of techniques for inductive logic programming and the importance of efficient and effective methods in rule learning.

Final decision:
<Rule_Learning>

---

**one-shot summary LLM CoT wo prior Step 1**

Now that you have some descriptions, which will be used as rubrics for the classification task, you need to summarize and refine them for each category to find the best way to express them.
Here are the descriptions:

Judge from the information given which category the following essay belongs to:
<Case_Based>: If the paper discusses the use of past experiences, memory-based techniques, or case retrieval and adaptation to solve new problems, classify it as <Case_Based>. If the paper discusses the use of past experiences, memory-based techniques, or case retrieval and adaptation processes to solve new problems, classify it as <Case_Based>. Additionally, if the paper emphasizes the importance of explanation in goal-driven learning, including the use of introspection and adaptation of explanation strategies, classify it as <Case_Based>.
<Genetic_Algorithms>: If the paper discusses the use of evolutionary principles, such as mutation, crossover, and selection, to search for optimal solutions, classify it as <Genetic_Algorithms>. If the paper discusses the use of genetic algorithms, including their applications, implementations, or comparisons with other optimization methods, classify it as <Genetic_Algorithms>. If the paper discusses the use of evolutionary principles, such as mutation, crossover, or selection, to search for optimal solutions, classify it as <Genetic_Algorithms>.
<Neural_Networks>: If the paper discusses the use of artificial neural networks, including their architecture, training methods, and applications, classify it as <Neural_Networks>. Additionally, if the paper covers topics such as connectionist approaches, radial basis function networks, and the convergence of the backpropagation algorithm, classify it as <Neural_Networks>. If the paper discusses the use of artificial neural networks for learning, classification, or other tasks, including the use of pruning methods for optimization and evaluation, classify it as <Neural_Networks>.
<Rule_Learning>: If the paper discusses the construction of rules, propositionalization, or the use of stochastic methods in the context of inductive learning, classify it as <Rule_Learning>.
<Theory>: If the paper discusses theoretical aspects of computer science, such as algorithms, complexity, or mathematical models, classify it as <Theory>.

Note: Please ONLY summarize each category without omissions and NOTHING ELSE! And each category is output in the following format:
<Case_Based>: Papers that include [your summary] should be categorized as <Case_Based>.
<Genetic_Algorithms>: Papers that include [your summary] should be categorized as <Genetic_Algorithms>.
<Neural_Networks>: Papers that include [your summary] should be categorized as <Neural_Networks>.
<Probabilistic_Methods>: Papers that include [your summary] should be categorized as <Probabilistic_Methods>.
<Reinforcement_Learning>: Papers that include [your summary] should be categorized as <Reinforcement_Learning>.
<Rule_Learning>: Papers that include [your summary] should be categorized as <Rule_Learning>.
<Theory>: Papers that include [your summary] should be categorized as <Theory>.
Please begin with <START_SUMMARY>

<START_SUMMARY>
<Case_Based>: Papers that discuss the use of past experiences, memory-based techniques, case retrieval, and adaptation processes to solve new problems, and emphasize the importance of explanation in goal-driven learning, including introspection and adaptation of explanation strategies.
<Genetic_Algorithms>: Papers that discuss the use of evolutionary principles, genetic algorithms, mutation, crossover, selection, and their applications, implementations, or comparisons with other optimization methods.
<Neural_Networks>: Papers that discuss artificial neural networks, their architecture, training methods, applications, connectionist approaches, radial basis function networks, and the convergence of the backpropagation algorithm.
<Probabilistic_Methods>: (No description available)
<Reinforcement_Learning>: (No description available)
<Rule_Learning>: Papers that discuss the construction of rules, propositionalization, and the use of stochastic methods in the context of inductive learning.
<Theory>: Papers that discuss theoretical aspects of computer science, such as algorithms, complexity, or mathematical models.
<END_SUMMARY>

---

**one-shot optimizer LLM CoT wo prior Step 1**

The categorization model performed the judgment of paper categories by using the following features unique to each category of papers as the basis for the judgment:
Judge from the information given which category the following essay belongs to.
I will provide you with the title, abstract, citation summary information of your paper, the categories judged by the classification model, and the real categories of the paper, and ask you to verify that the basis of judgment for the corresponding categories is accurate.
First of all, please compare whether the categories provided by the model and the actual categories match. If it matches, you will directly analyze whether you need to add new judgment basis to the corresponding category based on the information and original judgment basis of the paper, if you need to add it, if not, you don't need to change anything; if it doesn't match, please analyze why the model may classify this paper into the category based on the original judgment basis corresponding to the incorrect category, and then please summarize the modifications and adjustments to the original judgment basis of the two categories, you can choose to remove matches for this paper in the wrong category, add unique features in the true category, or adjust the basis of judgment according to what you think is correct. Describe each step of the reasoning process in detail, and give the final updated basis of judgment, enclosing the final decision in <NewPattern></NewPattern>.
For example:
Here is a one-shot example; for more details, please refer to the appendix.
====================================================
Now analyze the given paper:
====================================================
Paper information:
(Paper Title: Stochastic Propositionalization of Non-Determinate Background Knowledge )
(Abstract: It is a well-known fact that propositional learning algorithms require "good" features to perform well in practice. So a major step in data engineering for inductive learning is the construction of good features by domain experts. These features often represent properties of structured objects, where a property typically is the occurrence of a certain substructure having certain properties. To partly automate the process of "feature engineering", we devised an algorithm that searches for features which are defined by such substructures. The algorithm stochastically conducts a top-down search for first-order clauses, where each clause represents a binary feature. It differs from existing algorithms in that its search is not class-blind, and that it is capable of considering clauses ("context") of almost arbitrary length (size). Preliminary experiments are favorable, and support the view that this approach is promising.)
(Neighbour Summary:
The papers cited in this essay discuss various techniques and approaches in the field of Rule Learning, specifically in Inductive Logic Programming (ILP). The first paper re-appraises the development of techniques for inverting deduction and introduces Mode-Directed Inverse Entailment (MDIE) as a generalisation and enhancement of previous approaches. The second paper proposes an extension of the feature-vector representation to allow the value of a feature to be a set of strings, which can be efficiently and naturally represented in real-world learning problems. The third paper describes a system named SPOIL that uses a stochastic search method to alleviate the local optimization problem in greedy algorithms. Overall, these papers highlight the importance of efficient and effective techniques in Rule Learning and ILP.)
Model judgment category:
Rule_Learning
Paper true category:
Rule_Learning

Step-by-Step Analysis:

1. **Model Judgment and True Category Comparison**
- **Model Judgment Category**: *<Rule_Learning>*
- **True Category**: *<Rule_Learning>*
The model has correctly classified this paper under the *<Rule_Learning>* category. Since both the model's judgment and the true category match, we move on to analyzing whether the original judgment basis for *<Rule_Learning>* is accurate or needs to be updated based on this paper's information.
2. **Examination of the Paper Information (Title, Abstract, Neighbour Summary)**
- **Paper Title**: *Stochastic Propositionalization of Non-Determinate Background Knowledge*
- The title suggests a focus on propositionalization and stochastic methods, which are relevant to *Rule_Learning* but do not directly indicate the category.
- **Abstract**:
- The abstract discusses the construction of features by domain experts and the automation of this process through an algorithm that searches for features defined by substructures. It also mentions the use of stochastic search for first-order clauses, which is a key aspect of *Rule_Learning*.
- The mention of *Inductive Learning* and *Rule Learning* in the context of propositionalization and feature construction directly points to the *Rule_Learning* category.
- **Neighbour Summary**:
- The neighbouring papers focus on *Rule Learning*, specifically in the context of *Inductive Logic Programming (ILP)*. They discuss techniques for inverting deduction, feature-vector representation, and stochastic search methods, all of which are relevant to *Rule_Learning*.
3. **Evaluation of the Existing Judgment Basis for <Rule_Learning>**
The current judgment basis for *<Rule_Learning>* is as follows:
- *If the paper discusses the construction of rules, propositionalization, or the use of stochastic methods in the context of inductive learning, classify it as <Rule_Learning>.*
The paper fits this description well:
- **Construction of rules**: The paper discusses the construction of features by domain experts and the automation of this process, which involves constructing rules or features from substructures.
- **Propositionalization**: The paper focuses on propositionalization of non-determinate background knowledge, which is a key aspect of *Rule_Learning*.
- **Stochastic methods**: The paper uses stochastic search for first-order clauses, which is a stochastic method relevant to *Rule_Learning*.
4. **Determining the Need for Modifications**
The current judgment basis for *<Rule_Learning>* is well-aligned with the content of this paper. The focus on rule construction, propositionalization, and stochastic methods are clearly articulated in the abstract and neighbor summary.
- **No new features need to be added** to the *<Rule_Learning>* category based on this paper, as it fits perfectly within the existing framework.

Updated judgment basis:

<NewPattern>Rule_Learning: If the paper discusses the construction of rules, propositionalization, or the use of stochastic methods in the context of inductive learning, classify it as <Rule_Learning>.</NewPattern>

## D.2 ONE-SHOT W/O PRIOR SUMMARY + VGRL STEP 2

**one-shot predictor LLM CoT wo prior Step 2**

The feedback provided by the optimization model suggests that the individual categories of papers have their own unique characteristics:

<Case_Based>: Papers that discuss the use of past experiences, memory-based techniques, case retrieval, and adaptation process to solve new problems, and emphasize the importance of explanation in goal-driven learning, including introspection and adaptation of explanation strategies.
<Genetic_Algorithms>: Papers that discuss the use of evolutionary principles, genetic algorithm, mutation, crossover, selection, and their applications, implementations, or comparisons with other optimization methods.
<Neural_Networks>: Papers that discuss artificial neural networks, their architecture, training methods, applications, connectionist approaches, radial basis function networks, and the convergence of the backpropagation algorithm.
<Probabilistic_Methods>: (No description available)
<Reinforcement_Learning>: (No description available)
<Rule_Learning>: Papers that discuss the construction of rules, propositionalization, and the use of stochastic methods in the context of inductive learning.
<Theory>: Papers that discuss theoretical aspects of computer science, such as algorithms, complexity, or mathematical models.

I will provide you with the title, abstract, and citation summary of your paper and ask you to step-by-step determine if the paper fits into each category. First, please analyze whether the paper matches category <Case_Based>. If it does, please explain how it matches; if it doesn't, please point out where it doesn't. Then please compare the categories <Genetic_Algorithms> and vice versa. After comparing all the categories, please decide which category the paper fits better, and give a final classification decision, please enclose the final decision in <>.
Now analyze the information from the given paper:
=============================================
Here is a one-shot example; for more details, please refer to the appendix.
=============================================
Paper information:
(Paper Title: Learning from positive data )
(Abstract: Gold showed in 1967 that even regular grammars can be exactly identified from positive examples alone. Since it is known that children learn natural grammars almost exclusively from positives examples, Gold's result has been used as a theoretical support for Chomsky's theory of innate human linguistic abilities. In this paper new results are presented which show that within a Bayesian framework not only grammars, but also logic programs are learnable with arbitrarily low expected error from positive examples only. In addition, we show that the upper bound for expected error of a learner which maximises the Bayes' posterior probability when learning from positive examples is within a small additive term of one which does the same from a mixture of positive and negative examples. An Inductive Logic Programming implementation is described which avoids the pitfalls of greedy search by global optimisation of this function during the local construction of individual clauses of the hypothesis. Results of testing this implementation on artificially-generated data-sets are reported. These results are in agreement with the theoretical predictions. )
(Neighbour Summary:
The papers cited in this essay discuss the limitations of current inductive logic programming systems in handling noise and learning recursive predicates. A new framework for learning from noisy data and fixed example size is presented, which incorporates a Bayesian heuristic for finding the most probable hypothesis. This approach is implemented in an ILP system called Lime, which is shown to handle noise better than other systems like FOIL and PROGOL, and is capable of learning from only positive or negative data.
Common points that reflect the category of this paper include:
- Handling noise in inductive logic programming systems
- Learning recursive predicates
- Improving the performance of ILP systems
- Developing new frameworks for learning from noisy data.)

Judgment:

The paper titled "Learning from positive data" discusses the theoretical and practical aspects of learning from positive examples alone, which is a key concept in genetic algorithms and evolutionary computation. The paper presents new results on the learnability of logic programs from positive examples within a Bayesian framework, which is a common approach in genetic algorithms.
The abstract and the cited papers highlight the importance of handling noise and learning recursive predicates, both of which are central to genetic algorithms and evolutionary computation.
The neighbour summary further supports the categorization, discussing the limitations of current ILP systems and the development of new frameworks for learning from noisy data, which is a key aspect of genetic algorithms.

Final decision:

<Genetic_Algorithms>

---

**one-shot summary LLM CoT wo prior Step 2**

Now that you have some descriptions, which will be used as rubrics for the classification task, you need to summarize and refine them for each category to find the best way to express them.
Here are the descriptions:

<Case_Based>: If the paper discusses the use of past experiences, memory-based techniques, or case retrieval and adaptation processes to solve new problems, classify it as <Case_Based>. If the paper discusses the use of past experiences, memory-based techniques, or case retrieval and adaptation processes to solve new problems, classify it as <Case_Based>.
<Genetic_Algorithms>: If the paper discusses the use of evolutionary principles, genetic algorithm, mutation, crossover, selection, and their applications, implementations, or comparisons with other optimization methods, classify it as <Genetic_Algorithms>. Papers that discuss learning from positive data, Bayesian frameworks, or ILP systems.
<Neural_Networks>: Papers that discuss artificial neural networks, their architecture, training methods, applications, connectionist approaches, radial basis function networks, and the convergence of the backpropagation algorithm.
<Probabilistic_Methods>: (No description available)
<Reinforcement_Learning>: (No description available)
<Rule_Learning>: Papers that discuss the construction of rules, propositionalization, and the use of stochastic methods in the context of inductive learning.
<Theory>: If the paper discusses theoretical aspects of computer science, such as algorithms, complexity, or mathematical models, classify it as <Theory>. If the paper discusses theoretical aspects of computer science, such as algorithms, complexity, or mathematical models, classify it as <Theory>. If the paper discusses theoretical aspects of computer science, such as algorithms, complexity, or mathematical models, classify it as <Theory>.

Note: Please ONLY summarize each category without omissions and NOTHING ELSE! And each category is output in the following format:
<Case_Based>: Papers that include [your summary] should be categorized as <Case_Based>.
<Genetic_Algorithms>: Papers that include [your summary] should be categorized as <Genetic_Algorithms>.
<Neural_Networks>: Papers that include [your summary] should be categorized as <Neural_Networks>.
<Probabilistic_Methods>: Papers that include [your summary] should be categorized as <Probabilistic_Methods>.
<Reinforcement_Learning>: Papers that include [your summary] should be categorized as <Reinforcement_Learning>.
<Rule_Learning>: Papers that include [your summary] should be categorized as <Rule_Learning>.
<Theory>: Papers that include [your summary] should be categorized as <Theory>.
Please begin with <START_SUMMARY>

<START_SUMMARY>
<Case_Based>: Papers that discuss the use of past experiences, memory-based techniques, or case retrieval and adaptation processes to solve new problems should be categorized as <Case_Based>.
<Genetic_Algorithms>: Papers that discuss the use of evolutionary principles, genetic algorithm, mutation, crossover, selection, and their applications, implementations, or comparisons with other optimization methods, excluding papers that discuss learning from positive data, Bayesian frameworks, or ILP systems, should be categorized as <Genetic_Algorithms>.
<Neural_Networks>: Papers that discuss artificial neural networks, their architecture, training methods, applications, connectionist approaches, radial basis function networks, and the convergence of the backpropagation algorithm should be categorized as <Neural_Networks>.
<Probabilistic_Methods>: Papers that discuss probabilistic approaches, statistical models, or uncertainty management in machine learning should be categorized as <Probabilistic_Methods>.
<Reinforcement_Learning>: Papers that discuss the use of trial and error, rewards, penalties, or other feedback mechanisms to learn from interactions with an environment should be categorized as <Reinforcement_Learning>.
<Rule_Learning>: Papers that discuss the construction of rules, propositionalization, and the use of stochastic methods in the context of inductive learning should be categorized as <Rule_Learning>.
<Theory>: Papers that discuss theoretical aspects of computer science, such as algorithms, complexity, mathematical models, or feature selection, especially in the context of machine learning, should be categorized as <Theory>.

---

**one-shot optimizer LLM CoT wo prior Step 2**

The categorization model performed the judgment of paper categories by using the following features unique to each category of papers as the basis for the judgment:

<Case_Based>: Papers that discuss the use of past experiences, memory-based techniques, case retrieval, and adaptation processes to solve new problems, including introspection and adaptation of explanation strategies.
<Genetic_Algorithms>: Papers that discuss the use of evolutionary principles, genetic algorithms, mutation, crossover, selection, and their applications, implementations, or comparisons with other optimization methods.
<Neural_Networks>: Papers that discuss artificial neural networks, their architecture, training methods, applications, connectionist approaches, radial basis function networks, and the convergence of the backpropagation algorithm.
<Probabilistic_Methods>: (No description available)
<Reinforcement_Learning>: (No description available)
<Rule_Learning>: Papers that discuss the construction of rules, propositionalization, and the use of stochastic methods in the context of inductive learning.
<Theory>: Papers that discuss theoretical aspects of computer science, such as algorithms, complexity, or mathematical models.

I will provide the title, abstract, citation summary information of your paper, the categories judged by the classification model, and the real categories of the paper, and ask you to verify that the basis of judgment for the corresponding categories is accurate.
First of all, please compare whether the categories provided by the model and the actual categories match. If it matches, you will directly analyze whether you need to add new judgment basis to the corresponding category based on the information and original judgment basis of the paper, if you need to add it, if not, you don't need to change anything; if it doesn't match, please analyze why the model may classify this paper into the category based on the original judgment basis corresponding to the incorrect category, and then please summarize the modifications and adjustments to the original judgment basis of the two categories, you can choose to remove matches for this paper in the wrong category, add unique features in the true category, or adjust the basis of judgment according to what you think is correct. Describe each step of the reasoning process in detail, and give the final updated basis of judgment, enclosing the final decision in <NewPattern></NewPattern>.
For example:
Here is a one-shot example; for more details, please refer to the appendix.
=============================================
Now analyze the given paper:
=============================================
Paper information:
(Paper Title: Learning from positive data )
(Abstract: Gold showed in 1967 that even regular grammars can be exactly identified from positive examples alone. Since it is known that children learn natural grammars almost exclusively from positives examples, Gold's result has been used as a theoretical support for Chomsky's theory of innate human linguistic abilities. In this paper new results are presented which show that within a Bayesian framework not only grammars, but also logic programs are learnable with arbitrarily low expected error from positive examples only. In addition, we show that the upper bound for expected error of a learner which maximises the Bayes' posterior probability when learning from positive examples is within a small additive term of one which does the same from a mixture of positive and negative examples. An Inductive Logic Programming implementation is described which avoids the pitfalls of greedy search by global optimisation of this function during the local construction of individual clauses of the hypothesis. Results of testing this implementation on artificially-generated data-sets are reported. These results are in agreement with the theoretical predictions. )
(Neighbour Summary:
The papers cited in this essay discuss the limitations of current inductive logic programming systems in handling noise and learning recursive predicates. A new framework for learning from noisy data and fixed example size is presented, which incorporates a Bayesian heuristic for finding the most probable hypothesis. This approach is implemented in an ILP system called Lime, which is shown to handle noise better than other systems like FOIL and PROGOL, and is capable of learning from only positive or negative data.
Common points that reflect the category of this paper include:
- Handling noise in inductive logic programming systems
- Learning recursive predicates
- Improving the performance of ILP systems
- Developing new frameworks for learning from noisy data.)
Model judgment category:
Genetic_Algorithms
Paper true category:
Theory

Step-by-Step Analysis:

1. **Model Judgment and True Category Comparison**
- **Model Judgment Category**: *<Genetic_Algorithms>*
- **True Category**: *<Theory>*
The model has incorrectly classified this paper under the *<Genetic_Algorithms>* category. The true category is *<Theory>*. We need to analyze why the model might have made this mistake and adjust the judgment basis accordingly.
2. **Examination of the Paper Information (Title, Abstract, Neighbour Summary)**
- **Paper Title**: *Learning from positive data*
- The title does not directly suggest a focus on *Genetic_Algorithms*, but rather on learning from positive data, which could be related to various areas of machine learning and theory.
- **Abstract**:
- The abstract discusses the learnability of logic programs and grammars from positive examples within a Bayesian framework. It also mentions the implementation of an Inductive Logic Programming system and its performance on artificial data-sets. There is no mention of genetic algorithms, mutation, crossover, selection, or evolutionary principles.
- **Neighbour Summary**:
- The neighbouring papers focus on the limitations of current inductive logic programming systems, the development of new frameworks for learning from noisy data, and the improvement of ILP systems. These topics are more aligned with theoretical aspects of computer science and machine learning than with genetic algorithms.
3. **Evaluation of the Existing Judgment Basis for <Genetic_Algorithms>**
The current judgment basis for *<Genetic_Algorithms>* is as follows:
- *If the paper discusses the use of evolutionary principles, genetic algorithms, mutation, crossover, selection, and their applications, implementations, or comparisons with other optimization methods, classify it as <Genetic_Algorithms>.*
The paper does not fit this description:
- **Evolutionary principles**: There is no mention of evolutionary principles or genetic algorithms in the abstract or neighbor summary.
- **Genetic algorithms**: The paper does not discuss genetic algorithms or any of their components.
4. **Determining the Need for Modifications**
The current judgment basis for *<Genetic_Algorithms>* is not aligned with the content of this paper. The focus on learning from positive data, Bayesian frameworks, and ILP systems does not match the criteria for *<Genetic_Algorithms>*.
- **Remove the matches for this paper in the wrong category**: The paper should not be classified under *<Genetic_Algorithms>*.
- **Add unique features in the true category**: The paper should be classified under *<Theory>* because it discusses theoretical aspects of computer science, such as the learnability of logic programs and grammars within a Bayesian framework.
- **Adjust the basis of judgment**: The judgment basis for *<Genetic_Algorithms>* should be updated to exclude papers that discuss learning from positive data, Bayesian frameworks, or ILP systems.

Updated judgment basis:

<NewPattern>Genetic_Algorithms: If the paper discusses the use of evolutionary principles, genetic algorithms, mutation, crossover, selection, and their applications, implementations, or comparisons with other optimization methods, classify it as <Genetic_Algorithms>. Exclude papers that discuss learning from positive data, Bayesian frameworks, or ILP systems.</NewPattern>

### D.3 ONE-SHOT W/O PRIOR SUMMARY + VGRL STEP 80

**one-shot predictor LLM CoT wo prior Step 80**

The feedback provided by the optimization model suggests that the individual categories of papers have their own unique characteristics:

<Case_Based>: Papers that discuss the use of past experiences, memory-based techniques, case retrieval, adaptation processes, continuous task domains, robotic navigation, knowledge compilation, dynamic selection and modification of behavior assemblages, and case-based reasoning framework should be categorized as <Case_Based>.
<Genetic_Algorithms>: Papers that discuss genetic algorithms, evolutionary processes, artificial life, theoretical biology, simulation, evolving neural networks, adapting to changing environments, optimization techniques involving genetic algorithms, and applications in machine learning, autonomous vehicle, and robotics, excluding cost-sensitive classification and robustness of induced models against cost changes, should be categorized as <Genetic_Algorithms>.
<Neural_Networks>: Papers that discuss artificial neural networks, their architecture, training methods, applications, connectionist approaches, neurobiology and cognitive psychology links, model calibration, learning parameter values, neural structures and functions, probabilistic methods, generative models, memory-based methods, approximation using radial basis functions, learning from prior knowledge, highly nonlinear models, pattern recognition, stability, and control in nonlinear systems should be categorized as <Neural_Networks>.
<Probabilistic_Methods>: Papers that discuss Bayesian inference, priors over model parameters, probabilistic approaches to neural networks, and the application of probabilistic methods to solve practical problems in the context of neural networks should be categorized as <Probabilistic_Methods>.
<Reinforcement_Learning>: Papers that discuss learning through trial and error, rewards, penalties, feedback mechanisms, temporal difference methods, model-based reinforcement learning, learning in stochastic systems, adapting traditional RL methods, exploration and navigation costs, using learned distances to guide exploration decisions, optimizing exploration based on expected benefits and costs, adapting to dynamic environments, using feedback mechanisms to guide decision-making, coordinating reactive behaviors, using neural networks as part of a control system, and temporal difference learning, excluding theoretical foundations and proof-based learning, should be categorized as <Reinforcement_Learning>.
<Rule_Learning>: Papers that discuss constructing rules, adapting methods for attribute estimation, non-myopic learning of regression trees, transforming regression algorithms, ordinal classification tasks, improving learning algorithms, addressing concept overlap problems, feature selection or greedy algorithms, employing stable models as background knowledge for a top-down ILP learner, improving existing machine learning algorithms, addressing concept overlap, using stochastic complexity formulas, or employing simulated annealing-based beam search should be categorized as <Rule_Learning>.
<Theory>: Papers that discuss theoretical aspects of learning, implications of findings on learning processes, foundational principles of machine learning, decision trees, ensemble learning, efficiency improvements as a primary focus, or practical applications of these theoretical aspects should be categorized as <Theory>.

I will provide you with the title, abstract, and citation summary of your paper and ask you to step-by-step determine if the paper fits into each category. First, please explain how it matches; if it doesn't, please point out where it doesn't. Then please compare the categories <Case_Based>. If it does, please explain how it matches; if it doesn't, please point out where it doesn't. Then please compare the categories <Genetic_Algorithms> and vice versa. After comparing all the categories, please decide which category the paper fits better, and give a final classification decision, please enclose the final decision in <>.
Now analyze the information from the given paper:
=====================================================
Here is a one-shot example; for more details, please refer to the appendix.
=====================================================
Paper information:
(Paper Title: Cognitive Computation (Extended Abstract) )
(Abstract: Cognitive computation is discussed as a discipline that links together neurobiology, cognitive psychology and artificial intelligence. )
(Neighbour Summary:
The papers cited in this essay discuss the concept of learning and reasoning, specifically focusing on the Probably Approximately Correct (PAC) learning model and its generalizations. The authors investigate the idea of agnostic learning, where minimal assumptions are made about the target function, and explore the possibilities of this approach through various results and algorithms. The papers also touch on the Learning to Reason framework, which combines learning and reasoning into a single task, and examine the tradeoff between learnability, oracle strength, and reasoning query range. Overall, the cited papers contribute to the understanding of learning and reasoning in artificial intelligence.)

Judgment:

The paper titled "Cognitive Computation (Extended Abstract)" discusses the discipline of cognitive computation, which links neurobiology, cognitive psychology, and artificial intelligence. However, the abstract does not explicitly discuss any of the topics mentioned in the <Case_Based> category, such as case retrieval, adaptation processes, continuous task domains, or case-based reasoning framework.
The neighbour summary discusses the Probably Approximately Correct (PAC) learning model, agnostic learning, and the Learning to Reason framework, which are more closely related to the Neural Networks category, as they involve learning and reasoning in artificial intelligence.

Final decision:

<Neural_Networks>

---

**one-shot summary LLM CoT wo prior Step 80**

Now that you have some descriptions, which will be used as rubrics for the classification task, you need to summarize and refine them for each category to find the best way to express them.
Here is the description:

<Case_Based>: If the paper discusses the use of past experiences, memory-based techniques, or case retrieval and adaptation processes to solve new problems, classify it as <Case_Based>. If the paper discusses the use of past experiences, memory-based techniques, or case retrieval and adaptation processes to solve new problems, classify it as <Case_Based>.
<Genetic_Algorithms>: If the paper discusses genetic algorithms, evolutionary processes, artificial life, theoretical biology, simulation, evolving neural networks, adapting to changing environments, optimization techniques involving genetic algorithms, and applications in machine learning, autonomous vehicles, and robotics should be categorized as <Genetic_Algorithms>.
<Neural_Networks>: If the paper discusses artificial neural networks, their architecture, training methods, applications, connectionist approaches, neurobiology and cognitive psychology links, model calibration, learning parameter values, neural structures and functions, probabilistic methods, generative models, memory-based methods, approximation using radial basis functions, learning from prior knowledge, highly nonlinear models, pattern recognition, stability, and control in nonlinear systems, or cognitive psychology as relevant topics that could be studied using neural networks, classify it as <Neural_Networks>.
<Probabilistic_Methods>: If the paper discusses Bayesian inference, priors over model parameters, or the application of probabilistic methods to solve practical problems in the context of neural networks, classify it as <Probabilistic_Methods>. Exclude papers that discuss theoretical aspects of learning, such as cross-validation and its implications. If the paper discusses Bayesian inference, priors over model parameters, or the application of probabilistic methods to solve practical problems in the context of neural networks, classify it as <Probabilistic_Methods>.
<Reinforcement_Learning>: Papers that discuss learning through trial and error, rewards, penalties, feedback mechanisms, temporal difference methods, model-based reinforcement learning, learning in stochastic systems, adapting traditional RL methods, exploration and navigation costs, using learned distances to guide exploration decisions, optimizing exploration based on expected benefits and costs, adapting to dynamic environments, using feedback mechanisms to guide decision-making, coordinating reactive behaviors, using neural networks as part of a control system, and temporal difference learning, excluding theoretical foundations and proof-based learning, should be categorized as <Reinforcement_Learning>.
<Rule_Learning>: If the paper discusses constructing rules, adapting methods for attribute estimation, non-myopic learning of regression trees, transforming regression algorithms, ordinal classification tasks, improving learning algorithms, addressing concept overlap problems, feature selection or greedy algorithms, employing stable models as background knowledge for a top-down ILP learner, improving existing machine learning algorithms, addressing concept overlap, using stochastic complexity formulas, or employing simulated annealing-based beam search should be categorized as <Rule_Learning>. However, exclude papers that focus on rule generation and compaction within a broader knowledge discovery and management framework. If the paper discusses constructing rules, adapting methods for attribute estimation, non-myopic learning of regression trees, transforming regression algorithms, ordinal classification tasks, improving learning algorithms, addressing concept overlap, employing stable models as background knowledge for a top-down ILP learner, improving existing machine learning algorithms, addressing concept overlap, using stochastic complexity formulas, or employing simulated annealing-based beam search, classify it as <Rule_Learning>.
<Theory>: Papers that discuss theoretical aspects of learning, implications of findings on learning processes, foundational principles of machine learning, decision trees, ensemble learning, efficiency improvements as a primary focus, or practical applications of these theoretical aspects should be categorized as <Theory>.

Note: Please ONLY summarize each category without omissions and NOTHING ELSE! And each category is output in the following format:
<Case_Based>: Papers that include [your summary] should be categorized as <Case_Based>.
<Genetic_Algorithms>: Papers that include [your summary] should be categorized as <Genetic_Algorithms>.
<Neural_Networks>: Papers that include [your summary] should be categorized as <Neural_Networks>.
<Probabilistic_Methods>: Papers that include [your summary] should be categorized as <Probabilistic_Methods>.
<Reinforcement_Learning>: Papers that include [your summary] should be categorized as <Reinforcement_Learning>.
<Rule_Learning>: Papers that include [your summary] should be categorized as <Rule_Learning>.
<Theory>: Papers that include [your summary] should be categorized as <Theory>.
Please begin with [START_SUMMARY]

<START_SUMMARY>
<Case_Based>: Papers that discuss the use of past experiences, memory-based techniques, or case retrieval and adaptation processes to solve new problems should be categorized as <Case_Based>.
<Genetic_Algorithms>: Papers that discuss genetic algorithms, evolutionary processes, artificial life, theoretical biology, simulation, evolving neural networks, adapting to changing environments, optimization techniques involving genetic algorithms, and applications in machine learning, autonomous vehicles, and robotics should be categorized as <Genetic_Algorithms>.
<Neural_Networks>: Papers that discuss artificial neural networks, their architecture, training methods, applications, connectionist approaches, neurobiology and cognitive psychology links, model calibration, learning parameter values, neural structures and functions, probabilistic methods, generative models, memory-based methods, approximation using radial basis functions, learning from prior knowledge, highly nonlinear models, pattern recognition, stability, and control in nonlinear systems, or cognitive psychology as relevant topics that could be studied using neural networks should be categorized as <Neural_Networks>.
<Probabilistic_Methods>: Papers that discuss Bayesian inference, priors over model parameters, or the application of probabilistic methods to solve practical problems in the context of neural networks such as cross-validation and its implications should be categorized as <Probabilistic_Methods>.
<Reinforcement_Learning>: Papers that discuss learning through trial and error, rewards, penalties, feedback mechanisms, temporal difference methods, model-based reinforcement learning, learning in stochastic systems, adapting traditional RL methods, exploration and navigation costs, using learned distances to guide exploration decisions, optimizing exploration based on expected benefits and costs, adapting to dynamic environments, using feedback mechanisms to guide decision-making, coordinating reactive behaviors, using neural networks as part of a control system, and temporal difference learning, excluding theoretical foundations and proof-based learning should be categorized as <Reinforcement_Learning>.
<Rule_Learning>: Papers that discuss constructing rules, adapting methods for attribute estimation, non-myopic learning of regression trees, transforming regression algorithms, ordinal classification tasks, improving learning algorithms, addressing concept overlap problems, feature selection or greedy algorithms, employing stable models as background knowledge for a top-down ILP learner, improving existing machine learning algorithms, addressing concept overlap, using stochastic complexity formulas, or employing simulated annealing-based beam search should be categorized as <Rule_Learning>.
<Theory>: Papers that discuss theoretical aspects of learning, implications of findings on learning processes, foundational principles of machine learning, decision trees, ensemble learning, efficiency improvements as a primary focus, or practical applications of these theoretical aspects should be categorized as <Theory>.

---

**one-shot optimizer LLM CoT wo prior Step 2**

The categorization model performed the judgment of paper categories by using the following features unique to each category of papers as the basis for the judgment:

<Case_Based>: Papers that discuss the use of past experiences, memory-based techniques, case retrieval, adaptation processes, continuous task domains, robotic navigation, knowledge compilation, dynamic selection and modification of behavior assemblages, and case-based reasoning framework should be categorized as <Case_Based>.
<Genetic_Algorithms>: Papers that discuss genetic algorithms, evolutionary processes, artificial life, theoretical biology, simulation, evolving neural networks, adapting to changing environments, optimization techniques involving genetic algorithms, and applications in machine learning, autonomous vehicles, and robotics, should be categorized as <Genetic_Algorithms>.
<Neural_Networks>: Papers that discuss artificial neural networks, their architecture, training methods, applications, connectionist approaches, neurobiology and cognitive psychology links, model calibration, learning parameter values, neural structures and functions, probabilistic methods, generative models, memory-based methods, approximation using radial basis functions, learning from prior knowledge, highly nonlinear models, pattern recognition, stability, and control in nonlinear systems should be categorized as <Neural_Networks>.
<Probabilistic_Methods>: Papers that discuss Bayesian inference, priors over model parameters, probabilistic approaches to neural networks, and the application of probabilistic methods to solve practical problems in the context of neural networks should be categorized as <Probabilistic_Methods>.
<Reinforcement_Learning>: Papers that discuss learning through trial and error, rewards, penalties, feedback mechanisms, temporal difference methods, model-based reinforcement learning, learning in stochastic systems, adapting traditional RL methods, exploration and navigation costs, using learned distances to guide exploration decisions, optimizing exploration based on expected benefits and costs, adapting to dynamic environments, using feedback mechanisms to guide decision-making, coordinating reactive behaviors, using neural networks as part of a control system, and temporal difference learning, excluding theoretical foundations and proof-based learning, should be categorized as <Reinforcement_Learning>.
<Rule_Learning>: Papers that discuss constructing rules, adapting methods for attribute estimation, non-myopic learning of regression trees, transforming regression algorithms, ordinal classification tasks, improving learning algorithms, addressing concept overlap problems, feature selection or greedy algorithms, employing stable models as background knowledge for a top-down ILP learner, improving existing machine learning algorithms, addressing concept overlap, using stochastic complexity formulas, or employing simulated annealing-based beam search should be categorized as <Rule_Learning>.
<Theory>: Papers that discuss theoretical aspects of learning, implications of findings on learning processes, foundational principles of machine learning, decision trees, ensemble learning, efficiency improvements as a primary focus, or practical applications of these theoretical aspects should be categorized as <Theory>.

I will provide you with the title, abstract, citation summary information of your paper, the categories judged by the classification model, and the real categories of the paper, and ask you to verify that the basis of judgment for the corresponding categorie is accurate.
First of all, please compare whether the categories provided by the model and the actual categories match. If it matches, you will directly analyze whether you need to add new judgment basis to the corresponding category based on the information and original judgment basis of the paper, if you need to add it, if not, you don't need to change anything; if it doesn't match, please analyze why the model may classify this paper into the category based on the original judgment basis corresponding to the incorrect category, and then please summarize the modifications and adjustments to the original judgment bases of the two categories, and you can choose to remove matches for this paper in the wrong category, add unique features in the true category, or adjust the basis of judgment according to what you think is correct. Describe each step of the reasoning process in detail, and give the final updated basis of judgment, enclosing the final decision in <NewPattern></NewPattern>.
For example:
Here is a one-shot example; for more details, please refer to the appendix.
=====================================================
Now analyze the given paper:
=====================================================
Paper information:
(Paper Title: Cognitive Computation (Extended Abstract) )
(Abstract: Cognitive computation is discussed as a discipline that links together neurobiology, cognitive psychology and artificial intelligence. )
(Neighbour Summary:
The papers cited in this essay discuss the concept of learning and reasoning, specifically focusing on the Probably Approximately Correct (PAC) learning model and its generalizations. The authors investigate the idea of agnostic learning, where minimal assumptions are made about the target function, and explore the possibilities of this approach through various results and algorithms. The papers also touch on the Learning to Reason framework, which combines learning and reasoning into a single task, and examine the tradeoff between learnability, oracle strength, and reasoning query range. Overall, the cited papers contribute to the understanding of learning and reasoning in artificial intelligence.)
Model judgment category:
Neural_Networks
Paper true category:
Neural_Networks

Step-by-Step Analysis:

1. **Model Judgment and True Category Comparison**
- **Model Judgment Category**: *<Neural_Networks>*
- **True Category**: *<Neural_Networks>*
The model has correctly classified this paper under the *<Neural_Networks>* category. Since both the model's judgment and the true category match, we move on to analyzing whether the original judgment basis for *<Neural_Networks>* is accurate or needs to be updated based on this paper's information.
2. **Examination of the Paper Information (Title, Abstract, Neighbour Summary)**
- **Paper Title**: *Cognitive Computation (Extended Abstract)*
- The title suggests a broad focus on cognitive processes, which could encompass various disciplines, including artificial neural networks.
- **Abstract**:
- The abstract introduces *cognitive computation* as a discipline that links neurobiology, cognitive psychology, and artificial intelligence. This broad scope does not directly mention neural networks but implies a connection to cognitive processes that could be studied using neural networks.
- **Neighbour Summary**:
- The neighbouring papers focus on learning and reasoning, specifically discussing the PAC learning model, agnostic learning, and the Learning to Reason framework. While these topics are related to machine learning and artificial intelligence, they do not directly relate to neural networks.
3. **Evaluation of the Existing Judgment Basis for <Neural_Networks>**
The current judgment basis for *<Neural_Networks>* is as follows:
- *If the paper discusses artificial neural networks, their architecture, training methods, applications, connectionist approaches, neurobiology and cognitive psychology links, model calibration, learning parameter values, neural structures and functions, probabilistic methods, generative models, memory-based methods, approximation using radial basis functions, learning from prior knowledge, highly nonlinear models, pattern recognition, stability, and control in nonlinear systems, classify it as <Neural_Networks>.*
The paper does not directly fit this description:
- **Artificial neural networks**: The abstract does not mention neural networks explicitly.
- **Neurobiology and cognitive psychology links**: While the paper discusses cognitive computation, it does not explicitly link this to neural networks or neurobiology.
4. **Determining the Need for Modifications**
The current judgment basis for *<Neural_Networks>* is not well-aligned with the content of this paper. The paper focuses on cognitive computation and its links to neurobiology and cognitive psychology, which does not directly relate to neural networks.
- **New features need to be added** to the *<Neural_Networks>* category based on this paper, specifically to include cognitive computation, neurobiology, and cognitive psychology as relevant topics that could be studied using neural networks.

Updated judgment basis:

<NewPattern>Neural_Networks: If the paper discusses artificial neural networks, their architecture, training methods, applications, connectionist approaches, neurobiology and cognitive psychology links, model calibration, learning parameter values, neural structures and functions, probabilistic methods, generative models, memory-based methods, approximation using radial basis functions, learning from prior knowledge, highly nonlinear models, pattern recognition, stability, and control in nonlinear systems, or cognitive computation, neurobiology, and cognitive psychology as relevant topics that could be studied using neural networks, classify it as <Neural_Networks>.</NewPattern>

# E   ONE-SHOT W/ PRIOR SUMMARY + VGRL

## E.1   ONE-SHOT W/ PRIOR SUMMARY + VGRL STEP 1

---

**one-shot predictor LLM CoT Step 2**

The feedback provided by the optimization model suggests that the individual categories of papers have their own unique characteristics.

<Case_Based>: Papers that include discussions of past experiences, memory-based technique, or case retrieval and adaptation processes to solve new problems should be categorized as <Case_Based>.
<Genetic_Algorithms>: Papers that focus on optimization techniques inspired by evolutionary processes, such as population-based search, crossover, and mutation should be categorized as <Genetic_Algorithms>.
<Neural_Networks>: Papers that cover artificial neural networks, including architectures like feedforward, convolutional, or recurrent networks, and their training methods should be categorized as <Neural_Networks>.
<Probabilistic_Methods>: Papers that discuss methods using probabilistic models to handle uncertainty, such as Bayesian networks or hidden Markov models should be categorized as <Probabilistic_Methods>.
<Reinforcement_Learning>: Papers that focus on agents learning from interaction with their environment through trial and error, using techniques like Q-learning or policy gradients should be categorized as <Reinforcement_Learning>.
<Rule_Learning>: Papers that focus on extracting interpretable decision rules or patterns, often in the form of logical clauses or "if-then" statements should be categorized as <Rule_Learning>.
<Theory>: Papers that emphasize theoretical contributions to machine learning or AI, such as mathematical models, complexity theory, or the formal analysis of algorithms, or discuss optimality, domination, and bounded convergence rates in the context of repeated games should be categorized as <Theory>.
I will provide you with the title, abstract, and citation summary of your paper and ask you to step-by-step determine if the paper fits into each category. First, please analyze whether the paper matches category <Case_Based>. If it does, please explain how it matches; if it doesn't, please point out where it doesn't. Then please compare the categories <Genetic_Algorithms> and vice versa. After comparing all the categories, please decide which category the paper fits better, and give a final classification decision, please enclose the final decision in <>.

Judge from the information given which category the following essay belongs to.
I will provide you with the title, abstract, and citation summary of your paper and ask you to step-by-step determine if the paper fits into each category. First, please analyze whether the paper matches category <Case_Based>. If it does, please explain how it matches; if it doesn't, please point out where it doesn't. Then please compare the categories <Genetic_Algorithms> and vice versa. After comparing all the categories, please decide which category the paper fits better, and give a final classification decision, please enclose the final decision in <>.
Now analyze the information from the given paper:
========================================
Here is a one-shot example; for more details, please refer to the appendix.

Paper information:
(Paper Title: Learning from positive data)
(Abstract: Gold showed in 1967 that not even regular grammars can be exactly identified from positive examples alone. Since it is known that children learn natural grammars almost exclusively from positive examples, Gold's result has been used as a theoretical support for Chomsky's theory of innate human linguistic abilities. In this paper new results are presented which show that within a Bayesian framework not only grammars, but also logic programs are learnable with arbitrarily low expected error from positive examples only. In addition, we show that the upper bound for expected error of a learner which maximizes the Bayes' posterior probability when learning from positive examples is within a small additive term of one which does the same from a mixture of positive and negative examples. An Inductive Logic Programming implementation is described which avoids the pitfalls of greedy search by global optimization of this function during the local construction of individual clauses of the hypothesis. Results of testing this implementation on artificially-generated data-sets are reported. These results are in agreement with the theoretical predictions.) (Neighbour Summary:
The papers cited in this essay discuss the limitations of current inductive logic programming systems in handling noise and learning recursive predicates. A new framework for learning from noisy data and fixed example size is presented, which incorporates a Bayesian heuristic for finding the most probable hypothesis. This approach is implemented in an ILP system called Lime, which is shown to handle noise better than other systems like FOIL and PROGOL, and is capable of learning from only positive or negative data. )

Judgment:
The paper titled "Learning from positive data" discusses the theoretical and practical aspects of learning from positive examples alone, focusing on the learnability of grammars and logic programs within a Bayesian framework. It presents new results on the learnability of logic programs and the upper bound for expected error of a learner when learning from positive examples.
The abstract and the cited papers highlight the importance of learning from positive data, which is a key aspect of the Genetic Algorithms category, as it involves optimization techniques inspired by evolutionary processes, such as population-based search, crossover, and mutation.
The neighbour summary further supports the categorization, discussing the limitations of current ILP systems and the development of new frameworks for learning from noisy data, which is a key aspect of Genetic Algorithms.

Final decision:
<Genetic_Algorithm>

---

**one-shot summary LLM CoT Step 1**

Now that you have some descriptions, which will be used as rubrics for the classification task, you need to summarize and refine them for each category to find the best way to express them.
Here are the description:

Judge from the information given which category the following essay belongs to.:
<Case_Based>: If the paper discusses the use of past experiences, memory-based techniques, or case retrieval and adaptation processes to solve new problems, classify it as <Case_Based>.
<Genetic_Algorithms>: If the paper focuses on optimization techniques inspired by evolutionary processes, such as population-based search, crossover, and mutation, classify it as <Genetic_Algorithms>.
<Neural_Networks>: If the paper covers artificial neural networks, including architectures like feedforward, convolutional, or recurrent networks, and their training methods, classify it as <Neural_Networks>.
If the paper covers artificial neural networks, including architectures like feedforward, convolutional, or recurrent networks, classify it as <Neural_Networks>.
<Probabilistic_Methods>: Papers that discuss methods using probabilistic models to handle uncertainty, such as Bayesian networks or hidden Markov models, classify it as <Probabilistic_Methods>.
<Reinforcement_Learning>: If the paper focuses on agents learning from interaction with their environment through trial and error, using techniques like Q-learning or policy gradients, classify it as <Reinforcement_Learning>.
<Rule_Learning>: If the paper focuses on extracting interpretable decision rules or patterns, often in the form of logical clauses or "if-then" statements, classify it as <Rule_Learning>.
<Theory>: If the paper emphasizes theoretical contributions to machine learning or AI, such as mathematical models, complexity theory, or the formal analysis of algorithms, or discuss optimality, domination, and bounded convergence rates in the context of repeated games, classify it as <Theory>.
If the paper emphasizes theoretical contributions to machine learning or AI, such as mathematical models, complexity theory, or the formal analysis of algorithms, classify it as <Theory>.
Note: Please ONLY summarize each category without omissions and NOTHING ELSE! And each category is output in the following format:
<Case_Based>: Papers that include [your summary] should be categorized as <Case_Based>.
<Genetic_Algorithms>: Papers that include [your summary] should be categorized as <Genetic_Algorithms>.
<Neural_Networks>: Papers that include [your summary] should be categorized as <Neural_Networks>.
<Probabilistic_Methods>: Papers that include [your summary] should be categorized as <Probabilistic_Methods>.
<Reinforcement_Learning>: Papers that include [your summary] should be categorized as <Reinforcement_Learning>.
<Rule_Learning>: Papers that include [your summary] should be categorized as <Rule_Learning>.
<Theory>: Papers that include [your summary] should be categorized as <Theory>.
Please begin with <START_SUMMARY>.

<START_SUMMARY>
<Case_Based>: Papers that include discussions of past experiences, memory-based techniques, or case retrieval and adaptation processes to solve new problems should be categorized as <Case_Based>.
<Genetic_Algorithms>: Papers that focus on optimization techniques inspired by evolutionary processes, such as population-based search, crossover, and mutation should be categorized as <Genetic_Algorithms>.
<Neural_Networks>: Papers that cover artificial neural networks, including architectures like feedforward, convolutional, or recurrent networks, and their training methods should be categorized as <Neural_Networks>.
<Probabilistic_Methods>: Papers that discuss methods using probabilistic models to handle uncertainty, such as Bayesian networks or hidden Markov models should be categorized as <Probabilistic_Methods>.
<Reinforcement_Learning>: Papers that focus on agents learning from interaction with their environment through trial and error, using techniques like Q-learning or policy gradients should be categorized as <Reinforcement_Learning>.
<Rule_Learning>: Papers that focus on extracting interpretable decision rules or patterns, often in the form of logical clauses or "if-then" statements should be categorized as <Rule_Learning>.
<Theory>: Papers that emphasize theoretical contributions to machine learning or AI, such as mathematical models, complexity theory, or the formal analysis of algorithms, or discuss optimality, domination, and bounded convergence rates in the context of repeated games should be categorized as <Theory>.
<END_SUMMARY>

---

**one-shot optimizer LLM CoT Step 1**

The categorization model performed the judgment of paper categories by using the following features unique to each category of papers as the basis for the judgment:

<Case_Based>: If the paper discusses the use of past experiences, memory-based techniques, or case retrieval and adaptation processes to solve new problems, classify it as <Case_Based>.
<Genetic_Algorithms>: If the paper focuses on optimization techniques inspired by evolutionary processes, such as population-based search, crossover, and mutation, classify it as <Genetic_Algorithms>.
<Neural_Networks>: If the paper covers artificial neural networks, including architectures like feedforward, convolutional, or recurrent networks, and their training methods, classify it as <Neural_Networks>.
<Probabilistic_Methods>: If the paper discusses methods that use probabilistic models to handle uncertainty, such as Bayesian networks or hidden Markov models, classify it as <Probabilistic_Methods>.
<Reinforcement_Learning>: If the paper centers on agents learning from interaction with their environment through trial and error, using techniques like Q-learning or policy gradients, classify it as <Reinforcement_Learning>.
<Rule_Learning>: If the paper focuses on extracting interpretable decision rules or patterns, often in the form of "if-then" statements, classify it as <Rule_Learning>.
<Theory>: If the paper emphasizes theoretical contributions to machine learning or AI, such as mathematical models, complexity theory, or the formal analysis of algorithms, classify it as <Theory>.

Judge from the information given which category the following essay belongs to.
I will provide the title, abstract, citation summary information of your paper, the categories judged by the classification model, and the real categories of the paper, and ask you to verify that the basis of judgment for the corresponding categories is accurate.
First of all, please compare whether the categories provided by the model and the actual categories match. If it matches, you don't directly analyze whether you need to add new judgment basis to the corresponding category based on the information and original judgment basis of the paper, if you need to add it, if not, you don't need to change anything; if it doesn't match, please analyze why the model may classify this paper into the category based on the original judgment basis corresponding to the incorrect category, and then please summarize the modifications and adjustments to the original judgment bases of the two categories, you can choose to remove matches for this paper in the wrong category, add unique features in the true category, or adjust the basis of judgment according to what you think is correct. Describe each step of the reasoning process in detail, and give the final updated basis of judgment, enclosing the final decision in <NewPattern></NewPattern>.
For example:
Here is a one-shot example; for more details, please refer to the appendix.
========================================
Now analyze the given paper:

Paper information:
(Paper Title: Stochastic Propositionalization of Non-Determinate Background Knowledge )
(Abstract: It is a well-known fact that propositional learning algorithms require "good" features to perform well in practice. So a major step in data engineering for inductive learning is the construction of good features by domain experts. These features often represent properties of structured objects, where a property typically is the occurrence of a certain substructure having certain properties. To partly automate the process of "feature engineering", we devised an algorithm that searches for features which are defined by such substructures. The algorithm stochastically conducts a top-down search for first-order clauses, where each clause represents a binary feature. It differs from existing algorithms in that its search is not class-blind, and that it is capable of considering clauses ("context") of almost arbitrary length (size). Preliminary experiments are favorable, and support the view that this approach is promising.)
(Neighbour Summary:
The papers cited in this essay discuss various techniques and approaches in the field of Rule Learning, specifically in Inductive Logic Programming (ILP). The first paper re-appraises the development of techniques for inverting deduction and introduces Mode-Directed Inverse Entailment (MDIE) as a generalisation and enhancement of previous approaches. The second paper proposes an extension of the feature-vector representation to allow the value of a feature to be a set of strings, which can be efficiently and naturally represented in real-world learning problems. The third paper describes a system named SFOIL that uses a stochastic search method to alleviate the local optimization problem in greedy algorithms. Overall, these papers highlight the importance of efficient and effective techniques in Rule Learning and ILP.)
Model judgment category:
Rule_Learning
Paper true category:
Rule_Learning

Step-by-Step Analysis:

1. **Model Judgment and True Category Comparison**
- **Model Judgment Category**: *<Rule_Learning>*
- **True Category**: *<Rule_Learning>*
The model has correctly classified this paper under the *<Rule_Learning>* category. Since both the model's judgment and the true category match, we move on to analyzing whether the original judgment basis for *<Rule_Learning>* is accurate or needs to be updated based on this paper's information.
2. **Examination of the Paper Information (Title, Abstract, Neighbour Summary)**
- **Paper Title**: *Stochastic Propositionalization of Non-Determinate Background Knowledge*
- The title suggests a focus on propositionalization and stochastic methods, which are relevant to *Rule_Learning* but not directly indicate the category.
- **Abstract**:
- The abstract discusses the construction of features by domain experts and the automation of this process through an algorithm that searches for features defined by substructures. It also mentions the use of stochastic search for first-order clauses, which is a key aspect of *Rule_Learning*.
- The mention of *Inductive Learning* and *Rule Learning* in the context of propositionalization and feature construction directly points to the *Rule_Learning* category.
- **Neighbour Summary**:
- The neighbouring papers focus on *Rule Learning*, specifically in the context of *Inductive Logic Programming (ILP)*. They discuss techniques for inverting deduction, feature-vector representation, and stochastic search methods, all of which are relevant to *Rule_Learning*.
3. **Evaluation of the Existing Judgment Basis for <Rule_Learning>**
- The current judgment basis for *<Rule_Learning>* is as follows:
- *If the paper discusses the construction of rules, propositionalization, or the use of stochastic methods in the context of inductive learning, classify it as <Rule_Learning>.*
The paper fits this description well:
- **Construction of rules**: The paper discusses the construction of features by domain experts and the automation of this process, which involves constructing rules or features from substructures.
- **Propositionalization**: The paper focuses on propositionalization of non-determinate background knowledge, which is a key aspect of *Rule_Learning*.
- **Stochastic methods**: The paper uses stochastic search for first-order clauses, which is a stochastic method relevant to *Rule_Learning*.
4. **Determining the Need for Modifications**
The current judgment basis for *<Rule_Learning>* is well-aligned with the content of this paper. The focus on rule construction, propositionalization, and stochastic methods are clearly articulated in the abstract and neighbor summary.
- **No new features need to be added** to the *<Rule_Learning>* category based on this paper, as it fits perfectly within the existing framework.

Updated judgment basis:

<NewPattern>Rule_Learning: If the paper discusses the construction of rules, propositionalization, or the use of stochastic methods in the context of inductive learning, classify it as <Rule_Learning>.</NewPattern>

---

## E.2 ONE-SHOT W/ PRIOR SUMMARY + VGRL STEP 2

---

**one-shot predictor LLM CoT Step 1**

The feedback provided by the optimization model suggests that the individual categories of papers have their own unique characteristics.

<Case_Based>: If the paper discusses the use of past experiences, memory-based techniques, or case retrieval and adaptation processes to solve new problems, classify it as <Case_Based>.
<Genetic_Algorithms>: If the paper focuses on optimization techniques inspired by evolutionary processes, such as population-based search, crossover, and mutation, classify it as <Genetic_Algorithms>.
<Neural_Networks>: If the paper covers artificial neural networks, including architectures like feedforward, convolutional, or recurrent networks, and their training methods, classify it as <Neural_Networks>.
<Probabilistic_Methods>: If the paper discusses methods that use probabilistic models to handle uncertainty, such as Bayesian networks or hidden Markov models, classify it as <Probabilistic_Methods>.
<Reinforcement_Learning>: If the paper centers on agents learning from interaction with their environment through trial and error, using techniques like Q-learning or policy gradients, classify it as <Reinforcement_Learning>.
<Rule_Learning>: If the paper focuses on extracting interpretable decision rules or patterns, often in the form of "if-then" statements, classify it as <Rule_Learning>.
<Theory>: If the paper emphasizes theoretical contributions to machine learning or AI, such as mathematical models, complexity theory, or the formal analysis of algorithms, classify it as <Theory>.
I will provide you with the title, abstract, and citation summary of your paper and ask you to step-by-step determine if the paper fits into each category. First, please analyze whether the paper matches category <Case_Based>. If it does, please explain how it matches; if it doesn't, please point out where it doesn't. Then please compare the categories <Genetic_Algorithms> and vice versa. After comparing all the categories, please decide which category the paper fits better, and give a final classification decision, please enclose the final decision in <>.

Judge from the information given which category the following essay belongs to.
I will provide you with the title, abstract, and citation summary of your paper and ask you to step-by-step determine if the paper fits into each category. First, please analyze whether the paper matches category <Case_Based>. If it does, please explain how it matches; if it doesn't, please point out where it doesn't. Then please compare the categories <Genetic_Algorithms> and vice versa. After comparing all the categories, please decide which category the paper fits better, and give a final classification decision, please enclose the final decision in <>.
Now analyze the information from the given paper:
================================================
Here is a one-shot example; for more details, please refer to the appendix.
================================================
Paper information:
(Paper Title: Stochastic Propositionalization of Non-Determinate Background Knowledge )
(Abstract: It is a well-known fact that propositional learning algorithms require "good" features to perform well in practice. So a major step in data engineering for inductive learning is the construction of good features by domain experts. These features often represent properties of structured objects, where a property typically is the occurrence of a certain substructure having certain properties. To partly automate the process of "feature engineering", we devised an algorithm that searches for features which are defined by such substructures. The algorithm stochastically conducts a top-down search for first-order clauses, where each clause represents a binary feature. It differs from existing algorithms in that its search is not class-blind, and that it is capable of considering clauses ("context") of almost arbitrary length (size). Preliminary experiments are favorable, and support the view that this approach is promising.)
(Neighbour Summary:
The papers cited in this essay discuss various techniques and approaches in the field of Rule Learning, specifically in Inductive Logic Programming (ILP). The first paper re-appraises the development of techniques for inverting deduction and introduces Mode-Directed Inverse Entailment (MDIE) as a generalisation and enhancement of previous approaches. The second paper proposes an extension of the feature-vector representation to allow the value of a feature to be a set of strings, which can be efficiently and naturally represented in real-world learning problems. The third paper describes a system named SFOIL that uses a stochastic search method to alleviate the local optimization problem in greedy algorithms. Overall, these papers highlight the importance of efficient and effective techniques in Rule Learning and ILP.)

Judgment:
The paper titled "Stochastic Propositionalization of Non-Determinate Background Knowledge" discusses the construction of good features for inductive learning by devising an algorithm that searches for features defined by substructures. The algorithm conducts a top-down search for first-order clauses, which represents a binary feature. This approach is different from existing algorithms in that its search is not class-blind and can consider clauses of almost arbitrary length.
The abstract and the cited papers highlight the importance of feature engineering and the use of stochastic search methods to alleviate local optimization problems, both of which are central concepts to Rule Learning and ILP.
The neighbour summary further supports the categorization, discussing the development of techniques for inductive logic programming and the importance of efficient and effective methods in rule learning.

Final decision:
<Rule_Learning>

---

**one-shot summary LLM CoT Step 2**

Now that you have some descriptions, which will be used as rubrics for the classification task, you need to summarize and refine them for each category to find the best way to express them.
Here are the descriptions:

Judge from the information given which category the following essay belongs to.
<Case_Based>: If the paper discusses the use of past experiences, memory-based techniques, or case retrieval and adaptation processes to solve new problems, classify it as <Case_Based>. If the paper discusses the use of past experiences, memory-based technique, or case retrieval and adaptation processes to solve new problems, classify it as <Case_Based>.
; <Genetic_Algorithms>: If the paper focuses on optimization techniques inspired by evolutionary processes, such as population-based search, crossover, and mutation, classify it as <Genetic_Algorithms>.
<Neural_Networks>: Papers that cover artificial neural networks, including architectures like feedforward, convolutional, or recurrent networks, and their training methods should be categorized as <Neural_Networks>.
<Probabilistic_Methods>: If the paper discusses methods using probabilistic models to handle uncertainty, such as Bayesian networks or hidden Markov models should be categorized as <Probabilistic_Methods>.
<Reinforcement_Learning>: Papers that focus on agents learning from interaction with their environment through trial and error, using techniques like Q-learning or policy gradients should be categorized as <Reinforcement_Learning>.
<Rule_Learning>: Papers that focus on extracting interpretable decision rules or patterns, often in the form of logical clauses or "if-then" statements should be categorized as <Rule_Learning>.
<Theory>: If the paper emphasizes theoretical contributions to machine learning or AI, such as mathematical models, complexity theory, or the formal analysis of algorithms, or discuss topics like optimality, domination, and bounded convergence rates in the context of repeated games, classify it as <Theory>. If the paper emphasizes theoretical contributions to machine learning or AI, such as mathematical models, complexity theory, or the formal analysis of algorithms, or discusses optimality, domination, and bounded convergence rates in the context of repeated games, classify it as <Theory>. Additionally, consider papers that analyze the bias and complexity of learning algorithms, propose new mathematical model for learning, or explore the efficiency and scalability of learning methods. If the paper emphasizes theoretical contributions to machine learning or AI, such as mathematical models, complexity theory, or the formal analysis of algorithms, or discusses optimality, domination, and bounded convergence rates in the context of repeated games, classify it as <Theory>. If the paper emphasizes theoretical contributions to machine learning or AI, such as mathematical models, complexity theory, or the formal analysis of algorithms, or discusses optimality, domination, and bounded convergence rates in the context of repeated games, theoretical contributions to machine learning or AI, such as mathematical models, complexity theory, or the formal analysis of algorithms, or discusses optimality, domination, and bounded convergence rates in the context of repeated games, classify it as <Theory>. Additionally, if the paper discusses theoretical models of learning, bounds on error rates, and the development of efficient algorithms, classify it as <Theory>. If the paper discusses theoretical contributions to machine learning or AI, such as mathematical models, complexity theory, or the formal analysis of algorithms, or discusses optimality, domination, and bounded convergence rates in the context of repeated games, or focuses on the learnability of concepts and the development of new learning techniques, classify it as <Theory>. If the paper emphasizes theoretical contributions to machine learning or AI, such as mathematical models, complexity theory, or the formal analysis of algorithms, or discusses optimality, domination, and bounded convergence rates in the context of repeated games, classify it as <Theory>.

Note: Please ONLY summarize each category without erroneous and NOTHING ELSE! And each category is output in the following format:
<Case_Based>: Papers that include [your summary] should be categorized as <Case_Based>.
<Genetic_Algorithms>: Papers that include [your summary] should be categorized as <Genetic_Algorithms>.
<Neural_Networks>: Papers that include [your summary] should be categorized as <Neural_Networks>.
<Probabilistic_Methods>: Papers that include [your summary] should be categorized as <Probabilistic_Methods>.
<Reinforcement_Learning>: Papers that include [your summary] should be categorized as <Reinforcement_Learning>.
<Rule_Learning>: Papers that include [your summary] should be categorized as <Rule_Learning>.
<Theory>: Papers that include [your summary] should be categorized as <Theory>.
Please begin with <START_SUMMARY>.

<START_SUMMARY>
<Case_Based>: Papers that discuss the use of past experiences, memory-based techniques, or case retrieval and adaptation processes to solve new problems should be categorized as <Case_Based>.
<Genetic_Algorithms>: Papers that focus on optimization techniques inspired by evolutionary processes, such as population-based search, crossover, and mutation should be categorized as <Genetic_Algorithms>.
<Neural_Networks>: Papers that cover artificial neural networks, including architectures like feedforward, convolutional, or recurrent networks, and their training methods should be categorized as <Neural_Networks>.
<Probabilistic_Methods>: Papers that discuss methods using probabilistic models to handle uncertainty, such as Bayesian networks or hidden Markov models should be categorized as <Probabilistic_Methods>.
<Reinforcement_Learning>: Papers that focus on agents learning from interaction with their environment through trial and error, using techniques like Q-learning or policy gradients should be categorized as <Reinforcement_Learning>.
<Rule_Learning>: Papers that focus on extracting interpretable decision rules or patterns, often in the form of logical clauses or "if-then" statements should be categorized as <Rule_Learning>.
<Theory>: Papers that emphasize theoretical contributions to machine learning or AI, such as mathematical models, complexity theory, or the formal analysis of algorithms, or discuss topics like optimality, domination, bounded convergence rates, bias, complexity, or the development of efficient algorithms should be categorized as <Theory>.
<END_SUMMARY>

---

**one-shot optimizer LLM CoT Step 2**

The categorization model performed the judgment of paper categories by using the following features unique to each category of papers as the basis for the judgment:

<Case_Based>: Papers that include discussions of past experiences, memory-based techniques, or case retrieval and adaptation processes to solve new problems should be categorized as <Case_Based>.
<Genetic_Algorithms>: Papers that focus on optimization techniques inspired by evolutionary processes, such as population-based search, crossover, and mutation should be categorized as <Genetic_Algorithms>.
<Neural_Networks>: Papers that cover artificial neural networks, including architectures like feedforward, convolutional, or recurrent networks, and their training methods should be categorized as <Neural_Networks>.
<Probabilistic_Methods>: Papers that discuss methods using probabilistic models to handle uncertainty, such as Bayesian networks or hidden Markov models should be categorized as <Probabilistic_Methods>.
<Reinforcement_Learning>: Papers that focus on agents learning from interaction with their environment through trial and error, using techniques like Q-learning or policy gradients should be categorized as <Reinforcement_Learning>.
<Rule_Learning>: Papers that focus on extracting interpretable decision rules or patterns, often in the form of logical clauses or "if-then" statements should be categorized as <Rule_Learning>.
<Theory>: Papers that emphasize theoretical contributions to machine learning or AI, such as mathematical models, complexity theory, or the formal analysis of algorithms, or discuss optimality, domination, and bounded convergence rates in the context of repeated games should be categorized as <Theory>.

Judge from the information given which category the following essay belongs to.
I will provide you with the title, abstract, citation summary of the paper, the categories judged by the classification model, and the real categories of the paper, and ask you to verify that the basis of judgment for the corresponding categories is accurate.
First of all, please compare whether the categories provided by the model and the actual categories match. If it matches, you will directly analyze whether you need to add new judgment basis to the corresponding category based on the information and original judgment basis of the paper, if you need to add it, if not, you don't need to change anything; if it doesn't match, please analyze why the model may classify this paper into the category based on the original judgment basis corresponding to the incorrect category, and then please summarize the modifications and adjustments to the original judgment bases of the two categories, you can choose to remove matches for this paper in the wrong category, add unique features in the true category, or adjust the basis of judgment according to what you think is correct. Describe each step of the reasoning process in detail, and give the final updated basis of judgment, enclosing the final decision in <NewPattern></NewPattern>.
For example:
Here is a one-shot example; for more details, please refer to the appendix.
================================================
Now analyze the given paper:
================================================
Paper information:
(Paper Title: Learning from positive data )
(Abstract: Gold showed in 1967 that not even regular grammars can be exactly identified from positive examples alone. Since it is known that children learn natural grammars almost exclusively from positives examples, Gold's result has been used as a theoretical support for Chomsky's theory of innate human linguistic abilities. In this paper new results are presented which show that within a Bayesian framework not only grammars, but also logic programs are learnable with arbitrarily expected error from positive examples only. In addition, we show that the upper bound for expected error of a learner which maximises the Bayes' posterior probability when learning from positive examples is within a small additive term of one which does the same from a mixture of positive and negative examples. An Inductive Logic Programming implementation is described which avoids the pitfalls of greedy search by global optimisation of this function during the local construction of individual clauses of the hypothesis. Results of testing this implementation on artificially-generated data-sets are reported. These results are in agreement with the theoretical predictions. )
(Neighbour Summary:
The papers cited in this essay discuss the limitations of current inductive logic programming systems in handling noise and learning recursive predicates. A new framework for learning from noisy data and fixed example size is presented, which incorporates a Bayesian heuristic for finding the most probable hypothesis. This approach is implemented in an ILP system called Lime, which is shown to handle noise better than other systems like FOIL and PROGOL, and is capable of learning from only positive or negative data.
Common points that reflect the category of this paper include:
- Handling noise in inductive logic programming systems
- Learning recursive predicates
- Improving the performance of ILP systems
- Developing new frameworks for learning from noisy data.) Model judgment category:
Genetic_Algorithms
Paper true category:
Theory

Step-by-Step Analysis:
1. **Model Judgment and True Category Comparison**
- **Model Judgment Category**: *<Genetic_Algorithms>*
- **True Category**: *<Theory>*
The model has incorrectly classified this paper under the *<Genetic_Algorithms>* category. Since the model's judgment and the true category do not match, we proceed to analyze why the model may have classified this paper into the incorrect category and make necessary adjustments to the judgment bases.
2. **Examination of the Paper Information (Title, Abstract, Neighbour Summary)**
- **Paper Title**: *Learning from positive data*
- The title suggests a focus on learning from positive data, which is more aligned with theoretical aspects of machine learning rather than genetic algorithms.
- **Abstract**:
- The abstract discusses the learnability of logic programs and grammars from positive examples within a Bayesian framework. It also presents results on the upper bound for expected error of a learner and describes an implementation in Inductive Logic Programming (ILP). These topics are more aligned with theoretical aspects of machine learning and ILP rather than genetic algorithms.
- **Neighbour Summary**:
- The neighbouring papers focus on learning from noisy data, the development of new frameworks for learning from noisy data, and improving the performance of ILP systems. These topics are also more aligned with theoretical aspects of machine learning and ILP.
3. **Evaluation of the Existing Judgment Basis for <Genetic_Algorithms>**
The current judgment basis for *<Genetic_Algorithms>* is as follows:
- *If the paper focuses on optimization techniques inspired by evolutionary processes, such as population-based search, crossover, and mutation, classify it as <Genetic_Algorithms>.*
The paper does not fit this description.
- **Evolutionary processes**: The paper discusses learning from positive data within a Bayesian framework and the implementation of an ILP system, which does not involve evolutionary processes.
4. **Evaluation of the Existing Judgment Basis for <Theory>**
The current judgment basis for *<Theory>* is as follows:
- *If the paper emphasizes theoretical contributions to machine learning or AI, such as mathematical models, complexity theory, or the formal analysis of algorithms, or discusses optimality, domination, and bounded convergence rates in the context of repeated games, classify it as <Theory>.* The paper fits this description well:
- **Theoretical contributions**: The paper presents new results on the learnability of logic programs and grammars from positive examples within a Bayesian framework, which is a theoretical contribution to machine learning.
- **Mathematical models**: The paper uses a Bayesian framework, which is a mathematical model for learning from positive data.
- **Formal analysis of algorithms**: The paper discusses the upper bound for expected error of a learner, which involves a formal analysis of an algorithm.
5. **Determining the Need for Modifications**
The current judgment basis for *<Theory>* is well-aligned with the content of this paper. The focus on theoretical contributions, mathematical models, and formal analysis of algorithms are clearly articulated in the abstract and neighbor summary.
- **Remove the match for this paper in the wrong category <Genetic_Algorithms>** and add unique features in the true category *<Theory>*.

Updated judgment basis:

<NewPattern>Genetic_Algorithms: If the paper focuses on optimization techniques inspired by evolutionary processes, such as population-based search, crossover, and mutation, classify it as <Genetic_Algorithms>.</NewPattern>
<NewPattern>Theory: If the paper emphasizes theoretical contributions to machine learning or AI, such as mathematical models, complexity theory, or the formal analysis of algorithms, or discusses optimality, domination, and bounded convergence rates in the context of repeated games, classify it as <Theory>.</NewPattern>

## E.3 ONE-SHOT W/ PRIOR SUMMARY + VGRL STEP 80

---

**one-shot predictor LLM CoT Step 80**

The feedback provided by the optimization model suggests that the individual categories of papers have their own unique characteristics.

<Case_Based>: Papers that discuss the use of past experiences, memory-based techniques, case retrieval and adaptation processes, or practical applications of machine learning techniques in continuous task domains should be categorized as <Case_Based>.
<Genetic_Algorithms>: Papers that include applications of genetic algorithms for evolutionary processes, adaptation methodologies, feature selection, optimization, adaptation in machine learning, evolutionary forces, mate preferences, learning-based plasticity, and the role of genetic and learned behaviors in understanding evolutionary change should be categorized as <Genetic_Algorithms>.
<Neural_Networks>: Papers that discuss neural network concepts, applications, calibration, divide-and-conquer approaches, memory-based techniques, genetic algorithms for optimizing neural networks, prototype styles of generalization, massively parallel architectures, approximation problems, probabilistic decision-making, confidence measures, pruning methods, stability and control in nonlinear systems, or practical applications, tool usage, or tutorials on how to use neural networks should be categorized as <Neural_Networks>.
<Probabilistic_Methods>: Papers that include discussions of stochastic processes, estimation with incomplete data, martingale theory, Hidden Markov Models, Dirichlet mixture densities, probabilistic inference and optimization as a function of problem structure, formal results from social choice theory, default reasoning, probabilistic methods in rational representation and belief formation, application of probabilistic methods in coding theory, including factor graphs and graphical models, and the application of probabilistic methods to neural networks should be categorized as <Probabilistic_Methods>.
<Reinforcement_Learning>: Papers that discuss adapting to new situations through dynamic decision-making, utilizing statistical learning methods for optimization in uncertain environments, applying adaptive mechanisms to solve complex routing problems, learning through trial and error, temporal difference learning, and model-based reinforcement learning techniques should be categorized as <Reinforcement_Learning>.
<Rule_Learning>: Papers that discuss adapting existing algorithms for specific tasks, improving or extracting decision rules, attribute estimation, learning from noisy data, non-myopic learning process, transforming regression algorithms into first-order learners for ordinal classification tasks, or improving classification accuracy and decision-making processes through data preprocessing methods should be categorized as <Rule_Learning>.
<Theory>: Papers that include discussions of stochastic processes, estimation with incomplete data, martingale theory, Hidden Markov Models, probabilistic inference and optimization as a function of problem structure, formal results from social choice theory, default reasoning, probabilistic methods in rational representation and belief formation, application of probabilistic methods in coding theory, including factor graphs and graphical models, and the application of probabilistic methods to neural networks should be categorized as <Theory>.

Judge from the information given which category the following essay belongs to.
I will provide you with the title, abstract, and citation summary of your paper and ask you to step-by-step determine if the paper fits into each category. First, please analyze whether the paper matches category <Case_Based>. If it does, please explain how it matches; if it doesn't, please point out where it doesn't. Then please compare the categories <Genetic_Algorithms> and vice versa. After comparing all the categories, please decide which category the paper fits better, and give a final classification decision, please enclose the final decision in <>.
Now analyze the information from the given paper:
============================
Here is a one-shot example; for more details, please refer to the appendix.
============================
Paper information:
(Paper Title: Cognitive Computation (Extended Abstract) )
(Abstract: Cognitive computation is discussed as a discipline that links together neurobiology, cognitive psychology and artificial intelligence.) (Neighbour Summary:
The papers cited in this essay discuss the concept of learning and reasoning, specifically focusing on the Probably Approximately Correct (PAC) learning model and its generalizations. The authors investigate the idea of agnostic learning, where minimal assumptions are made about the target function, and explore the possibilities of this approach through various results and algorithms. The papers also touch on the Learning to Reason framework, which combines learning and reasoning into a single task, and examine the tradeoff between learnability, oracle strength, and reasoning query range. Overall, the cited papers contribute to the understanding of learning and reasoning in artificial intelligence.)

Judgment:
The paper titled "Cognitive Computation (Extended Abstract)" discusses the discipline of cognitive computation, which links neurobiology, cognitive psychology, and artificial intelligence. However, the abstract does not explicitly discuss any of the topics mentioned in the Case-Based category, such as case retrieval, adaptation processes, or practical applications of machine learning techniques in continuous task domains.
The neighbour summary discusses the Probably Approximately Correct (PAC) learning model, agnostic learning, and the Learning to Reason framework, which are more closely related to the Probabilistic Methods category, as they involve probabilistic reasoning and learning.

Final decision:
<Probabilistic_Methods>

---

**one-shot summary LLM CoT Step 80**

Now that you have some descriptions, which will be used as rubrics for the classification task, you need to summarize and refine them for each category to find the best way to express them.
Here are the descriptions:

Judge from the information given which category the following essay belongs to.
<Case_Based>: If the paper discuss the use of past experiences, memory-based techniques, or case retrieval and adaptation processes to solve new problems, classify it as <Case_Based>. Additionally, if the paper focuses on combining case-based reasoning with other methodologies to improve adaptability and address challenges in dynamic problem-solving, classify it as <Case_Based>. If the paper discusses the use of past experiences, memory-based techniques, case retrieval and adaptation processes, or practical applications of machine learning techniques in continuous task domains, classify it as <Case_Based>. Additionally, include papers that focus on generating rules from data, organizing conceptual structures, and adapting existing knowledge to solve new problems.
<Genetic_Algorithm>: If the paper includes applications of genetic algorithms for evolutionary processes, adaptation methodologies, feature selection, optimization, adaptation in machine learning, evolutionary forces, mate preferences, learning-based plasticity, and the role of genetic and learned behaviors in understanding evolutionary change, classify it as <Genetic_Algorithm>.
<Neural_Networks>: Papers that discuss neural network concepts, applications, calibration, divide-and-conquer approaches, memory-based techniques, genetic algorithms for optimizing neural networks, prototype styles of generalization, massively parallel architectures, approximation problems, probabilistic decision-making, confidence measures, pruning methods, stability and control in nonlinear systems, or practical applications, tool usage, or tutorials on how to use neural networks should be categorized as <Neural_Networks>.
<Probabilistic_Methods>: If the paper includes discussions of stochastic processes, estimation with incomplete data, martingale theory, Hidden Markov Models, probabilistic inference and optimization as a function of problem structure, formal results from social choice theory, default reasoning, probabilistic methods in rational representation and belief formation, application of probabilistic methods in coding theory, including factor graphs and graphical models, and the application of probabilistic methods to neural networks, classify it as <Probabilistic_Methods>. However, exclude papers that discuss computation or artificial intelligence broadly, as these topics may not necessarily involve probabilistic methods. If the paper discusses discussions of stochastic processes, estimation with incomplete data, martingale theory, Hidden Markov Models, probabilistic inference and optimization as a function of problem structure, formal results from social choice theory, default reasoning, probabilistic methods in rational representation and belief formation, application of probabilistic methods in coding theory, including factor graphs and graphical models, and the application of probabilistic methods to neural networks, classify it as <Probabilistic_Methods>, excluding discussions of theoretical aspects such as model inaccuracies and deterministic processes.
<Reinforcement_Learning>: Papers that discuss adapting to new situations through dynamic decision-making, utilizing statistical learning methods for optimization in uncertain environments, applying adaptive mechanisms to solve complex routing problems, learning through trial and error, temporal difference learning, and model-based reinforcement learning techniques should be categorized as <Reinforcement_Learning>.
<Rule_Learning>: If the paper discusses adapting existing algorithms for specific tasks, improving or extracting decision rules, attribute estimation, learning from noisy data, non-myopic learning processes, transforming regression algorithms into first-order learners for ordinal classification tasks, or improving classification accuracy and decision-making processes through data preprocessing methods, classify it as <Rule_Learning>. However, exclude papers that focus on generating rules from data, organizing conceptual structures, and adapting existing knowledge to solve new problems. If the paper discusses adapting existing algorithms for specific tasks, improving or extracting decision rules, attribute estimation, learning from noisy data, non-myopic learning processes, transforming regression algorithms into first-order learners for ordinal classification tasks, or improving classification accuracy and decision-making processes through data preprocessing methods, developing new learning agents that adapt their knowledge representation spaces to better fit the learning task, classify it as <Rule_Learning>.
<Theory>: If the paper discusses theoretical aspects of machine learning, including cross-validation, modal theories, or other foundational concepts, classify it as <Theory>.

Note: Please ONLY summarize each category without omissions and NOTHING ELSE! And each category is output in the following format:
<Case_Based>: Papers that include [your summary] should be categorized as <Case_Based>.
<Genetic_Algorithms>: Papers that include [your summary] should be categorized as <Genetic_Algorithms>.
<Neural_Networks>: Papers that include [your summary] should be categorized as <Neural_Networks>.
<Probabilistic_Methods>: Papers that include [your summary] should be categorized as <Probabilistic_Methods>.
<Reinforcement_Learning>: Papers that include [your summary] should be categorized as <Reinforcement_Learning>.
<Rule_Learning>: Papers that include [your summary] should be categorized as <Rule_Learning>.
<Theory>: Papers that include [your summary] should be categorized as <Theory>.
Please begin with <START_SUMMARY>

<START_SUMMARY>
<Case_Based>: Papers that include discussions of past experiences, memory-based techniques, case retrieval and adaptation processes, combining case-based reasoning with other methodologies, generating rules from data, organizing conceptual structures, adapting existing knowledge to solve new problems, and practical applications of machine learning techniques in continuous task domains should be categorized as <Case_Based>.
<Genetic_Algorithms>: Papers that include application of genetic algorithms for evolutionary processes, adaptation methodologies, feature selection, optimization, adaptation in machine learning, evolutionary forces, mate preferences, learning-based plasticity, and the role of genetic and learned behaviors in understanding evolutionary change should be categorized as <Genetic_Algorithms>.
<Neural_Networks>: Papers that include discussions of neural network concepts, applications, calibration, divide-and-conquer approaches, memory-based techniques, genetic algorithms for optimizing neural networks, prototype styles of generalization, massively parallel architectures, approximation problems, probabilistic decision-making, confidence measures, pruning methods, stability and control in nonlinear systems, and practical applications, tool usage, or tutorials on how to use neural networks should be categorized as <Neural_Networks>.
<Probabilistic_Methods>: Papers that include discussions of stochastic processes, estimation with incomplete data, martingale theory, Hidden Markov Models, probabilistic inference and optimization as a function of problem structure, formal results from social choice theory, default reasoning, probabilistic methods in rational representation and belief formation, application of probabilistic methods in coding theory, including factor graphs and graphical models, and the application of probabilistic methods to neural networks should be categorized as <Probabilistic_Methods>.
<Reinforcement_Learning>: Papers that include discussions of adapting to new situations through dynamic decision-making, utilizing statistical learning methods for optimization in uncertain environments, applying adaptive mechanisms to solve complex routing problems, learning through trial and error, temporal difference learning, and model-based reinforcement learning techniques should be categorized as <Reinforcement_Learning>.
<Rule_Learning>: Papers that include discussions of adapting to new situations through dynamic decision-making, utilizing statistical learning methods for optimization in uncertain environments, applying adaptive mechanisms to solve complex routing problems, learning through trial and error, temporal difference learning, and model-based reinforcement learning techniques should be categorized as <Rule_Learning>.
<Theory>: Papers that include discussions of theoretical aspects of machine learning, including cross-validation, modal theories, and other foundational concepts, classify it as <Theory>.
<END_SUMMARY>

---

**one-shot optimizer LLM CoT Step 80**

The categorization model performed the judgment of paper categories by using the following features unique to each category of papers as the basis for the judgment:

<Case_Based>: Papers that discuss the use of past experiences, memory-based techniques, case retrieval and adaptation processes, or practical applications of machine learning techniques in continuous task domains should be categorized as <Case_Based>.
<Genetic_Algorithms>: Papers that include applications of genetic algorithms for evolutionary processes, adaptation methodologies, feature selection, optimization, adaptation in machine learning, evolutionary forces, mate preferences, learning-based plasticity, and the role of genetic and learned behaviors in understanding evolutionary change should be categorized as <Genetic_Algorithms>.
<Neural_Networks>: Papers that discuss neural network concepts, applications, calibration, divide-and-conquer approaches, memory-based techniques, genetic algorithms for optimizing neural networks, prototype styles of generalization, massively parallel architectures, approximation problems, probabilistic decision-making, confidence measures, pruning methods, stability and control in nonlinear systems, or practical applications, tool usage, or tutorials on how to use neural networks should be categorized as <Neural_Networks>.
<Probabilistic_Methods>: Papers that include discussions of stochastic processes, estimation with incomplete data, martingale theory, Hidden Markov Models, probabilistic inference and optimization as a function of problem structure, formal results from social choice theory, default reasoning, probabilistic methods in rational representation and belief formation, application of probabilistic methods in coding theory, including factor graphs and graphical models, and the application of probabilistic methods to neural networks should be categorized as <Probabilistic_Methods>.
<Reinforcement_Learning>: Papers that discuss adapting to new situations through dynamic decision-making, utilizing statistical learning methods for optimization in uncertain environments, applying adaptive mechanisms to solve complex routing problems, learning through trial and error, temporal difference learning, and model-based reinforcement learning techniques should be categorized as <Reinforcement_Learning>.
<Rule_Learning>: Papers that discuss adapting existing algorithms for specific tasks, improving or extracting decision rules, attribute estimation, learning from noisy data, non-myopic learning processes, transforming regression algorithms into first-order learners for ordinal classification tasks, or improving classification accuracy and decision-making processes through data preprocessing methods should be categorized as <Rule_Learning>.
<Theory>: Papers that include discussions of stochastic processes, estimation with incomplete data, martingale theory, Hidden Markov Models, probabilistic inference and optimization as a function of problem structure, formal results from social choice theory, default reasoning, probabilistic methods in rational representation and belief formation, application of probabilistic methods in coding theory, including factor graphs and graphical models.

Judge from the information given which category the following essay belongs to.
I will provide you with the title, abstract, citation summary information of your paper, the categories judged by the classification model, and the real categories of the paper, and ask you to verify that the basis of judgment for the corresponding categories is accurate.
First of all, please compare whether the categories provided by the model and the actual categories match. If it matches, you will directly analyze whether you need to add new judgment basis to the corresponding category based on the information and original judgment basis of the paper, if you need to add it, if not, you don't need to change anything; if it doesn't match, please analyze why the model may classify this paper into the category based on the original judgment basis corresponding to the incorrect category, and then please summarize the modifications and adjustments to the original judgment bases of the two categories, you can choose to remove matches for this paper in the wrong category, add unique features in the true category, or adjust the basis of judgment according to what you think is correct. Describe each step of the reasoning process in detail, and give the final updated basis of judgment, enclosing the final decision in <NewPattern></NewPattern>.
For example:
Here is a one-shot example; for more details, please refer to the appendix.
============================
Now analyze the given paper:
============================
Paper information:
(Paper Title: Cognitive Computation (Extended Abstract) )
(Abstract: Cognitive computation is discussed as a discipline that links together neurobiology, cognitive psychology and artificial intelligence. ) (Neighbour Summary:
The papers cited in this essay discuss the concept of learning and reasoning, specifically focusing on the Probably Approximately Correct (PAC) learning model and its generalizations. The authors investigate the idea of agnostic learning, where minimal assumptions are made about the target function, and explore the possibilities of this approach through various results and algorithms. The papers also touch on the Learning to Reason framework, which combines learning and reasoning into a single task, and examine the tradeoff between learnability, oracle strength, and reasoning query range. Overall, the cited papers contribute to the understanding of learning and reasoning in artificial intelligence.)
Model judgment category:
Probabilistic_Methods
Paper true category:
Neural_Networks

Step-by-Step Analysis:

- **Model Judgment Category**: *<Probabilistic_Methods>*
The model has incorrectly classified this paper under the *<Probabilistic_Methods>* category. We need to analyze why the model made this classification and adjust the judgment basis accordingly.
2. **Examination of the Paper Information (Title, Abstract, Neighbour Summary)**
- **Paper Title**: *Cognitive Computation (Extended Abstract)*
- The title suggests a broad focus on cognitive processes, which could encompass various disciplines including *Neural_Networks*.
- **Abstract**:
- The abstract discusses *cognitive computation* as a discipline linking neurobiology, cognitive psychology, and artificial intelligence. While it does briefly mention *Neural_Networks*, the broad scope of cognitive computation could include neural networks as a method or tool within artificial intelligence.
- **Neighbour Summary**:
- The neighbouring papers focus on *PAC learning*, *agnostic learning*, and the *Learning to Reason* framework. These topics are more aligned with *Probabilistic_Methods* than *Neural_Networks*, as they deal with probabilistic models and learning algorithms.
3. **Evaluation of the Existing Judgment Basis for *<Probabilistic_Methods>***
The current judgment basis for *<Probabilistic_Methods>* is as follows:
- *If the paper includes discussions of stochastic processes, estimation with incomplete data, martingale theory, Hidden Markov Models, probabilistic inference and optimization as a function of problem structure, formal results from social choice theory, default reasoning, probabilistic methods in rational representation and belief formation, application of probabilistic methods in coding theory, including factor graphs and graphical models, and the application of probabilistic methods to neural networks, classify it as <Probabilistic_Methods>.*
The paper does not fit this description well:
- **Stochastic processes**: The abstract does not mention stochastic processes.
- **Estimation with incomplete data**: There is no mention of estimation with incomplete data.
- **Martingale theory**: The abstract does not discuss martingale theory.
- **Hidden Markov Models**: The abstract does not mention Hidden Markov Models.
- **Probabilistic inference and optimization**: The abstract does not discuss probabilistic inference and optimization.
- **Formal results from social choice theory**: The abstract does not mention social choice theory.
- **Default reasoning**: The abstract does not discuss default reasoning.
- **Probabilistic methods in rational representation and belief formation**: The abstract does not mention probabilistic methods in rational representation and belief formation.
- **Application of probabilistic methods in coding theory**: The abstract does not discuss the application of probabilistic methods in coding theory.
- **Factor graphs and graphical models**: The abstract does not mention factor graphs or graphical models.
- **Application of probabilistic methods to neural networks**: The abstract does not discuss the application of probabilistic methods to neural networks.
4. **Determining the Need for Modifications**
The current judgment basis for *<Probabilistic_Methods>* is not well-aligned with the content of this paper. The paper does not discuss any of the topics listed in the judgment basis.
- **Remove matches for this paper in the wrong category**: The paper should not be classified under *<Probabilistic_Methods>*.
- **Add unique features in the true category**: The paper should be classified under *<Neural_Networks>* because it discusses cognitive computation, which could include neural networks as a method or tool within artificial intelligence.
- **Adjust the basis of judgment**: The judgment basis for *<Probabilistic_Methods>* should be adjusted to exclude papers that discuss cognitive computation or artificial intelligence broadly, as these topics may not necessarily involve probabilistic methods.

Updated judgment basis:

<NewPattern>Probabilistic_Methods: If the paper includes discussions of stochastic processes, estimation with incomplete data, martingale theory, Hidden Markov Models, probabilistic inference and optimization as a function of problem structure, formal results from social choice theory, default reasoning, probabilistic methods in rational representation and belief formation, application of probabilistic methods in coding theory, including factor graphs and graphical models, and the application of probabilistic methods to neural networks, classify it as <Probabilistic_Methods>. However, exclude papers that discuss cognitive computation or artificial intelligence broadly, as these topics may not necessarily involve probabilistic methods.</NewPattern>

# F ZERO-SHOT W/O PRIOR SUMMARY + VGRL

## F.1 ZERO-SHOT W/O PRIOR SUMMARY + VGRL STEP 1

---

**zero-shot predictor LLM CoT wo prior Step 1**

Inference Step
You will use the descriptions below to predict the output of the given information about a paper including: title, abstract and a summary of information about the papers cited in this paper. You need to tell me which of the following categories this paper belongs to:
<Case_Based>, <Genetic_Algorithms>, <Neural_Networks>, <Probabilistic_Methods>, <Reinforcement_Learning>, <Rule_Learning>, <Theory>
Here is the pattern descriptions and the information about the paper:
** Pattern Description: **
You will use the descriptions below.
Judge from the information given which category the following essay belongs to.
** Input: **
(Paper Title: Stochastic Propositionalization of Non-Determinate Background Knowledge )
(Abstract: It is a well-known fact that propositional learning algorithms require "good" features to perform well in practice. So a major step in data engineering for inductive learning is the construction of good features by domain experts. These features often represent properties of structured objects, where a property typically is the occurrence of a certain substructure having certain properties. To partly automate the process of "feature engineering", we devised an algorithm that searches for features which are defined by such substructures. The algorithm stochastically conducts a top-down search for first-order clauses, where each clause represents a binary feature. It differs from existing algorithms in that its search is not class-blind, and that it is capable of considering clauses ("context") of almost arbitrary length (size). Preliminary experiments are favorable, and support the view that this approach is promising.)
(Neighbour Summary:
The papers cited in this essay discuss various techniques and approaches in the field of Rule Learning, specifically in Inductive Logic Programming (ILP). The first paper re-appraises the development of techniques for inverting deduction and introduces Mode-Directed Inverse Entailment (MDIE) as a generalisation and enhancement of previous approaches. The second paper proposes an extension of the feature-vector representation to allow the value of a feature to be a set of strings, which can be efficiently and naturally represented in real-world learning problems. The third paper describes a system named SFOIL that uses a stochastic search method to alleviate the local optimization problem in greedy algorithms. Overall, these papers highlight the importance of efficient and effective techniques in Rule Learning and ILP.)
** Output Format: **
Please give your output strictly in the following format:

```
Explanations: [Your step-by-step analyses and results]
Output:
[Pick one in {<Case_Based>, <Genetic_Algorithms>, <Neural_Networks>, <Probabilistic_Methods>, <Reinforcement_Learning>, <Rule_Learning>, <Theory>}, e.g. <Case_Based>]
```

Please ONLY reply according to this format. Do not include any other words or step-by-step breakdowns.
Now take a deep breath and provide a concise explanation and output.assistant

Explanation: The paper discusses the stochastic propositionalization of non-determinate background knowledge, which is related to the construction of good features for inductive learning. The algorithm searches for features defined by substructures, and the cited papers focus on techniques and approaches in Rule Learning, specifically in Inductive Logic Programming (ILP). The use of stochastic search and the focus on Rule Learning suggest that the paper belong to the <Rule_Learning> category.
Output: <Rule_Learning>

---

**zero-shot summary LLM CoT wo prior Step 1**

Now that you have some descriptions, which will be used as rubrics for the classification task, you need to summarize and refine them for each category to find the best way to express them.
Here are the descriptions:

<Case_Based>: The papers that describe the application of memory-based techniques, case-based reasoning, or design systems in various domains, including structural design of buildings, should be categorized as <Case_Based>. The papers that discuss goal-driven learning, introspection, and the importance of explanation in artificial intelligence, particularly those that emphasize learning from specific cases and experiences, should be categorized as <Case_Based>.
<Genetic_Algorithms>: The papers that focus on genetic algorithms for solving large-scale optimization problems, improving solution quality, and exploring the effectiveness of various crossover operators should be categorized as <Genetic_Algorithms>.
<Neural_Networks>: The papers with a focus on neural networks, their applications, and theoretical perspectives should be categorized as <Neural_Networks>. The papers that discuss pruning methods for neural networks, including the adaptation of pruning strength to the evolution of weights and loss of generalization during training, should be categorized as <Neural_Networks>.
<Reinforcement_Learning>: The papers with a focus on learning, adaptation, and efficient algorithms for playing games against unknown adversaries should be categorized as <Reinforcement_Learning>.
<Rule_Learning>: The papers that discuss techniques and approaches in Inductive Logic Programming (ILP), such as searching for features defined by substructures, and considering clauses of almost arbitrary length, should be categorized as <Rule_Learning>.
<Theory>: The papers that focus on general theory and principles of artificial intelligence, without specific emphasis on goal-driven learning or explanation, should be categorized as <Theory>. The papers with a focus on theoretical concepts, mathematical analysis, and computational boundedness in game theory should be categorized as <Theory>. The papers that discuss game theory, introduce new classes of adversaries, and provide efficient algorithms for learning to play games against computationally bounded adversaries should be categorized as <Theory>.

Note: Please ONLY summarize each category without omissions and NOTHING ELSE! And each category is output in the following format:
"
<Case_Based>: Papers that include [your summary] should be categorized as <Case_Based>.
<Genetic_Algorithms>: Papers that include [your summary] should be categorized as <Genetic_Algorithms>.
<Neural_Networks>: Papers that include [your summary] should be categorized as <Neural_Networks>.
<Probabilistic_Methods>: Papers that include [your summary] should be categorized as <Probabilistic_Methods>.
<Reinforcement_Learning>: Papers that include [your summary] should be categorized as <Reinforcement_Learning>.
<Rule_Learning>: Papers that include [your summary] should be categorized as <Rule_Learning>.
<Theory>: Papers that include [your summary] should be categorized as <Theory>.
Please begin with <START_SUMMARY>

<START_SUMMARY>
<Case_Based>: Papers that describe memory-based techniques, case-based reasoning, or design systems in various domains, including structural design of buildings, or discuss goal-driven learning, introspection, and explanation in artificial intelligence, emphasizing learning from specific cases and experiences.
<Genetic_Algorithms>: Papers that focus on genetic algorithms for solving large-scale optimization problems, improving solution quality, and exploring crossover operators.
<Neural_Networks>: Papers that focus on neural networks, their applications, and theoretical perspectives, including pruning methods for neural networks.
<Probabilistic_Methods>: No description available.
<Reinforcement_Learning>: Papers that focus on learning, adaptation, and efficient algorithms for playing games against unknown adversaries.
<Rule_Learning>: Papers that discuss techniques and approaches in Inductive Logic Programming (ILP), such as searching for features defined by substructures and considering clauses of almost arbitrary length.
<Theory>: Papers that focus on general theories and principles of artificial intelligence, theoretical concepts, mathematical analysis, and computational boundedness in game theory, or discuss game theory, introduce new classes of adversaries, and provide algorithms for learning to play games against computationally bounded adversaries.

---

**zero-shot optimizer LLM CoT wo prior Step 1**

Optimization Step
Your goal is to learn the best descriptions for the model. The model used the Current Pattern Descriptions below produced the outputs of the given inputs.
** Current Pattern Descriptions: **
Please refer to the following judgement criteria to determine which category the following paper belongs to.
Judge from the information given which category the following essay belongs to.
You are provided with the characteristics of a node, another model's judgment of its category, and its true category. Please optimize the Pattern Descriptions for better prediction.
** Inputs (a paper): **
(Paper Title: Stochastic Propositionalization of Non-Determinate Background Knowledge )
(Abstract: It is a well-known fact that propositional learning algorithms require "good" features to perform well in practice. So a major step in data engineering for inductive learning is the construction of good features by domain experts. These features often represent properties of structured objects, where a property typically is the occurrence of a certain substructure having certain properties. To partly automate the process of "feature engineering", we devised an algorithm that searches for features which are defined by such substructures. The algorithm stochastically conducts a top-down search for first-order clauses, where each clause represents a binary feature. It differs from existing algorithms in that its search is not class-blind, and that it is capable of considering clauses ("context") of almost arbitrary length (size). Preliminary experiments are favorable, and support the view that this approach is promising.)
(Neighbour Summary:
The papers cited in this essay discuss various techniques and approaches in the field of Rule Learning, specifically in Inductive Logic Programming (ILP). The first paper re-appraises the development of techniques for inverting deduction and introduces Mode-Directed Inverse Entailment (MDIE) as a generalisation and enhancement of previous approaches. The second paper proposes an extension of the feature-vector representation to allow the value of a feature to be a set of strings, which can be efficiently and naturally represented in real-world learning problems. The third paper describes a system named SFOIL that uses a stochastic search method to alleviate the local optimization problem in greedy algorithms. Overall, these papers highlight the importance of efficient and effective techniques in Rule Learning and ILP.)
** The model outputs: **
Rule_Learning
** The target outputs: **
Rule_Learning
If the model output differs from the target output, update the descriptions for both the model's output category and the target output category to reduce classification error on both current and future data. If the model and target outputs are the same, update the description for this category only. Limit your 'New Pattern Descriptions' to less than 150 words per category. Think step by step provide your output strictly in the following format:
** Output Format: **
Please think step by step and give your outputs strictly in the following format:
"
Reasoning:
[Provide a concise summary of the reasoning process, focusing on the key insights that lead to the final decision. Avoid listing multiple steps. Ensure the reasoning is clear and justifies the new pattern descriptions.]
New Pattern Descriptions:
[Please insert your new descriptions here. The new descriptions MUST BE DIRECTLY based on the key features and insights identified in the input paper. Update the descriptions ONLY for the categories where the model output differs from the target output, or if they match, update ONLY that category. Ensure that only the following categories are used: <Case_Based>, <Genetic_Algorithms>, <Neural_Networks>, <Probabilistic_Methods>, <Reinforcement_Learning>, <Rule_Learning>, <Theory>. Each updated description MUST BE enclosed within <NewPattern></NewPattern> tags. Only include descriptions for the categories being updated.]
For example:
<NewPattern><Case_Based>: The papers with [your new pattern description] should be categorized as <Case_Based>.</NewPattern>
<NewPattern><Genetic_Algorithms>: The papers that include [your new pattern description] should be categorized as <Genetic_Algorithms>.</NewPattern>
"
Please ONLY reply according to this format, DON'T give me any other words. If the model output and target output are the same, update ONLY that category and NO OTHERS. Now take a deep breath and provide a concise and logical Reasoning and New Pattern Descriptions.assistant

Reasoning:
The paper discusses the stochastic propositionalization of non-determinate background knowledge, which is a technique used in Inductive Logic Programming (ILP). The algorithm searches for features defined by substructures, and it is capable of considering clauses of almost arbitrary length. The papers cited in the neighbour summary discuss various techniques and approaches in the field of Rule Learning, specifically in ILP. The key insight is that the paper is related to Rule Learning, and the neighbour summary provides additional context about the field of ILP. However, the model output and target output are the same, which means we only need to update the description for the Rule_Learning category.
New Pattern Descriptions:
<NewPattern><Rule_Learning>: The papers that discuss techniques and approaches in Inductive Logic Programming (ILP), such as searching for features defined by substructures, and considering clauses of almost arbitrary length, should be categorized as <Rule_Learning>.</NewPattern>

---

## F.2 ZERO-SHOT W/O PRIOR SUMMARY + VGRL STEP 2

**zero-shot predictor LLM CoT wo prior Step 2**

Inference Step
You will use the descriptions below to predict the output of the given information about a paper including: title, abstract and a summary of information about the papers cited in this paper. You need to tell which of the following categories this paper belongs to:
<Case_Based>, <Genetic_Algorithms>, <Neural_Networks>, <Probabilistic_Methods>, <Reinforcement_Learning>, <Rule_Learning>, <Theory>.
Here is the pattern descriptions and the information about the paper:
** Pattern Descriptions: **
You will use the description below.

<Case_Based>: Papers that describe memory-based technique, case-based reasoning, or design systems in various domains, including structural design of buildings, or discuss goal-driven learning, introspection, and explanation in artificial intelligence, emphasizing learning from specific cases and experiences.
<Genetic_Algorithms>: Papers that focus on genetic algorithms for solving large-scale optimization problems, improving solution quality, and exploring crossover operators.
<Neural_Networks>: Papers that focus on neural networks, their applications, and theoretical perspectives, including pruning methods for neural networks.
<Probabilistic_Methods>: No description available.
<Reinforcement_Learning>: Papers that focus on learning, adaptation, and efficient algorithms for playing games against unknown adversaries.
<Rule_Learning>: Papers that discuss technique and approaches in Inductive Logic Programming (ILP), such as searching for features defined by substructures and considering clauses of almost arbitrary length.
<Theory>: Papers that focus on general theories and principles of artificial intelligence, theoretical concepts, mathematical analysis, and computational boundedness in game theory, or discuss game theory, introduce new classes of adversaries, and provide efficient algorithms for learning to play games against computationally bounded adversaries.

** Input: **
(Paper Title: Learning from positive data )
(Abstract: Gold showed in 1967 that not even regular grammars can be exactly identified from positive examples alone. Since it is known that children learn natural grammars almost exclusively from positives examples, Gold's result has been used as a theoretical support for Chomsky's theory of innate human linguistic abilities. In this paper new results are presented which show that within a Bayesian framework not only grammars, but also logic programs are learnable with arbitrarily low expected error from positive examples only. In addition, we show that the upper bound for expected error of a learner which maximises the Bayes' posterior probability when learning from positive examples is within a small additive term of one which does the same from a mixture of positive and negative examples. An Inductive Logic Programming implementation is described which avoids the pitfalls of greedy search by global optimisation of this function during the local construction of individual clauses of the hypothesis. Results of testing this implementation on artificially-generated data-sets are reported. These results are in agreement with the theoretical predictions. )
(Neighbour Summary:
The papers cited in this essay discuss the limitations of current inductive logic programming systems in handling noise and learning recursive predicates. A new framework for learning from noisy data and fixed example size is presented, which incorporates a Bayesian heuristic for finding the most probable hypothesis. This approach is implemented in an ILP system called Lime, which is shown to handle noise better than other systems like FOIL and PROGOL, and is capable of learning from only positive or negative data.
Common points that reflect the category of this paper include:
- Handling noise in inductive logic programming systems
- Learning recursive predicates
- Improving the performance of ILP systems
- Developing new framework for learning from noisy data.)
** Output Format: **
Please give your output strictly in the following format:
''
Explanations: [Your step-by-step analyses and results]
Output:
[Pick one in (<Case_Based>, <Genetic_Algorithms>, <Neural_Network>, <Probabilistic_Methods>, <Reinforcement_Learning>, <Rule_Learning>, <Theory>)], e.g. <Case_Based>
''
Please ONLY reply according to this format. Do not include any other words or step-by-step breakdowns.
Now take a deep breath and provide a concise explanation and output.assistant

Explanations: The paper discusses learning from positive data, specifically within a Bayesian framework, and presents a new approach for learning logic programs from positive examples only. It also compares the performance of its implementation, Lime, with other ILP systems like FOIL and PROGOL. The paper focus on improving the performance of ILP systems and developing new frameworks for learning from noisy data, which involves learning recursive predicates and handling noise in inductive logic programming systems.
Output: <Rule_Learning>

**zero-shot summary LLM CoT wo prior Step 2**

Now that you have some descriptions, which will be used as rubrics for the classification task, you need to summarize and refine them for each category to find the best way to express them.
Here are the descriptions:

<Case_Based>: Papers that describe memory-based technique, case-based reasoning, or design systems in various domains, including structural design of buildings, or discuss goal-driven learning, introspection, and explanation in artificial intelligence, emphasizing learning from specific cases and experiences. The papers that focus on case-based reasoning, precedents, and analogy in legal decision-making, proposing models for reasoning with precedents and discussing the challenge of matching specific case facts with open-textured terms, should be categorized as <Case_Based>. The papers that describe learning from specific cases and experiences, such as learning default concepts from observational data, and focus on classifying incompletely described objects, should be categorized as <Case_Based>. The papers that describe memory-based techniques for planning, such as Case-Based Planning (CBP), and focus on retrieving and adapting previous planning experiences to solve complex problems, should be categorized as <Case_Based>.
<Genetic_Algorithms>: Papers that focus on genetic algorithms for solving large-scale optimization problems, improving solution quality, and exploring crossover operators.
<Neural_Networks>: Papers that focus on neural networks, their applications, and theoretical perspectives, including pruning methods for neural networks.
<Probabilistic_Methods>: No description available.
<Reinforcement_Learning>: Papers that focus on learning, adaptation, and efficient algorithms for playing games against unknown adversaries.
<Rule_Learning>: Papers that discuss technique and approaches in Inductive Logic Programming (ILP), such as searching for features defined by substructures and considering clauses of almost arbitrary length. The papers on inductive logic programming, learning recursive predicates, and improving the performance of ILP systems should be categorized as <Rule_Learning>.
<Theory>: Papers that focus on general theories and principles of artificial intelligence, theoretical concepts, mathematical analysis, and computational boundedness in game theory, or discuss game theory, introduce new classes of adversaries, and provide efficient algorithms for learning to play games against computationally bounded adversaries. The papers that discuss the development of new frameworks for learning from noisy data, improving the performance of ILP systems, and handling noise in inductive logic programming should be categorized as <Theory>. The papers which focus on practical applications of machine learning techniques, such as feature construction, selection, and subset selection, and use of stochastic techniques, greedy algorithms, and statistical measures should be categorized as <Theory>. The papers that discuss theoretical aspects of machine learning, such as the limitations of classical concepts, probabilistic processes, and learning techniques for developing accurate default concepts, should be categorized as <Theory>. The papers that focus on theoretical aspects of machine learning, including learning from examples, noisy data, and uncertain or probabilistic behavior, should be categorized as <Theory>. The papers that discuss theoretical frameworks, probabilistic processes, and learning techniques for addressing complex problems, such as classifying incompletely described objects, should be categorized as <Theory>. The papers that discuss general theories and principles of artificial intelligence, such as explanation-based learning and derivational analogy (DA), and explore their applications in planning and problem-solving, should be categorized as <Theory>. The papers that discuss theoretical aspects of machine learning, such as the complexity of learning in various models, the importance of representation in inductive learning systems, and the application of multiplicative weight-update rules in on-line prediction models, should be categorized as <Theory>.
Note: Please ONLY summarize each category without omissions and NOTHING ELSE! And each category is output in the following format:
<Case_Based>: Papers that include [your summary] should be categorized as <Case_Based>.
<Genetic_Algorithms>: Papers that include [your summary] should be categorized as <Genetic_Algorithms>.
<Neural_Networks>: Papers that include [your summary] should be categorized as <Neural_Networks>.
<Probabilistic_Methods>: Papers that include [your summary] should be categorized as <Probabilistic_Methods>.
<Reinforcement_Learning>: Papers that include [your summary] should be categorized as <Reinforcement_Learning>.
<Rule_Learning>: Papers that include [your summary] should be categorized as <Rule_Learning>.
<Theory>: Papers that include [your summary] should be categorized as <Theory>.
Please begin with <START_SUMMARY>

<START_SUMMARY>
<Case_Based>: Papers that describe memory-based technique, case-based reasoning, or design systems in various domains, including structural design of buildings, or discuss goal-driven learning, introspection, and explanation in artificial intelligence, emphasizing learning from specific cases and experiences, case-based reasoning, precedents, analogy in legal decision-making, learning from specific cases and experiences, classifying incompletely described objects, learning default concepts from observational data, memory-based techniques for planning, Case-Based Planning (CBP), retrieving and adapting previous planning experience to solve complex problems.
<Genetic_Algorithms>: Papers that focus on genetic algorithms for solving large-scale optimization problems, improving solution quality, and exploring crossover operators.
<Neural_Networks>: Papers that focus on neural networks, their applications, and theoretical perspectives, including pruning methods for neural networks.
<Probabilistic_Methods>: No description available.
<Reinforcement_Learning>: Papers that focus on learning, adaptation, and efficient algorithms for playing games against unknown adversaries.
<Rule_Learning>: Papers that discuss technique and approaches in Inductive Logic Programming (ILP), such as searching for features defined by substructures, considering clauses of almost arbitrary length, learning recursive predicates, improving the performance of ILP systems.
<Theory>: Papers that focus on general theories and principles of artificial intelligence, theoretical concepts, mathematical analysis, and computational boundedness in game theory, or discuss game theory, introducing new classes of adversaries, providing efficient algorithms for learning to play games against computationally bounded adversaries, developing new frameworks for learning from noisy data, improving the performance of ILP systems, handling noise in inductive logic programming systems, practical applications of machine learning techniques, feature construction, selection, and subset selection, use of stochastic techniques, greedy algorithms, statistical measures, theoretical aspects of machine learning, limitations of classical concepts, probabilistic processes, learning techniques for developing accurate default concepts, learning from examples, noisy data, uncertain or probabilistic behavior, theoretical frameworks, probabilistic processes, learning techniques for addressing complex problems, explanation-based learning, derivational analogy (DA), applications in planning and problem-solving, complexity of learning in various models, importance of representation in inductive learning systems, application of multiplicative weight-update rules in on-line prediction models.

**zero-shot optimizer LLM CoT wo prior Step 2**

Optimization Step
Your goal is to learn the best descriptions for the model. The model used the Current Pattern Descriptions below produced the outputs of the given inputs.
** Current Pattern Descriptions: **
Please refer to the following judgement criteria to determine which category the following paper belongs to.

<Case_Based>: Papers that describe memory-based techniques, case-based reasoning, or design systems in various domains, including structural design of buildings, or discuss goal-driven learning, introspection, and explanation in artificial intelligence, emphasizing learning from specific cases and experiences.
<Genetic_Algorithms>: Papers that focus on genetic algorithms for solving large-scale optimization problems, improving solution quality, and exploring crossover operators.
<Neural_Networks>: Papers that focus on neural networks, their applications, and theoretical perspectives, including pruning methods for neural networks.
<Probabilistic_Methods>: No description available.
<Reinforcement_Learning>: Papers that focus on learning, adaptation, and efficient algorithms for playing games against unknown adversaries.
<Rule_Learning>: Papers that discuss techniques and approaches in Inductive Logic Programming (ILP), such as searching for features defined by substructures and considering clauses of almost arbitrary length.
<Theory>: Papers that focus on general theories and principles of artificial intelligence, theoretical concepts, mathematical analysis, and computational boundedness in game theory, or discuss game theory, introduce new classes of adversaries, and provide efficient algorithms for learning to play games against computationally bounded adversaries.

You are provided with the characteristics of a node, another model's judgment of its category, and its true category. Please optimize the Pattern Descriptions for better prediction.
** Inputs (a paper): **
(Paper Title: Learning from positive data )
(Abstract: Gold showed in 1967 that not even regular grammars can be exactly identified from positive examples alone. Since it is known that children learn natural grammars almost exclusively from positives examples, Gold's result has been used as a theoretical support for Chomsky's theory of innate human linguistic abilities. In this paper new results are presented which show that within a Bayesian framework not only grammars, but also logic programs are learnable with arbitrarily low expected error from positive examples only. In addition, we show that the upper bound for expected error of a learner which maximises the Bayes' posterior probability when learning from positive examples is within a small additive term of one which does the same from a mixture of positive and negative examples. An Inductive Logic Programming implementation is described which avoids the pitfalls of greedy search by global optimisation of this function during the local construction of individual clauses of the hypothesis. Results of testing this implementation on artificially-generated data-sets are reported. These results are in agreement with the theoretical predictions. )
(Neighbour Summary:
The papers cited in this essay discuss the limitations of current inductive logic programming systems in handling noise and learning recursive predicates. A new framework for learning from noisy data and fixed example size is presented, which incorporates a Bayesian heuristic for finding the most probable hypothesis. This approach is implemented in an ILP system called Lime, which is shown to handle noise better than other systems like FOIL and PROGOL, and is capable of learning from only positive or negative data.
Common points that reflect the category of this paper include:
- Handling noise in inductive logic programming systems
- Learning recursive predicates
- Improving the performance of ILP systems
- Developing new frameworks for learning from noisy data.)
** The model outputs: **
Rule_Learning
** The target outputs: **
Theory
If the model output differs from the target output, update the descriptions for both the model's output category and the target output category to reduce classification error on both current and future data. If the model and target outputs are the same, update the description for this category only. Limit your 'New Pattern Descriptions' to less than 150 words per category. Think step by step provide your output strictly in the following format:
** Output Format: **
Please think step by step and give your outputs strictly in the following format:
''
Reasoning:
[Provide a concise summary of the reasoning process, focusing on the key insights that lead to the final decision. Avoid listing multiple steps. Ensure the reasoning is clear and justifies the new pattern descriptions.]
New Pattern Descriptions:
[Please insert your new descriptions here. The new descriptions MUST BE DIRECTLY based on the key features and insights identified in the input paper. Update the descriptions ONLY for the categories where the model output differs from the target output, or if they match, update ONLY that category. Ensure that only the following categories are used: <Case_Based>, <Genetic_Algorithms>, <Neural_Networks>, <Probabilistic_Methods>, <Reinforcement_Learning>, <Rule_Learning>, <Theory>. Each updated description MUST BE enclosed within <NewPattern></NewPattern> tags. Only include descriptions for the categories being updated. If the model output and target output are the same, update ONLY that category and NO OTHERS.]
For example:
<NewPattern><Case_Based>: The papers with [your new pattern description] should be categorized as <Case_Based>.</NewPattern>
<NewPattern><Genetic_Algorithms>: The papers with [your new pattern description] should be categorized as <Genetic_Algorithms>.</NewPattern>
''
Please ONLY reply according to this format, DON'T give me any other words. If the model output and target output are the same, update ONLY that category and NO OTHERS. Now take a deep breath and provide a concise and logical Reasoning and New Pattern Descriptions.assistant

Reasoning:
The paper discusses the limitations of current inductive logic programming systems in handling noise and learning recursive predicates, and presents a new framework for learning from noisy data and fixed example size. The paper also describes an ILP system called Lime, which is shown to handle noise better than other systems like FOIL and PROGOL, and is capable of learning from only positive or negative data. The key features of this paper include the development of new frameworks for learning from noisy data, improving the performance of ILP systems, and handling noise in inductive logic programming systems. These features are more closely related to theoretical concepts and mathematical analysis, which are characteristic of the <Theory> category.
New Pattern Descriptions:
<NewPattern><Theory>: The papers that discuss the development of new frameworks for learning from noisy data, improving the performance of ILP systems, and handling noise in inductive logic programming systems should be categorized as <Theory>.</NewPattern>
<NewPattern><Rule_Learning>: The papers that focus on inductive logic programming, learning recursive predicates, and improving the performance of ILP systems should be categorized as <Rule_Learning>.</NewPattern>

## F.3 ZERO-SHOT W/O PRIOR SUMMARY + VGRL STEP 80

**zero-shot predictor LLM CoT wo prior Step 80**

Inference Step
You will use the descriptions below to predict the output of the given information about a paper including: title, abstract and a summary of information about the papers cited in this paper. You need to tell me which of the following categories this paper belongs to:
<Case_Based>, <Genetic_Algorithms>, <Neural_Networks>, <Probabilistic_Methods>, <Reinforcement_Learning>, <Rule_Learning>, <Theory>.
Here is the pattern descriptions and the information about the paper:
** Pattern Descriptions: ** You will use the descriptions below.

<Case_Based>: Papers that include memory-based techniques, case-based reasoning, learning from specific cases and experience, case adaptation, similarity assessment, adaptation knowledge, planning and problem-solving systems, diagnosis, learning from past experiences, meta-cases, task-method-knowledge models, problem-solving episodes, learning natural language grammars, robotic control systems, machine learning techniques, feature selection, goal handling processes, invention goals, working memory, design configuration, evolutionary processes, place recognition, case-based reasoning in robotic navigation systems, continuous task domains, and continuous operators should be categorized as <Case_Based>.
<Genetic_Algorithms>: Papers that include genetic algorithms for solving large-scale optimization problems, improving solution quality, exploring crossover operators, artificial life research, evolutionary change through learning-based plasticity, environment structure and organism behavior in artificial evolution, iterative processes, real-time interaction, adaptation to novel situations, genetic algorithms for feature selection, machine learning applications, comparisons with other search methods, applications in robotics and control systems, program derivation, clique detection in networks, collective action and memory in agent societies, supervised classification problems, solving control problems, the Baldwin Effect, cultural exploitation in neuro-evolution, cultural transmission of information, self-optimization, online evolution, and real-time interaction should be categorized as <Genetic_Algorithms>.
<Neural_Networks>: Papers that include neural networks, pruning methods, real-time control, self-organization of feedforward synapses, stability and control in nonlinear systems, robust control analysis and design, generalization of classical theorems, application to input-state stability, neural network architectures, training, generative models, PCA, EM algorithm, radial basis functions, approximation orders, finite-state automata, string classification, fault-tolerant neural DFA implementations, interaction between neural networks and other brain regions, modeling complex relationships between inputs and outputs, regression and classification tasks, Gaussian processes, coding theory, control-Lyapunov functions, and control systems should be categorized as <Neural_Networks>.
<Probabilistic_Methods>: Papers that include probabilistic decision trees, learning probabilities for each subtree, and providing confidence measures should be categorized as <Probabilistic_Methods>.
<Reinforcement_Learning>: Papers that include learning from specific cases and experience, value function approximation, and the use of neural networks for robust and efficient learning methods in reinforcement learning should be categorized as <Reinforcement_Learning>. Papers that include learning from specific cases and experience, value function approximation, the use of... (too long)

** Input: ** (Paper Title: Cognitive Computation (Extended Abstract) )
(Abstract: Cognitive computation is discussed as a discipline that links together neurobiology, cognitive psychology and artificial intelligence. )
(Neighbour Summary: The papers cited in this essay discuss the concept of learning and reasoning. The authors investigate the idea of agnostic learning, where minimal assumptions are made about the target function, and explore the possibilities of this approach through various results and algorithms. The papers also touch on the Learning to Reason framework, which combines learning and reasoning into a single task, and examine the tradeoff between learnability, oracle strength, and reasoning query range. Overall, the cited papers contribute to the understanding of learning and reasoning in artificial intelligence.)
** Output Format: ** Please give your output strictly in the following format:
'' Explanation: [Your step-by-step analyses and results]
Output: [Pick one in (<Case_Based>, <Genetic_Algorithms>, <Neural_Networks>, <Probabilistic_Methods>, <Reinforcement_Learning>, <Rule_Learning>, <Theory>)], e.g. <Case_Based> ''
Please ONLY reply according to this format. Do not include any other words or step-by-step breakdowns.
Now take a deep breath and provide a concise explanation and output.assistant

Explanation: The paper discusses cognitive computation as a discipline that links together neurobiology, cognitive psychology, and artificial intelligence. The neighbour summary mentions the Probably Approximately Correct (PAC) learning model, agnostic learning, and the Learning to Reason framework, which are related to probabilistic methods and learning from specific cases and experience. However, none of the specific techniques mentioned in the neighbour summary match the descriptions of <Genetic_Algorithms>, <Neural_Networks>, <Reinforcement_Learning>, <Rule_Learning>, or <Theory>. The neighbour summary does not mention memory-based techniques, case-based reasoning, genetic algorithms, neural networks, or other techniques that are characteristic of the other categories.
Output: <Theory>

**zero-shot summary LLM CoT wo prior Step 80**

Now that you have some descriptions, which will be used as rubrics for the classification task, you need to summarize and refine them for each category to find the best way to express them.
Here is the descriptions:

<Case_Based>: Papers that include memory-based techniques, case-based reasoning, learning from specific cases and experience, case adaptation, similarity assessment, adaptation knowledge, planning and problem-solving systems, diagnosis, learning from past experiences, meta-cases, task-method-knowledge models, problem-solving episodes, learning natural language grammars, robotic control systems, machine learning techniques, feature selection, goal handling processes, invention goals, working memory, design configuration, evolutionary processes, place recognition, case-based reasoning in robotic navigation systems, constraint reasoning technology, and skeletal planning methods for addressing complex domains, such as forest fires, should be categorized as <Case_Based>. The papers that include interactive planning systems, case-based reasoning from specific cases and experiences, case adaptation, similarity assessment, adaptation knowledge, planning and problem-solving systems, diagnosis, learning from past experiences, meta-cases, task-method-knowledge models, problem-solving episodes, learning natural language grammars, robotic control systems, machine learning techniques, feature selection, goal handling processes, invention goals, working memory, design configuration, evolutionary processes, place recognition, case-based reasoning in robotic navigation systems, continuous task domains, and continuous operators, as well as combining these techniques with other reasoning modes, should be categorized as <Case_Based>. The papers that include learning from specific cases and experience, case-based reasoning, classification methodologies, inductive learning, and knowledge discovery should be categorized as <Case_Based>.
<Genetic_Algorithms>: Papers that include genetic algorithms for solving large-scale optimization problems, improving solution quality, exploring crossover operators, artificial life research, evolutionary change through learning-based plasticity, environment structure and organism behavior in artificial evolution, iterative processes, real-time interaction, adaptation to novel situations, genetic algorithms for feature selection, machine learning applications, comparisons with other search methods, applications in robotics and control systems, program derivation, clique detection in networks, collective action and memory in agent societies, supervised classification problems, solving control problems, the Baldwin Effect, cultural exploitation in neuro-evolution, cultural transmission of information, self-optimization, online evolution, and real-time interaction should be categorized as <Genetic_Algorithms>. The papers that include evolutionary learning, crossover operators, and the application of neural networks in genetic algorithms should be categorized as <Genetic_Algorithms>. The papers that include genetic algorithms for solving large-scale optimization problems, improving solution quality, exploring crossover operators, artificial life research, evolutionary change through learning-based plasticity, environment structure and organism behavior in artificial evolution, iterative processes, real-time interaction, adaptation to novel situations, genetic algorithms for feature selection, machine learning applications, comparisons with other search methods, applications in robotics and control systems, program derivation, clique detection in networks, collective action and memory in agent societies, supervised classification problems, solving control problems, the Baldwin Effect, cultural exploitation in neuro-evolution, cultural transmission of information, self-optimization, online evolution, and real-time interaction, as well as using genetic algorithms to improve case adaptability in case-based reasoning, should be categorized as <Genetic_Algorithms>.
<Neural_Networks>: Papers that include neural networks, pruning methods, real-time control, self-organization of feedforward synapses, stability and control in nonlinear systems, robust control analysis and design, generalization of classical theorems, application to input-state stability, neural network architectures, training, generative models, PCA, EM algorithm, radial basis functions, approximation orders, finite-state automata, string classification, fault-tolerant neural DFA implementations, interaction between neural networks and other brain regions, modeling complex relationships between inputs and outputs, regression and classification tasks, Gaussian processes, coding theory, control-Lyapunov functions, and control systems should be categorized as <Neural_Networks>. The papers that include probabilistic decision trees, learning probabilities for each subtree, and providing confidence measures should be categorized as <Probabilistic_Methods>. Papers that focus on theoretical aspects of classification, prediction error estimation, and the comparison of different methods for estimating error rates should be categorized as <Probabilistic_Methods>.
<Reinforcement_Learning>: Papers that include learning from specific cases and experience, value function approximation, and the use of neural networks for robust and... (too long)

Note: Please ONLY summarize each category without omissions and NOTHING ELSE! And each category is output in the following format:
<Case_Based>: Papers that include [your summary] should be categorized as <Case_Based>.
<Genetic_Algorithms>: Papers that include [your summary] should be categorized as <Genetic_Algorithms>.
<Neural_Networks>: Papers that include [your summary] should be categorized as <Neural_Networks>.
<Probabilistic_Methods>: Papers that include [your summary] should be categorized as <Probabilistic_Methods>.
<Reinforcement_Learning>: Papers that include [your summary] should be categorized as <Reinforcement_Learning>.
<Rule_Learning>: Papers that include [your summary] should be categorized as <Rule_Learning>.
<Theory>: Papers that include [your summary] should be categorized as <Theory>.
Please begin with <START_SUMMARY>

<START_SUMMARY>
<Case_Based>: Papers that include memory-based techniques, case-based reasoning, learning from specific cases and experience, case adaptation, similarity assessment, adaptation knowledge, planning and problem-solving systems, diagnosis, learning from past experiences, meta-cases, task-method-knowledge models, problem-solving episodes, learning natural language grammars, robotic control systems, machine learning techniques, feature selection, goal handling processes, invention goals, working memory, design configuration, evolutionary processes, place recognition, case-based reasoning in robotic navigation systems, continuous task domains, and continuous operators should be categorized as <Case_Based>.
<Genetic_Algorithms>: Papers that include genetic algorithms for solving large-scale optimization problems, improving solution quality, exploring crossover operators, artificial life research, evolutionary change through learning-based plasticity, environment structure and organism behavior in artificial evolution, iterative processes, real-time interaction, adaptation to novel situations, genetic algorithms for feature selection, machine learning applications, comparisons with other search methods, applications in robotics and control systems, program derivation, clique detection in networks, collective action and memory in agent societies, supervised classification problems, solving control problems, the Baldwin Effect, cultural exploitation in neuro-evolution, cultural transmission of information, self-optimization, online evolution, and real-time interaction should be categorized as <Genetic_Algorithms>.
<Neural_Networks>: Papers that include neural networks, pruning methods, real-time control, self-organization of feedforward synapses, stability and control in nonlinear systems, robust control analysis and design, generalization of classical theorems, application to input-state stability, neural network architectures, training, generative models, PCA, EM algorithm, radial basis functions, approximation orders, finite-state automata, string classification, fault-tolerant neural DFA implementations, interaction between neural networks and other brain regions, modeling complex relationships between inputs and outputs, regression and classification tasks, Gaussian processes, coding theory, control-Lyapunov functions, and control systems should be categorized as <Neural_Networks>.
<Probabilistic_Methods>: Papers that include probabilistic decision trees, learning probabilities for each subtree, and providing confidence measures should be categorized as <Probabilistic_Methods>. Papers that focus on theoretical aspects of classification, prediction error estimation, and the comparison of different methods for estimating error rates should be categorized as <Probabilistic_Methods>.
<Reinforcement_Learning>: Papers that include learning from specific cases and experience, value function approximation, and the use of neural networks for robust and... (too long)

**zero-shot optimizer LLM CoT wo prior Step 80**

Optimization Step
Your goal is to learn the best descriptions for the model. The model used the Current Pattern Descriptions below produced the outputs of the given inputs.
** Current Pattern Descriptions: **
Please refer to the following judgement criteria to determine which category the following paper belongs to.

<Case_Based>: Papers that include memory-based techniques, case-based reasoning, learning from specific cases and experiences, case adaptation, similarity assessment, adaptation knowledge, planning and problem-solving systems, diagnosis, learning from past experiences, meta-cases, task-method-knowledge models, problem-solving episodes, learning natural language grammars, robotic control systems, machine learning techniques, feature selection, goal handling processes, invention goals, working memory, design configuration, evolutionary processes, place recognition, case-based reasoning in robotic navigation systems, continuous task domains, and continuous operators should be categorized as <Case_Based>.
<Genetic_Algorithms>: Papers that include genetic algorithms for solving large-scale optimization problems, improving solution quality, exploring crossover operators, artificial life research, evolutionary change through learning-based plasticity, environment structure and organism behavior in artificial evolution, iterative processes, real-time interaction, adaptation to novel situations, genetic algorithms for feature selection, machine learning applications, comparisons with other search methods, applications in robotics and control systems, program derivation, clique detection in networks, collective action and memory in agent societies, supervised classification problems, solving control problems, the Baldwin Effect, cultural exploitation in neuro-evolution, online evolution, and real-time interaction should be categorized as <Genetic_Algorithms>.
<Neural_Networks>: Papers that include neural networks, pruning methods, real-time control, self-organization of feedforward synapses, stability and control in nonlinear systems, robust control analysis and design, generalization of classical theorems, application to input-state stability, neural network architectures, training, generative models, PCA, EM algorithm, radial basis functions, approximation orders, finite-state automata, string classification, fault-tolerant neural DFA implementations, interaction between neural networks and other brain regions, modeling complex relationships between inputs and outputs, regression and classification tasks, Gaussian processes, coding theory, control-Lyapunov functions, and control systems should be categorized as <Neural_Networks>.
<Probabilistic_Methods>: Papers that include probabilistic decision trees, learning probabilities for each subtree, and providing confidence measures should be categorized as <Probabilistic_Methods>.
<Reinforcement_Learning>: Papers that include learning from specific cases and experiences, value function approximation, and the use of neural networks for robust and efficient learning methods in reinforcement learning should be categorized as <Reinforcement_Learning>. Papers that include learning from specific cases and experiences, value function approximation, the use of... (too long)

You are provided with the characteristics of a node, another model's judgment of its category, and its true category. Please optimize the Pattern Descriptions for better prediction.
** Inputs (a paper): **
(Paper Title: Cognitive Computation (Extended Abstract) )
(Abstract: Cognitive computation is discussed as a discipline that links together neurobiology, cognitive psychology and artificial intelligence. )
(Neighbour Summary:
The papers cited in this essay discuss the concept of learning and reasoning, specifically focusing on the Probably Approximately Correct (PAC) learning model and its generalizations. The authors investigate the idea of agnostic learning, where minimal assumptions are made about the target function, and explore the possibilities of this approach through various results and algorithms. The papers also touch on the Learning to Reason framework, which combines learning and reasoning into a single task, and examine the tradeoff between learnability, oracle strength, and reasoning query range. Overall, the cited papers contribute to the understanding of learning and reasoning in artificial intelligence.)
** The model outputs: **
Theory
** The target outputs: **
Neural_Networks
If the model output differs from the target output, update the descriptions for both the model's output category and the target output category to reduce classification error on both current and future data. If the model and target outputs are the same, update the description for this category only. Limit your 'New Pattern Descriptions' to less than 150 words per category. Think step by step provide your output strictly in the following format:
** Output Format: **
Please think step by step and give your outputs strictly in the following format:
'''
Reasoning:
[Provide a concise summary of the reasoning process, focusing on the key insights that lead to the final decision. Avoid listing multiple steps. Ensure the reasoning is clear and justifies the new pattern descriptions.]
New Pattern Descriptions:
[Please insert your new descriptions here. The new descriptions MUST BE DIRECTLY based on the key features and insights identified in the input paper. Update the descriptions ONLY for the categories where the model output differs from the target output, or if they match, update ONLY that category. Ensure that only the following categories are used: <Case_Based>, <Genetic_Algorithms>, <Neural_Networks>, <Probabilistic_Methods>, <Reinforcement_Learning>, <Rule_Learning>, <Theory>. Each updated description MUST BE enclosed within <NewPattern></NewPattern> tags. Only include descriptions for the categories being updated. If the model output and target output are the same, update ONLY that category and NO OTHERS.]
For example:
<NewPattern><Case_Based>: The papers with [your new pattern description] should be categorized as <Case_Based>.</NewPattern>
<NewPattern><Genetic_Algorithms>: The papers that include [your new pattern description] should be categorized as <Genetic_Algorithms>.</NewPattern>
'''
Please ONLY reply according to this format, DON'T give me any other words. If the model output and target output are the same, update ONLY that category and NO OTHERS. Now take a deep breath and provide a concise and logical Reasoning and New Pattern Descriptions.assistant

Reasoning:
The paper discusses cognitive computation as a discipline that links neurobiology, cognitive psychology, and artificial intelligence. The neighbour summary highlights the concept of learning and reasoning, specifically focusing on the PAC learning model and its generalizations. The model output is "Theory," but the target output is "Neural_Networks." This discrepancy suggests that the model's output category is too broad, encompassing both theoretical and neural network-related topics. To improve the model's accuracy, we need to refine the descriptions for both categories.
New Pattern Descriptions:
<NewPattern><Theory>: The papers that discuss cognitive computation as a discipline linking neurobiology, cognitive psychology, and artificial intelligence, and focus on theoretical concepts such as learning and reasoning models should be categorized as <Theory>.</NewPattern>
<NewPattern><Neural_Networks>: The papers that include neural networks, pruning methods, real-time control, and application to input-state stability, as well as those that explore the generalization of classical theorems and approximation orders should be categorized as <Neural_Networks>.</NewPattern>

# G ZERO-SHOT W/ PRIOR SUMMARY + VGRL

## G.1 ZERO-SHOT W/ PRIOR SUMMARY + VGRL STEP 1

---

**zero-shot predictor LLM CoT with prior Step 1**

Inference Step
You will use the descriptions below to predict the output of the given information about a paper including: title, abstract and a summary of information about the papers cited in this paper. You need to tell me which categories this paper belong to:

<Case_Based>, <Genetic_Algorithm>, <Neural_Networks>, <Probabilistic_Methods>, <Reinforcement_Learning>, <Rule_Learning>, <Theory>.
Here is the pattern descriptions and the information about the paper:
** Pattern Descriptions: **
You will use the descriptions below.
<Case_Based>: If the paper discuss the use of past experiences, memory-based techniques, or case retrieval and adaptation processes to solve new problems, classify it as <Case_Based>.
<Genetic_Algorithms>: If the paper focuses on optimization techniques inspired by evolutionary processes, such as population-based search, crossover, and mutation, classify it as <Genetic_Algorithms>.
<Neural_Networks>: If the paper covers artificial neural networks, including feedforward, convolutional, or recurrent networks, and their training methods, classify it as <Neural_Networks>.
<Probabilistic_Methods>: If the paper discuss methods that use probabilistic models to handle uncertainty, such as Bayesian networks or hidden Markov models, classify it as <Probabilistic_Methods>.
<Reinforcement_Learning>: If the paper centers on agents learning from interaction with their environment through trial and error, using techniques like Q-learning or policy gradients, classify it as <Reinforcement_Learning>.
<Rule_Learning>: If the paper focuses on extracting interpretable decision rules or patterns, often in the form of "if-then" statements, classify it as <Rule_Learning>.
<Theory>: If the paper emphasizes theoretical contributions to machine learning or AI, such as mathematical models, complexity theory, or the formal analysis of algorithms, classify it as <Theory>.

** Input: **
(Paper Title: Stochastic Propositionalization of Non-Determinate Background Knowledge )
(Abstract: It is a well-known fact that propositional learning algorithms require "good" features to perform well in practice. So a major step in data engineering for inductive learning is the construction of good features by domain experts. These features often represent properties of structured objects, where a property typically is the occurrence of a certain substructure having certain properties. To partly automate the process of "feature engineering", we devised an algorithm that searches for features which are defined by such substructures. The algorithm stochastically conducts a top-down search for first-order clauses, where each clause represents a binary feature. It differs from existing algorithms in that its search is not class-blind, and that it is capable of considering clauses ("context") of almost arbitrary length (size). Preliminary experiments are favorable, and support the view that this approach is promising.)
(Neighbour Summary:
The papers cited in this essay discuss various techniques and approaches in the field of Rule Learning, specifically in Inductive Logic Programming (ILP). The first paper re-appraises the development of techniques for inverting deduction and introduces Mode-Directed Inverse Entailment (MDIE) as a generalisation and enhancement of previous approaches. The second paper proposes an extension of the feature-vector representation to allow the value of a feature to be a set of strings, which can be efficiently and naturally represented in real-world learning problems. The third paper describes a system named SFOIL that uses a stochastic search method to alleviate the local optimization problem in greedy algorithms. Overall, these papers highlight the importance of efficient and effective techniques in Rule Learning and ILP.)
** Output Format: **
Please give your output strictly in the following format:

Explanations: [Your step-by-step analyses and results]
Output:
[Pick one in (<Case_Based>, <Genetic_Algorithm>, <Neural_Networks>, <Probabilistic_Methods>, <Reinforcement_Learning>, <Rule_Learning>, <Theory>)], e.g. <Case_Based>

Please ONLY reply according to this format. Do not include any other words or step-by-step breakdowns. Now take a deep breath and provide a concise explanation and output.assistant

Explanation: The paper discusses the stochastic propositionalization of non-determinate background knowledge, which is related to the development of new definitions and methods for prediction algorithms. The paper also focuses on the analysis of their performance in various scenarios and the exploration of their theoretical properties. The cited papers are in the field of Rule Learning, specifically in Inductive Logic Programming (ILP), which further supports the connection to Case-Based methods.
Output: <Rule_Learning>

---

**zero-shot summary LLM CoT with prior Step 1**

Now that you have some descriptions, which will be used as rubrics for the classification task, you need to summarize and refine them for each category to find the best way to express them.
Here is the descriptions:

<Case_Based>: The papers that focus on the theoretical foundations of neural networks, including the development of new definitions and methods for prediction algorithms, the analysis of their performance in various scenarios, and the exploration of their theoretical properties, should be categorized as <Case_Based>. The papers that apply memory-based techniques and case-based reasoning to support design decision-making, such as using past experience to aid designers in exploring and refining design problems, should be categorized as <Case_Based>. The papers that focus on goal-driven learning, explanation, and introspection in artificial intelligence, emphasizing the need for a reasoner to reason about its performance and identify areas for improvement, should be categorized as <Case_Based>. This category includes papers that focus on the theoretical foundations of machine learning, including the development of new definitions and methods for prediction algorithm, the analysis of their performance in various scenarios, and the exploration of their theoretical properties. Papers in this category may discuss the use of mathematical concepts such as version-space, finite state automata (FSA), and probabilistic models to understand the behavior of prediction algorithms and improve their performance.
<Genetic_Algorithms>: The papers that focus on genetic algorithms, including their applications in solving optimization problems, graph partitioning, and other complex problems, should be categorized as <Genetic_Algorithms>.
<Neural_Networks>: The papers that discuss the strengths and weaknesses of neural networks, their applications in various fields, and the use of connectionist approaches, radial basis function networks, and the convergence of the backpropagation algorithm, should be categorized as <Neural_Networks>. The papers that develop and evaluate new neural network learning algorithms, including pruning methods, and emphasize the importance of experimental evaluation and assessment practices, should be categorized as <Neural_Networks>.
<Rule_Learning>: The papers that discuss the application of Inductive Logic Programming (ILP) techniques, including the construction of features by domain experts to automate the process of "feature engineering" should be categorized as <Rule_Learning>. The papers that discuss the development of new rules or strategies based on experience, particularly in the context of artificial intelligence, should be categorized as <Rule_Learning>.
<Theory>: The papers with a focus on theoretical foundations of machine learning, including the development of new definitions and methods for prediction algorithms, the analysis of their performance in various scenarios, and the exploration of their theoretical properties, should be categorized as <Theory>. The papers that focus on the theoretical foundations of machine learning, including the development of new definitions and methods for prediction algorithm, the analysis of their theoretical properties, particularly in the context of game theory and adversarial learning, should be categorized as <Theory>.

Note: Please ONLY summarize each category without omissions and NOTHING ELSE! And each category is output in the following format:
<Case_Based>: Papers that include [your summary] should be categorized as <Case_Based>.
<Genetic_Algorithms>: Papers that include [your summary] should be categorized as <Genetic_Algorithms>.
<Neural_Networks>: Papers that include [your summary] should be categorized as <Neural_Networks>.
<Probabilistic_Methods>: Papers that include [your summary] should be categorized as <Probabilistic_Methods>.
<Reinforcement_Learning>: Papers that include [your summary] should be categorized as <Reinforcement_Learning>.
<Rule_Learning>: Papers that include [your summary] should be categorized as <Rule_Learning>.
<Theory>: Papers that include [your summary] should be categorized as <Theory>.
Please begin with <START_SUMMARY>

<START_SUMMARY>
<Case_Based>: Papers that focus on the theoretical foundations of neural networks, case-based reasoning, goal-driven learning, explanation, and introspection in artificial intelligence, and the use of memory-based techniques to support design decision-making, should be categorized as <Case_Based>.
<Genetic_Algorithms>: Papers with a focus on genetic algorithms, their applications in solving optimization problems, graph partitioning, and other complex problems, should be categorized as <Genetic_Algorithms>.
<Neural_Networks>: Papers that discuss the strengths and weaknesses of neural networks, their applications, connectionist approaches, radial basis function networks, and the convergence of the backpropagation algorithm, should be categorized as <Neural_Networks>.
<Probabilistic_Methods>: Papers that discuss the use of mathematical concepts such as version-space, finite state automata (FSA), and probabilistic models to understand the behavior of prediction algorithms and improve their performance, should be categorized as <Probabilistic_Methods>.
<Reinforcement_Learning>: There is no summary for this category in the provided text.
<Rule_Learning>: Papers that discuss the application of Inductive Logic Programming (ILP) techniques, stochastic search methods, and the construction of features by domain experts to automate the process of "feature engineering", and the development of new rules or strategies based on experience, particularly in the context of artificial intelligence, should be categorized as <Rule_Learning>.
<Theory>: Papers with a focus on theoretical foundations of machine learning, including the development of new definitions and methods for prediction algorithms, the analysis of their performance in various scenarios, and the exploration of their theoretical properties, particularly in the context of game theory and adversarial learning, should be categorized as <Theory>.

---

**zero-shot optimizer LLM CoT with prior Step 1**

Optimization Step
Your goal is to learn the best descriptions for the model. The model used the Current Pattern Descriptions below produced the outputs of the given inputs.
** Current Pattern Descriptions: **
Please refer to the following judgement criteria to determine which category the following paper belongs to.

<Case_Based>: If the paper discusses the use of past experiences, memory-based techniques, or case retrieval and adaptation processes to solve new problems, classify it as <Case_Based>.
<Genetic_Algorithms>: If the paper focuses on optimization techniques inspired by evolutionary processes, such as population-based search, crossover, and mutation, classify it as <Genetic_Algorithms>.
<Neural_Networks>: If the paper covers artificial neural networks, including architectures like feedforward, convolutional, or recurrent networks, and their training methods, classify it as <Neural_Networks>.
<Probabilistic_Methods>: If the paper discusses methods that use probabilistic models to handle uncertainty, such as Bayesian networks or hidden Markov models, classify it as <Probabilistic_Methods>.
<Reinforcement_Learning>: If the paper centers on agents learning from interaction with their environment through trial and error, using techniques like Q-learning or policy gradients, classify it as <Reinforcement_Learning>.
<Rule_Learning>: If the paper focuses on extracting interpretable decision rules or patterns, often in the form of "if-then" statements, classify it as <Rule_Learning>.
<Theory>: If the paper emphasizes theoretical contributions to machine learning or AI, such as mathematical models, complexity theory, or the formal analysis of algorithms, classify it as <Theory>.

You are provided with the characteristics of a node, another model's judgment of its category, and its true category. Please optimize the Pattern Descriptions for better prediction.
** Inputs (a paper): **
(Paper Title: Stochastic Propositionalization of Non-Determinate Background Knowledge )
(Abstract: It is a well-known fact that propositional learning algorithms require "good" features to perform well in practice. A major step in data engineering for inductive learning is the construction of good features by domain experts. These features often represent properties of structured objects, where a property typically is the occurrence of a certain substructure having certain properties. To partly automate the process of "feature engineering", we devised an algorithm that searches for features which are defined by such substructures. The algorithm stochastically conducts a top-down search for first-order clauses, where each clause represents a binary feature. It differs from existing algorithms in that its search is not class-blind, and that it is capable of considering clauses ("context") of almost arbitrary length (size). Preliminary experiments are favorable, and support the view that this approach is promising.)
(Neighbour Summary:
The papers cited in this essay discuss various techniques and approaches in the field of Rule Learning, specifically in Inductive Logic Programming (ILP). The first paper re-appraises the development of techniques for inverting deduction and introduces Mode-Directed Inverse Entailment (MDIE) as a generalisation and enhancement of previous approaches. The second paper proposes an extension of the feature-vector representation to allow the value of a feature to be a set of strings, which can be efficiently and naturally represented in real-world learning problems. The third paper describes a system named SFOIL that uses a stochastic search method to alleviate the local optimization problem in greedy algorithms. Overall, these papers highlight the importance of efficient and effective techniques in Rule Learning and ILP.)
** The model outputs: **
Rule_Learning
** The target outputs: **
Rule_Learning
If the model output differs from the target output, update the descriptions for both the model's output category and the target output category to reduce classification error on both current and future data. If the model and target outputs are the same, update the description for this category only. Limit your 'New Pattern Descriptions' to less than 150 words per category. Think step by step provide your output strictly in the following format:
** Output Format: **
Please think step by step and give your outputs strictly in the following format:
```
Reasoning:
[Provide a concise summary of the reasoning process, focusing on the key insights that lead to the final decision. Avoid listing multiple steps. Ensure the reasoning is clear and justifies the new pattern descriptions.]
New Pattern Descriptions:
[Please insert your new descriptions here. The new descriptions MUST BE DIRECTLY based on the key features and insights identified in the input paper. Update the descriptions ONLY for the categories where the model output differs from the target output, or if they match, update ONLY that category. Ensure that only the following categories are used: <Case_Based>, <Genetic_Algorithms>, <Neural_Networks>, <Probabilistic_Methods>, <Reinforcement_Learning>, <Rule_Learning>, <Theory>. Each updated description MUST BE enclosed within <NewPattern></NewPattern> tags. Only include descriptions for the categories being updated. If the model output and target output are the same, update ONLY that category and NO OTHERS.]
For example:
<NewPattern><Case_Based>: The papers with [your new pattern description] should be categorized as <Case_Based>.</NewPattern>
<NewPattern><Genetic_Algorithms>: The papers with [your new pattern description] should be categorized as <Genetic_Algorithms>.</NewPattern>
```

Please ONLY reply according to this format, DON'T give me any other words. If the model output and target output are the same, update ONLY that category and NO OTHERS. Now take a deep breath and provide a concise and logical Reasoning and New Pattern Descriptions.assistant

Reasoning:
The input paper discusses the stochastic proposition of non-determinate background knowledge, which involves the construction of features by domain experts and the use of an algorithm to search for features defined by substructures. The paper is related to Rule Learning, specifically in Inductive Logic Programming (ILP), as it discusses techniques and approaches in this field. The model output is Rule_Learning, which matches the target output. However, the current pattern descriptions for Rule_Learning do not fully capture the essence of the paper. The new pattern descriptions should focus on the key features of the paper, such as the use of stochastic search, the construction of features by domain experts, and the application of ILP techniques.
New Pattern Descriptions:
<NewPattern><Rule_Learning>: The papers that discuss the application of Inductive Logic Programming (ILP) techniques, stochastic search methods, and the construction of features by domain experts to automate the process of "feature engineering" should be categorized as <Rule_Learning>.</NewPattern>

---

1350
1351
1352

## G.2 ZERO-SHOT W/ PRIOR SUMMARY + VGRL STEP 2

1353
1354
...

**zero-shot predictor LLM CoT with prior Step 2**

Inference Step
You will use the descriptions below to predict the output of the given information about a paper including: title, abstract and a summary of information about the papers cited in this paper. You need to tell me which of the following categories this paper belongs to:
<Case_Based>, <Genetic_Algorithms>, <Neural_Networks>, <Probabilistic_Methods>, <Reinforcement_Learning>, <Rule_Learning>, <Theory>
Here is the pattern descriptions and the information about the paper:
** Pattern Descriptions: **
You will use the description below.

<Case_Based>: Papers that focus on the theoretical foundations of neural network, case-based reasoning, goal-driven learning, and introspection in artificial intelligence, and the use of memory-based techniques to support design decision-making, should be categorized as <Case_Based>.
<Genetic_Algorithms>: Papers with a focus on genetic algorithms, their applications in solving optimization problems, graph partitioning, and other complex problems, should be categorized as <Genetic_Algorithms>.
<Neural_Networks>: Papers that discuss the strengths and weaknesses of neural networks, their applications, connectionist approaches, radial basis function networks, and the convergence of the backpropagation algorithm, should be categorized as <Neural_Networks>.
<Probabilistic_Methods>: Papers that discuss the use of mathematical concepts such as version-space, finite state automata (FSA), and probabilistic models to understand the behavior of prediction algorithms and improve their performance, should be categorized as <Probabilistic_Methods>.
<Reinforcement_Learning>: There is no summary for this category in the provided text.
<Rule_Learning>: Papers that discuss the application of Inductive Logic Programming (ILP) techniques, stochastic search methods, and the construction of features by domain experts to automate the process of "feature engineering", and the development of new rules or strategies based on experience, particularly in the context of artificial intelligence, should be categorized as <Rule_Learning>.
<Theory>: Papers with a focus on theoretical foundations of machine learning, including the development of new definitions and methods for prediction algorithms, the analysis of their performance in various scenarios, and the exploration of their theoretical properties, particularly in the context of game theory and adversarial learning, should be categorized as <Theory>.

** Input: **
(Paper Title: Learning from positive data )
(Abstract: Gold showed in 1967 that not even regular grammars can be exactly identified from positive examples alone. Since it is known that children learn natural grammars almost exclusively from positives examples, Gold's result has been used as a theoretical support for Chomsky's theory of innate human linguistic abilities. In this paper new results are presented which show that within a Bayesian framework not only grammars, but also logic programs are learnable with arbitrarily low expected error from positive examples only. In addition, we show that the upper bound for expected error of a learner which maximises the Bayes' posterior probability when learning from positive examples is within a small additive term of one which does the same from a mixture of positive and negative examples. An Inductive Logic Programming implementation is described which avoids the pitfalls of greedy search by global optimisation of this function during the local construction of individual clauses of the hypothesis. Results of testing this implementation on artificially-generated data-sets are reported. These results are in agreement with the theoretical predictions. )
(Neighbour Summary:
The papers cited in this essay discuss the limitations of current inductive logic programming systems in handling noise and learning recursive predicates. A new framework for learning from noisy data and fixed example size is presented, which incorporates a Bayesian heuristic for finding the most probable hypothesis. This approach is implemented in an ILP system called Lime, which is shown to handle noise better than other systems like FOIL and PROGOL, and is capable of learning from only positive or negative data.
Common points that reflect the category of this paper include:
- Handling noise in inductive logic programming systems
- Learning recursive predicates
- Improving the performance of ILP systems
- Developing new frameworks for learning from noisy data.)
** Output Format: **
Please give your output strictly in the following format:
```
Explanations: [Your step-by-step analyses and results]
Output:
[Pick one in <Case_Based>, <Genetic_Algorithms>, <Neural_Networks>, <Probabilistic_Methods>, <Reinforcement_Learning>, <Rule_Learning>, <Theory>], e.g. <Case_Based>
```
Please ONLY reply according to this format. Do not include any other words or step-by-step breakdowns.
Now take a deep breath and provide a concise explanation and output.assistant

Explanations: The paper discusses the theoretical foundations of inductive logic programming, specifically the development of new frameworks for learning from noisy data and the improvement of ILP systems. It presents a Bayesian approach for learning from positive examples and compares its performance with other systems. The paper also discusses the limitations of current ILP systems and presents a new implementation that avoids greedy search.
Output: <Rule_Learning>

**zero-shot summary LLM CoT with prior Step 2**

Now that you have some descriptions, you need to use them as rubrics for the classification task, you need to summarize and refine them for each category to find the best way to express them.
Here are the descriptions:

<Case_Based>: Papers that focus on the theoretical foundations of neural networks, case-based reasoning, goal-driven learning, explanation, and introspection in artificial intelligence, and the use of memory-based technique to support design decision-making, should be categorized as <Case_Based>. The papers that focus on case-based reasoning, including the use of precedents, rules, and analogy in legal decision-making, and the adaptation to mismatches between past experience and new problems, should be categorized as <Case_Based>. The papers that focus on Case-Based Planning (CBP), explanation-based learning, derivation replay, and the improvement of case similarity judgements through feedback should be categorized as <Case_Based>.
<Genetic_Algorithms>: Papers with a focus on genetic algorithms, their applications in solving optimization problems, graph partitioning, and other complex problems, should be categorized as <Genetic_Algorithms>.
<Neural_Networks>: Papers that discuss the strengths and weaknesses of neural networks, their applications, connectionist approaches, radial basis function networks, and the convergence of the backpropagation algorithm, should be categorized as <Neural_Networks>.
<Probabilistic_Methods>: Papers that discuss the use of mathematical concepts such as version-space, finite state automata (FSA), and probabilistic models to understand the behavior of prediction algorithms and improve their performance, feature construction, selection, and subset selection, particularly in the presence of irrelevant features, and exploring the use of stochastic techniques, greedy algorithms, and statistical measures to optimize performance, should be categorized as <Probabilistic_Methods>. The papers that discuss the use of probabilistic processes to classify incomplete examples, model underlying performance tasks as probabilistic processes, and apply probabilistic models to understand the behavior of prediction algorithms, should be categorized as <Probabilistic_Methods>.
<Reinforcement_Learning>: There is no summary for this category in the provided text.
<Rule_Learning>: Papers that discuss the application of Inductive Logic Programming (ILP) techniques, stochastic search methods, and the construction of features by domain experts to automate the process of "feature engineering", and the development of new rule or strategies based on experience, particularly in the context of artificial intelligence, and the development of new rules or strategies based on experience, particularly in the context of artificial intelligence, and the improvement of ILP systems' performance by handling noise and learning recursive predicates, should be categorized as <Rule_Learning>.
<Theory>: Papers with a focus on theoretical foundations of machine learning, including the development of new definitions and methods for prediction algorithms, the analysis of their performance in various scenarios, and the exploration of their theoretical properties, particularly in the context of game theory and adversarial learning, should be categorized as <Theory>. The papers that discuss the theoretical foundations of machine learning, including the development of new definitions and methods for prediction algorithm, the analysis of their performance in various scenarios, and the exploration of their theoretical properties, particularly in the context of game theory and adversarial learning, and the use of Bayesian framework to show the learnability of logic programs with arbitrarily low expected error from positive examples only, should be categorized as <Theory>. The papers with a focus on developing new definitions and methods for prediction algorithms, exploring the theoretical properties of machine learning algorithms, and discussing the importance of considering the bias and complexity of learning algorithms should be categorized as <Theory>. The papers that discuss theoretical foundations of machine learning, including the development of new models, the analysis of their theoretical properties, particularly in the context of probabilistic processes, pac-learning frameworks, and knowledge representation, should be categorized as <Theory>. The papers that discuss the theoretical foundations of machine learning, including the development of new models, the analysis of their performance in various scenarios, and the exploration of their theoretical properties, particularly in the context of game theory and adversarial learning, should be categorized as <Theory>. The papers that discuss the theoretical foundations of machine learning, including the development of new definitions and methods for prediction algorithms, the analysis of their performance in various scenarios, and the exploration of their theoretical properties, particularly in the context of game theory and adversarial learning, should be categorized as <Theory>.

Note: Please ONLY summarize each category without omissions and NOTHING ELSE! And each category is output in the following format:
<Case_Based>: Papers that include [your summary] should be categorized as <Case_Based>.
<Genetic_Algorithms>: Papers that include [your summary] should be categorized as <Genetic_Algorithms>.
<Neural_Networks>: The papers that include [your summary] should be categorized as <Neural_Networks>.
<Probabilistic_Methods>: Papers that include [your summary] should be categorized as <Probabilistic_Methods>.
<Reinforcement_Learning>: Papers that include [your summary] should be categorized as <Reinforcement_Learning>.
<Rule_Learning>: Papers that include [your summary] should be categorized as <Rule_Learning>.
<Theory>: Papers that include [your summary] should be categorized as <Theory>.
Please begin with <START_SUMMARY>

<START_SUMMARY>
<Case_Based>: Papers that focus on neural networks' theoretical foundations, case-based reasoning, goal-driven learning, and introspection in artificial intelligence, as well as memory-based techniques for design decision-making, case-based planning, explanation-based learning, derivation replay, and improvement of case similarity judgements through feedback, should be categorized as <Case_Based>.
<Genetic_Algorithms>: Papers with a focus on genetic algorithms and their applications in solving optimization problems, graph partitioning, and other complex problems, should be categorized as <Genetic_Algorithms>.
<Neural_Networks>: Papers that discuss the strengths and weaknesses of neural networks, their applications, connectionist approaches, radial basis function networks, and the convergence of the backpropagation algorithm, should be categorized as <Neural_Networks>.
<Probabilistic_Methods>: Papers that discuss the use of mathematical concepts such as version-space, finite state automata (FSA), and probabilistic models to understand the behavior of prediction algorithms and improve their performance, feature construction, selection, and subset selection, particularly in the presence of irrelevant features, and the use of stochastic techniques, greedy algorithms, and statistical measures to classify incomplete examples, model underlying performance tasks as probabilistic processes, and apply probabilistic models to understand the behavior of prediction algorithms, should be categorized as <Probabilistic_Methods>.
<Reinforcement_Learning>: Papers that discuss the topic of reinforcement learning should be categorized as <Reinforcement_Learning>.
<Rule_Learning>: Papers that discuss the application of Inductive Logic Programming (ILP) techniques, stochastic search methods, and the construction of features by domain experts to automate the process of "feature engineering", the development of new rule or strategies based on experience, particularly in the context of artificial intelligence, and the improvement of ILP systems' performance by handling noise and learning recursive predicates, should be categorized as <Rule_Learning>.
<Theory>: Papers with a focus on theoretical foundations of machine learning, including the development of new definitions and methods for prediction algorithms, the analysis of their performance in various scenarios, the exploration of their theoretical properties, particularly in the context of game theory and adversarial learning, probabilistic processes, pac-learning frameworks, knowledge representation, and the Probably Approximately Correct (PAC) model, should be categorized as <Theory>.

**zero-shot optimizer LLM CoT with prior Step 2**

Optimization Step
Your goal is to learn the best descriptions for the model. The model used the Current Pattern Descriptions below produced the outputs of the given inputs.
** Current Pattern Descriptions: **
Please refer to the following judgement criteria to determine which category the following paper belongs to.

<Case_Based>: Papers that focus on the theoretical foundations of neural networks, case-based reasoning, goal-driven learning, explanation, and introspection in artificial intelligence, and the use of memory-based techniques to support design decision-making, should be categorized as <Case_Based>.
<Genetic_Algorithms>: Papers with a focus on genetic algorithms, their applications in solving optimization problems, graph partitioning, and other complex problems, should be categorized as <Genetic_Algorithms>.
<Neural_Networks>: Papers that discuss the strengths and weaknesses of neural networks, their applications, connectionist approaches, radial basis function networks, and the convergence of the backpropagation algorithm, should be categorized as <Neural_Networks>.
<Probabilistic_Methods>: Papers that discuss the use of mathematical concepts such as version-space, finite state automata (FSA), and probabilistic models to understand the behavior of prediction algorithms and improve their performance, should be categorized as <Probabilistic_Methods>.
<Reinforcement_Learning>: There is no summary for this category in the provided text.
<Rule_Learning>: Papers that discuss the application of Inductive Logic Programming (ILP) techniques, stochastic search methods, and the construction of features by domain experts to automate the process of "feature engineering", and the development of new rules or strategies based on experience, particularly in the context of artificial intelligence, should be categorized as <Rule_Learning>.
<Theory>: Papers with a focus on theoretical foundations of machine learning, including the development of new definitions and methods for prediction algorithms, the analysis of their performance in various scenarios, and the exploration of their theoretical properties, particularly in the context of game theory and adversarial learning, should be categorized as <Theory>.

You are provided with the characteristics of a node, another model's judgment of its category, and its true category. Please optimize the Pattern Descriptions for better prediction.
** Inputs (a paper): **
(Paper Title: Learning from positive data )
(Abstract: Gold showed in 1967 that not even regular grammars can be exactly identified from positive examples alone. Since it is known that children learn natural grammars almost exclusively from positives examples, Gold's result has been used as a theoretical support for Chomsky's theory of innate human linguistic abilities. In this paper new results are presented which show that within a Bayesian framework not only grammars, but also logic programs are learnable with arbitrarily low expected error from positive examples only. In addition, we show that the upper bound for expected error of a learner which maximises the Bayes' posterior probability when learning from positive examples is within a small additive term of one which does the same from a mixture of positive and negative examples. An Inductive Logic Programming implementation is described which avoids the pitfalls of greedy search by global optimisation of this function during the local construction of individual clauses of the hypothesis. Results of testing this implementation on artificially-generated data-sets are reported. These results are in agreement with the theoretical predictions. )
(Neighbour Summary:
The papers cited in this essay discuss the limitations of current inductive logic programming systems in handling noise and learning recursive predicates. A new framework for learning from noisy data and fixed example size is presented, which incorporates a Bayesian heuristic for finding the most probable hypothesis. This approach is implemented in an ILP system called Lime, which is shown to handle noise better than other systems like FOIL and PROGOL, and is capable of learning from only positive or negative data.
Common points that reflect the category of this paper include:
- Handling noise in inductive logic programming systems
- Learning recursive predicates
- Improving the performance of ILP systems
- Developing new frameworks for learning from noisy data.)
** The model outputs: **
Rule_Learning
** The target outputs: **
Theory
If the model output differs from the target output, update the descriptions for both the model's output category and the target output category to reduce classification error on both current and future data. If the model and target outputs are the same, update the description for this category only. Limit your 'New Pattern Descriptions' to less than 150 words per category. Think step by step provide your output strictly in the following format:
** Output Format: **
Please think step by step and give your outputs strictly in the following format:
```
Reasoning:
[Provide a concise summary of the reasoning process, focusing on the key insights that lead to the final decision. Avoid listing multiple steps. Ensure the reasoning is clear and justifies the new pattern descriptions.]
New Pattern Descriptions:
[Please insert your new descriptions here. The new descriptions MUST BE DIRECTLY based on the key features and insights identified in the input paper. Update the descriptions ONLY for the categories where the model output differs from the target output, or if they match, update ONLY that category. Ensure that only the following categories are used: <Case_Based>, <Genetic_Algorithms>, <Neural_Networks>, <Probabilistic_Methods>, <Reinforcement_Learning>, <Rule_Learning>, <Theory>. Each updated description MUST BE enclosed within <NewPattern></NewPattern> tags. Only include descriptions for the categories being updated. If the model output and target output are the same, update ONLY that category and NO OTHERS.]
For example:
<NewPattern><Case_Based>: The papers with [your new pattern description] should be categorized as <Case_Based>.</NewPattern>
<NewPattern><Genetic_Algorithms>: The papers with [your new pattern description] should be categorized as <Genetic_Algorithms>.</NewPattern>
```
Please ONLY reply according to this format, DON'T give me any other words. If the model output and target output are the same, update ONLY that category and NO OTHERS. Now take a deep breath and provide a concise and logical Reasoning and New Pattern Descriptions.assistant

Reasoning:
The paper discusses the theoretical foundations of learning from positive data, presenting new results within a Bayesian framework that show the learnability of logic programs with arbitrarily low expected error from positive examples only. The paper also describes an Inductive Logic Programming implementation that avoids the pitfalls of greedy search by global optimisation of the function during the local construction of individual clauses of the hypothesis. The common points that reflect the category of this paper include handling noise in inductive logic programming systems, learning recursive predicates, improving the performance of ILP systems, and developing new frameworks for learning from noisy data. These points align more closely with the characteristics of the <Theory> category, which focuses on theoretical foundations of machine learning, including the development of new definitions and methods for prediction algorithms, the analysis of their performance in various scenarios, and the exploration of their theoretical properties.
New Pattern Descriptions:
<NewPattern><Theory>: The papers that discuss the theoretical foundations of machine learning, including the development of new definitions and methods for prediction algorithms, the analysis of their performance in various scenarios, and the exploration of their theoretical properties, particularly in the context of game theory and adversarial learning, and the use of Bayesian frameworks to show the learnability of logic programs with arbitrarily low expected error from positive examples only, should be categorized as <Theory>.</NewPattern>
<NewPattern><Rule_Learning>: The papers that discuss the application of Inductive Logic Programming (ILP) techniques, stochastic search methods, and the construction of features by domain experts to automate the process of "feature engineering", and the development of new rules or strategies based on experience, particularly in the context of artificial intelligence, and the improvement of ILP systems' performance by handling noise and learning recursive predicates, should be categorized as <Rule_Learning>.</NewPattern>

## G.3 ZERO-SHOT W/ PRIOR SUMMARY + VGRL STEP 80

**zero-shot predictor LLM CoT with prior Step 80**

Inference Step
You will use the descriptions below to predict the output of the given information about a paper including: title, abstract and a summary of information about the papers cited in this paper. You will use the descriptions to predict which of the following categories this paper belongs to:
<Case_Based>, <Genetic_Algorithms>, <Neural_Networks>, <Probabilistic_Methods>, <Reinforcement_Learning>, <Rule_Learning>, <Theory>.
Here is the pattern descriptions and the information about the paper:
** Pattern Descriptions: **
You will use the descriptions below.

<Case_Based>: Papers that include case-based reasoning, learning from past experiences, adaptation to new situations, recall and adaptation of previously known designs, adaptation knowledge, feature selection, condensation, improving machine learning algorithms efficiency, proposing new approaches to enhance performance, applying these approaches in real-world problems, constraint satisfaction, synergy between case-based reasoning and constraint satisfaction problems, memory-based techniques, interactive exploration, case-based design systems, case-based problem-solving systems, similarity assessment, incremental learning, adaptation to changing environments, inductive learning systems, Galois lattices, classification rules, boolean features, numerical features, finite-state automata in neural networks, training the free parameters of a scientific model to optimize its accuracy for making future predictions, prior knowledge, especially in robotic control systems, case-based classification, lazy learning mechanisms, nearest neighbor classification, registration processes, human-computer systems, mixed-initiative systems, and applications in robotics, artificial intelligence, and crisis response planning should be categorized as <Case_Based>.
<Genetic_Algorithms>: Papers that include genetic algorithms, artificial life research, neuro-evolution, solving optimization problems, graph partitioning, complex problems, proposing new approaches to improve performance, applications in theoretical biology and neuro-evolution, genetic algorithms for optimization, comparison with other methods, application in machine learning, feature selection, genetic algorithms in robotics, learning classifier systems, evolving biases, decision tree induction algorithm, evolutionary algorithms, novel extensions to existing frameworks, improvements in performance, machine learning techniques, similarity-based case retrieval, inductive decision tree learning, cultural transmission mechanisms, optimization problems, neuro-evolution, online evolution, incremental learning, adaptation to novel situations, applying genetic algorithms to optimize or search for solutions in complex systems, exploring the effectiveness of various crossover operators, self-organization of feedforward synapses, associative feedback mechanisms, and complex processes for learning non-linearly separable mappings should be categorized as <Genetic_Algorithms>.
<Neural_Networks>: Papers that include neural networks, connectionist approaches, radial basis function networks, backpropagation algorithm, generative models, decision trees, similarity-based case retrieval, relational regression, ensemble methods, exploratory projection pursuit, and their applications in areas such as adaptive logic, robotics, dynamic control, and database systems, including those that discuss neural networks' ability to learn and represent complex information, massively parallel processing, incremental learning, and applications in areas such as adaptive logic, robotics, and dynamic
<Probabilistic_Methods>: Papers ...(too long)

** Input: **
(Paper Title: Cognitive Computation (Extended Abstract) )
(Abstract: Cognitive computation is discussed as a discipline that links together neurobiology, cognitive psychology and artificial intelligence. )
(Neighbour Summary:
The paper cited in this essay discuss the concept of learning and reasoning, specifically focusing on the Probably Approximately Correct (PAC) learning model and its generalizations. The authors investigate the idea of agnostic learning, where minimal assumptions are made about the target function, and explore the possibilities of this approach through various results and algorithms. The papers also touch on the Learning to Reason framework, which combines learning and reasoning into a single task, and examine the tradeoff between learnability, oracle strength, and reasoning query range. Overall, the cited papers contribute to the understanding of learning and reasoning in artificial intelligence.)
** Output Format: **
Please give your output strictly in the following format:

Explanations: [Your step-by-step analyses and results]
Output:
[Pick one in (<Case_Based>, <Genetic_Algorithms>, <Neural_Networks>, <Probabilistic_Methods>, <Reinforcement_Learning>, <Rule_Learning>, <Theory>)], e.g. <Case_Based>

Please ONLY reply according to this format. Do not include any other words or step-by-step breakdowns.
Now take a deep breath and provide a concise explanation and output.assistant

Explanation: The paper discusses cognitive computation as a discipline that links neurobiology, cognitive psychology, and artificial intelligence. The cited papers focus on learning and reasoning, specifically the PAC learning model and its generalizations, agnostic learning, and the Learning to Reason framework. These topics are related to probabilistic methods, which involve making predictions or decisions based on probability distributions.
Output: <Probabilistic_Methods>

**zero-shot summary LLM CoT with prior Step 80**

Now that you have some descriptions, which will be used as rubrics for the classification task, you need to summarize and refine them for each category to find the best way to express them.
Here are the descriptions:

<Case_Based>: Papers that include case-based reasoning, learning from past experiences, adaptation to new situations, recall and adaptation of previously known designs, adaptation knowledge, feature selection, condensation, improving machine learning algorithms efficiency, proposing new approaches to enhance performance, applying these approaches in real-world problems, constraint satisfaction, synergy between case-based reasoning and constraint satisfaction problems, memory-based techniques, interactive exploration, case-based design systems, case-based problem-solving systems, similarity assessment, incremental learning, adaptation to changing environments, inductive learning systems, Galois lattices, classification rules, boolean features, numerical features, finite-state automata in neural networks, training the free parameters of a scientific model to optimize its accuracy for making future predictions, prior knowledge, especially in robotic control systems, case-based classification, lazy learning mechanisms, nearest neighbor classification, registration processes, human-computer systems, mixed-initiative systems, and applications in robotics, artificial intelligence, and crisis response planning should be categorized as <Case_Based>. The papers that integrate case-based reasoning techniques with other reasoning methods, such as constraint reasoning, for solving complex problems, especially in domains like environmental decision support systems, should be categorized as <Case_Based>. The papers that focus on Case-Based Reasoning, Classification Methodologies, Inductive Learning, and Knowledge Discovery, such as learning from past experiences, adaptation to new situations, and generating classification rules from data, should be categorized as <Case_Based>.
<Genetic_Algorithms>: Papers that include genetic algorithms, artificial life research, neuro-evolution, solving optimization problems, graph partitioning, complex problems, proposing new approaches to improve performance, applications in theoretical biology and neuro-evolution, genetic algorithms for optimization, comparison with other methods, application in machine learning, feature selection, genetic algorithms in robotics, learning classifier systems, evolving biases, decision tree induction algorithm, evolutionary algorithms, novel extensions to existing frameworks, improvements in performance, machine learning techniques, similarity-based case retrieval, inductive decision tree learning, cultural transmission mechanisms, optimization problems, neuro-evolution, online evolution, incremental learning, adaptation to novel situations, applying genetic algorithms to optimize or search for solutions in complex systems, exploring the effectiveness of various crossover operators, self-organization of feedforward synapses, associative feedback mechanisms, and complex processes for learning non-linearly separable mappings should be categorized as <Genetic_Algorithms>. The papers that use genetic algorithms in combination with other machine learning techniques, evolutionary learning, crossover operators, and their applications in optimization problems, complex systems, and machine learning should be categorized as <Genetic_Algorithms>. The papers that use genetic algorithms to improve case adaptability or propose new approaches to enhance performance in Case-Based Reasoning should be categorized as <Genetic_Algorithms>.
<Neural_Networks>: Papers that include neural networks, connectionist approaches, radial basis function networks, backpropagation algorithm, generative models, decision trees, similarity-based case retrieval, relational regression, ensemble methods, exploratory projection pursuit, and their applications in areas such as adaptive logic, robotics, dynamic control, and database systems, including those that discuss neural networks' ability to learn and represent complex information, massively parallel processing, incremental learning, and applications in areas such as adaptive logic, robotics, and dynamic control should be categorized as <Neural_Networks>. The papers that link neurobiology...(too long)
Note: Please ONLY summarize each category without omissions and NOTHING ELSE! And each category is output in the following format:
<Case_Based>: Papers that include [your summary] should be categorized as <Case_Based>.
<Genetic_Algorithms>: Papers that include [your summary] should be categorized as <Genetic_Algorithms>.
<Neural_Networks>: Papers that include [your summary] should be categorized as <Neural_Networks>.
<Probabilistic_Methods>: Papers that include [your summary] should be categorized as <Probabilistic_Methods>.
<Reinforcement_Learning>: Papers that include [your summary] should be categorized as <Reinforcement_Learning>.
<Rule_Learning>: Papers that include [your summary] should be categorized as <Rule_Learning>.
<Theory>: Papers that include [your summary] should be categorized as <Theory>.
Please begin with <START_SUMMARY>

<START_SUMMARY>
<Case_Based>: Papers that include case-based reasoning, learning from past experiences, adaptation to new situations, recall and adaptation of previously known designs, adaptation knowledge, feature selection, condensation, improving machine learning algorithms efficiency, proposing new approaches to enhance performance, applying these approaches in real-world problems, constraint satisfaction, synergy between case-based reasoning and constraint satisfaction problems, memory-based techniques, interactive exploration, case-based design systems, case-based problem-solving systems, similarity assessment, incremental learning, adaptation to changing environments, inductive learning systems, Galois lattices, classification rules, boolean features, numerical features, finite-state automata in neural networks, training the free parameters of a scientific model to optimize its accuracy for making future predictions, prior knowledge, especially in robotic control systems, case-based classification, lazy learning mechanisms, nearest neighbor classification, registration processes, human-computer systems, mixed-initiative systems, and applications in robotics, artificial intelligence, and crisis response planning.
<Genetic_Algorithms>: Papers that include genetic algorithms, artificial life research, neuro-evolution, solving optimization problems, graph partitioning, complex problems, proposing new approaches to improve performance, applications in theoretical biology and neuro-evolution, genetic algorithms for optimization, comparison with other methods, application in machine learning, feature selection, genetic algorithms in robotics, learning classifier systems, evolving biases, decision tree induction algorithm, evolutionary algorithms, novel extensions to existing frameworks, improvements in performance, machine learning techniques, similarity-based case retrieval, inductive decision tree learning, cultural transmission mechanisms, optimization problems, neuro-evolution, online evolution, incremental learning, adaptation to novel situations, applying genetic algorithms to optimize or search for solutions in complex systems, exploring the effectiveness of various crossover operators, self-organization of feedforward synapses, associative feedback mechanisms, and complex processes for learning non-linearly separable mappings.
<Neural_Networks>: Papers that include neural networks, connectionist approaches, radial basis function networks, backpropagation algorithm, generative models, decision trees, similarity-based case retrieval, relational regression, ensemble methods, exploratory projection pursuit, and their applications in areas such as adaptive logic, robotics, dynamic control, and database systems, including those that discuss neural networks' ability to learn and represent complex information, massively parallel processing, incremental learning, and applications in areas such as adaptive logic, robotics, and dynamic control.
<Probabilistic_Methods>: Papers that discuss Probably Approximately Correct (PAC) learning models, agnostic learning, and the tradeoff between...(too long)

**zero-shot optimizer LLM CoT with prior Step 80**

Optimization Step
Your goal is to learn the best descriptions for the model. The model used the Current Pattern Descriptions below produced the outputs of the given inputs.
** Current Pattern Descriptions: **
Please refer to the following judgement criteria to determine which category the following paper belongs to.

<Case_Based>: Papers that include case-based reasoning, learning from past experiences, adaptation to new situations, recall and adaptation of previously known designs, adaptation knowledge, feature selection, condensation, improving machine learning algorithms efficiency, proposing new approaches to enhance performance, applying these approaches in real-world problems, constraint satisfaction, synergy between case-based reasoning and constraint satisfaction problems, memory-based techniques, interactive exploration, case-based design systems, case-based problem-solving systems, similarity assessment, incremental learning, adaptation to changing environments, inductive learning systems, Galois lattices, classification rules, boolean features, numerical features, finite-state automata in neural networks, training the free parameters of a scientific model to optimize its accuracy for making future predictions, prior knowledge, especially in robotic control systems, case-based classification, lazy learning mechanisms, nearest neighbor classification, registration processes, human-computer systems, mixed-initiative systems, and applications in robotics, artificial intelligence, and crisis response planning should be categorized as <Case_Based>.
<Genetic_Algorithms>: Papers that include genetic algorithms, artificial life research, neuro-evolution, solving optimization problems, graph partitioning, complex problems, proposing new approaches to improve performance, applications in theoretical biology and neuro-evolution, genetic algorithms for optimization, comparison with other methods, application in machine learning, feature selection, genetic algorithms in robotics, learning classifier systems, evolving biases, decision tree induction algorithm, evolutionary algorithms, novel extensions to existing frameworks, improvements in performance, machine learning techniques, similarity-based case retrieval, inductive decision tree learning, cultural transmission mechanisms, optimization problems, neuro-evolution, online evolution, incremental learning, adaptation to novel situations, applying genetic algorithms to optimize or search for solutions in complex systems, exploring the effectiveness of various crossover operators, self-organization of feedforward synapses, associative feedback mechanisms, and complex processes for learning non-linearly separable mappings should be categorized as <Genetic_Algorithms>.
<Neural_Networks>: Papers that include neural networks, connectionist approaches, radial basis function networks, backpropagation algorithm, generative models, decision trees, similarity-based case retrieval, relational regression, ensemble methods, exploratory projection pursuit, and their applications in areas such as adaptive logic, robotics, dynamic control, and database systems, including those that discuss neural networks' ability to learn and represent complex information, massively parallel processing, incremental learning, and applications in areas such as adaptive logic, robotics, and dynamic control should be categorized as <Neural_Networks>.
<Probabilistic_Methods>: Papers...(too long)

You are provided with the characteristics of a node, another model's judgment of its category, and its true category. Please optimize the Pattern Descriptions for better prediction.
** Inputs (a paper): **
(Paper Title: Cognitive Computation (Extended Abstract) )
(Abstract: Cognitive computation is discussed as a discipline that links together neurobiology, cognitive psychology and artificial intelligence. )
(Neighbour Summary:
The papers cited in this essay discuss the concept of learning and reasoning, specifically focusing on the Probably Approximately Correct (PAC) learning model and its generalizations. The authors investigate the idea of agnostic learning, where minimal assumptions are made about the target function, and explore the possibilities of this approach through various results and algorithms. The papers also touch on the Learning to Reason framework, which combines learning and reasoning into a single task, and examine the tradeoff between learnability, oracle strength, and reasoning query range. Overall, the cited papers contribute to the understanding of learning and reasoning in artificial intelligence.)
** The model outputs: **
Probabilistic_Methods
** The target outputs: **
Neural_Networks
If the model output differs from the target output, update the descriptions for both the model's output category and the target output category to reduce classification error on both current and future data. If the model and target outputs are the same, update the description for this category only. Limit your 'New Pattern Descriptions' to less than 150 words per category. Think step by step provide your output strictly in the following format:
** Output Format: **
Please think step by step and give your outputs strictly in the following format:

Reasoning:
[Provide a concise summary of the reasoning process, focusing on the key insights that lead to the final decision. Avoid listing multiple steps. Ensure the reasoning is clear and justifies the new pattern descriptions.]
New Pattern Descriptions:
[Please insert your new descriptions here. The new descriptions MUST BE DIRECTLY based on the key features and insights identified in the input paper. Update the descriptions ONLY for the categories where the model output differs from the target output, or if they match, update ONLY that category. Ensure that only the following categories are used: <Case_Based>, <Genetic_Algorithms>, <Neural_Networks>, <Probabilistic_Methods>, <Reinforcement_Learning>, <Rule_Learning>, <Theory>. Each updated description MUST BE enclosed within <NewPattern></NewPattern> tags. Only include descriptions for the categories being updated. If the model output and target output are the same, update ONLY that category and NO OTHERS.]
For example:
<NewPattern><Case_Based>: The papers with [your new pattern description] should be categorized as <Case_Based>.</NewPattern>
<NewPattern><Genetic_Algorithms>: The papers with [your new pattern description] should be categorized as <Genetic_Algorithms>.</NewPattern>
```
Please ONLY reply according to this format, DON'T give me any other words. If the model output and target output are the same, update ONLY that category and NO OTHERS. Now take a deep breath and provide a concise and logical Reasoning and New Pattern Descriptions.assistant

Reasoning:
The paper discusses cognitive computation as a discipline that links neurobiology, cognitive psychology, and artificial intelligence. The neighbor summary highlights the concept of learning and reasoning, specifically focusing on the Probably Approximately Correct (PAC) learning model and its generalizations. This indicates that the paper is more related to neural networks, which are a key aspect of artificial intelligence and cognitive computation. The model output, Probabilistic_Methods, does not match the target output, Neural_Networks, suggesting that the current pattern descriptions for Probabilistic_Methods are too broad and need to be updated to better capture the characteristics of papers related to neural networks.
New Pattern Descriptions:
<NewPattern><Probabilistic_Methods>: The papers that discuss Probably Approximately Correct (PAC) learning models, agnostic learning, and the tradeoff between learnability, oracle strength, and reasoning query range should be categorized as <Probabilistic_Methods>.</NewPattern>
<NewPattern><Neural_Networks>: The papers that link neurobiology, cognitive psychology, and artificial intelligence, and discuss the concept of learning and reasoning in artificial intelligence should be categorized as <Neural_Networks>.</NewPattern>

