# OpenReview forum: "Verbalized Graph Representation Learning: A Fully Interpretable Graph Model Based on Large Language Models Throughout the Entire Process"
_ICLR.cc/2025/Conference — ICLR 2025 Conference Withdrawn Submission_

### Official Review · Reviewer_gEHU · 2024-10-19

**Soundness:** 1
**Presentation:** 1
**Contribution:** 1
**Rating:** 3
**Confidence:** 5

**Summary:**

This article mainly studies how to improve the explainability of Graph Neural Networks (GNNs), particularly focusing on the joint explainability across three levels: input, training process, and decision making. The authors propose to address this problem in the text space and put forward a framework that utilizes a Large Language Model (LLM) as both a predictor and an optimizer to generate explanations in natural language.

**Strengths:**

1. This article explores an interesting question: how to explore the explainability of traditional models, such as GNNs, in the text space? To address this question, the article proposes a viable solution through performing optimization in the text space.

**Weaknesses:**

1. This article's writing suffers from vagueness, often employing ambiguous sentences that hinder readers from effectively grasping the technical details. For instance, the use of "stepping stone" in line 31 and "deeper insight" in line 44 lack concrete explanations or specific examples. Additionally, the use of certain technical terms needs to be more precise. For example, "training dynamics" in line 109 is mentioned without any subsequent elaboration or relevant content in the later sections. This lack of clarity and precision in language can significantly impede the reader's understanding of the proposed methods and contributions.
2. For lines 85-100, it's confusing that authors directly turn to the discussion of Graph LLMs without motivating its relationship to the explainability of GNNs
3. I strongly recommend that authors check the rigorous definition of explainability in machine learning, such as [1]. The author seems to misunderstand the concept of explainability.  They assume that expressing the prediction logic in natural language automatically equates to explainability. However, this is not the case.  Natural language can generate irrelevant or even misleading explanations that do not reflect the true underlying reasoning of the model.
4. Following the previous points, there's no experiment evaluating the explainability of the model but focusing on the accuracy of prediction. A case study is not a rigorous way to check the effectiveness.
5. No explainability-related work is considered in the related works part. Specifically, some highly relevant works like [2] are omitted. In terms of prompt optimization, the general philosophy is highly similar to [3].
6. The methodology part is hard to follow. I strongly recommend that the authors summarize it into some algorithms.
7. The theoretical part is a simple replication of the one in [4], which can't well explain the empirical part.

[1] Carvalho, D. V., Pereira, E. M., & Cardoso, J. S. (2019). Machine learning interpretability: A survey on methods and metrics. Electronics, 8(8), 832.

[2] Zhang, J., Liu, J., Luo, D., Neville, J., & Wei, H. (2024). LLMExplainer: Large Language Model based Bayesian Inference for Graph Explanation Generation. arXiv preprint arXiv:2407.15351.

[3] Yuksekgonul, M., Bianchi, F., Boen, J., Liu, S., Huang, Z., Guestrin, C., & Zou, J. (2024). TextGrad: Automatic" Differentiation" via Text. arXiv preprint arXiv:2406.07496.

[4] He, X., Bresson, X., Laurent, T., Perold, A., LeCun, Y., & Hooi, B. (2023). Harnessing explanations: Llm-to-lm interpreter for enhanced text-attributed graph representation learning. arXiv preprint arXiv:2305.19523.

**Questions:**

1. I do not quite understand the whole process of the methodology. Could you summarize it as an algorithm?
2. What's the "embedding" in line 268? How do you implement it?

---

### Official Review · Reviewer_U9Wf · 2024-10-31

**Soundness:** 1
**Presentation:** 1
**Contribution:** 1
**Rating:** 1
**Confidence:** 4

**Summary:**

The paper introduces an interpretable framework, VGRL, which constrains the model parameter space to text descriptions to ensure full interpretability throughout the entire process. Specifically, VGRL performs node classification on TAGs by utilizing a frozen LLM as an enhancer, predictor, optimizer, and summarizer to simulate iterative training.

**Strengths:**

This paper introduces a new way to utilize LLMs for interpretable graph learning.

**Weaknesses:**

- The experiments include only a single dataset. The authors validate their method solely on the Cora dataset, which is a small TAG. Experiments on additional TAGs, such as Citeseer and ogbn-arxiv, should be considered.
- Lack of baselines and SOTA models. The authors compare their method only with a vanilla LLM, without any comparison to existing related work. This omission prevents readers from assessing whether the proposed method represents an improvement and by how much compared to existing work.
- Section 6 (THEORETICAL ANALYSIS) and Appendix A are nearly identical to Section 4.4 and Appendix A in [1], with only variable names changed, which constitutes plagiarism. It is recommended that the authors address this issue seriously and provide clarification.

[1] Harnessing Explanations: LLM-to-LM Interpreter for Enhanced Text-Attributed Graph Representation Learning, ICLR 2024, https://arxiv.org/pdf/2305.19523

**Questions:**

See Weaknesses above.

---

### Official Review · Reviewer_57YY · 2024-11-02

**Soundness:** 2
**Presentation:** 1
**Contribution:** 2
**Rating:** 3
**Confidence:** 4

**Summary:**

This paper tries to solve two major challenges in representation learning on text-attributed graphs, including the interpretability of models and efficiency in model optimization. The former is solved by creating intuitive connections and generating textual explanations, while the latter is addressed by leveraging prompt engineering approaches.

**Strengths:**

The authors seek to develop a fully interpretable method for graph representation learning, which a promising research direction.

The authors provide open-source code for peer review.

**Weaknesses:**

The presentation requires significant improvement.  For example,

    In line 141, “And iterates over the input mini-batch B one-pass input.”
    In line 144, the definition of one-hop neighbors.

In the experimental section, the authors present only ablation studies without any comparison to SOTA methods.

While the authors emphasize interpretability and efficiency as their primary contributions, they provide no empirical results or theoretical analysis to substantiate these claims.

In the theoretical analysis section, the theorem and proof closely resemble existing work [1] but lack proper citation, which may constitute plagiarism.

[1] Xiaoxin He, Xavier Bresson, Thomas Laurent, Adam Perold, Yann LeCun, and Bryan Hooi. Harnessing explanations: Llm-to-lm interpreter for enhanced text-attributed graph representation learning. arXiv preprint arXiv:2305.19523, 2023.

**Questions:**

See weeknesses

---

### Official Review · Reviewer_4JTH · 2024-11-04

**Soundness:** 1
**Presentation:** 2
**Contribution:** 2
**Rating:** 1
**Confidence:** 4

**Summary:**

This paper discussses a training-free LLM framework VGRL for node classification on graph structured data. The core idea is to optimize the verbalization prompt called LLM optimizer.

**Strengths:**

I don't recognize a notable strength of this paper.

**Weaknesses:**

1. The paper seems to combine the recent work [1] and verbalized prompt based node classification work [2] together and only shows improvements on one obsolete graph benchmark: Cora.

2. The proposed work doesn't seem to be tailored for graph tasks, which doesn't solve challenges specific on graph strctured data. And author also don't mention graph in the two challenges they proposed.

[1] Verbalized machine learning: Revisiting machine learning with language models
[2] HARNESSING EXPLANATIONS: LLM-TO-LM INTERPRETER FOR ENHANCED TEXT-ATTRIBUTED GRAPH REPRESENTATION LEARNING

**Questions:**

Same as weakenss.

**Details Of Ethics Concerns:**

As mentioned by the other reviewer, I agree that the theoretical contribution is identical to the paper "HARNESSING EXPLANATIONS: LLM-TO-LM INTERPRETER FOR ENHANCED TEXT-ATTRIBUTED GRAPH REPRESENTATION LEARNING" appendix A at ICLR 2024.

---

### Note · Authors · 2024-11-12

I have read and agree with the venue's withdrawal policy on behalf of myself and my co-authors.